# Score-based Generative Modeling through Stochastic Evolution Equations in Hilbert Spaces

**Sungbin Lim**[1, 6, 7]\*, **Eunbi Yoon**[1]†, **Taehyun Byun**[2], **Taewon Kang**[3],
**Seungwoo Kim**[4], **Kyungjae Lee**[5], **Sungjoon Choi**[2]\*
[1]Department of Statistics, Korea University
[2]Department of Artificial Intelligence, Korea University
[3]Department of Computer Science and Engineering, Korea University
[4]Artificial Intelligence Graduate School, UNIST
[5]Department of Artificial Intelligence, Chung-Ang University
[6]LG AI Research
[7]SNU-LG AI Research Center

## Abstract

Continuous-time score-based generative models consist of a pair of stochastic differential equations (SDEs)—a forward SDE that smoothly transitions data into a noise space and a reverse SDE that incrementally eliminates noise from a Gaussian prior distribution to generate data distribution samples—are intrinsically connected by the time-reversal theory on diffusion processes. In this paper, we investigate the use of stochastic evolution equations in Hilbert spaces, which expand the applicability of SDEs in two aspects: sample space and evolution operator, so they enable encompassing recent variations of diffusion models, such as generating functional data or replacing drift coefficients with image transformation. To this end, we derive a generalized time-reversal formula to build a bridge between probabilistic diffusion models and stochastic evolution equations and propose a score-based generative model called **H**ilbert **D**iffusion **M**odel (HDM). Combining with Fourier neural operator, we verify the superiority of HDM for sampling functions from functional datasets with a power of kernel two-sample test of 4.2 on Quadratic, 0.2 on Melbourne, and 3.6 on Gridwatch, which outperforms existing diffusion models formulated in function spaces. Furthermore, the proposed method shows its strength in motion synthesis tasks by utilizing the Wiener process with values in Hilbert space. Finally, our empirical results on image datasets also validate a connection between HDM and diffusion models using heat dissipation, revealing the potential for exploring evolution operators and sample spaces.

## 1 Introduction

Score-based generative models [52] have shown success in various domains, including the generation of images [25], texts [30], videos [26], motions [31]. [52] proposes a framework for continuous-time score-based generative models that use a pair of stochastic differential equations (SDEs); a forward SDE smoothly transitions data into noise space, and a reverse SDE incrementally eliminates noise from a Gaussian prior distribution to generate samples from the data distribution. Both SDEs have the same marginal distribution when their coefficients satisfy the time-reversal formula [1, 24], which is an ingenious application of the Kolmogorov-Fokker-Planck equation for diffusion processes.

---

\*Corresponding Author. e-mail: `sungbin@korea.ac.kr`.
†This work is done at UNIST.

37th Conference on Neural Information Processing Systems (NeurIPS 2023).

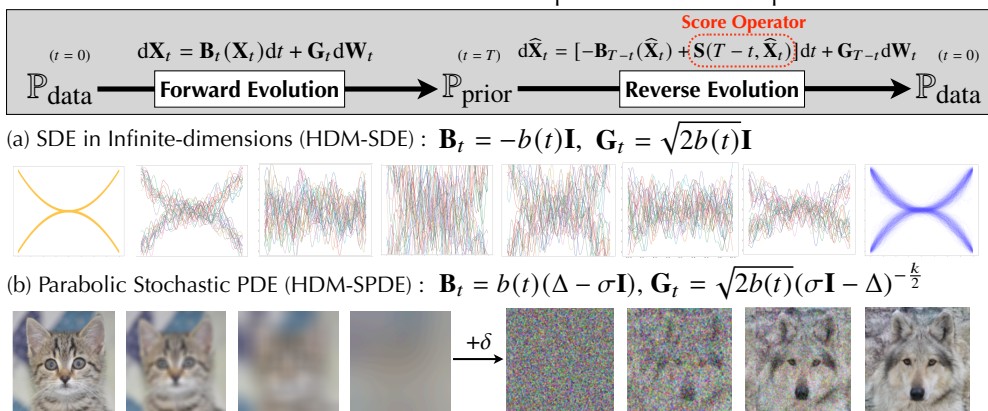

Figure 1: A unified framework with various stochastic evolution equations to yield score-based generative models for different datasets. (a) SDE in infinite-dimensional spaces can generate functions as samples from the function space. (b) Utilizing a prior distribution generated from heat dissipation-type forward evolution $\mathbf{B}_t = b(t)(\Delta - \sigma\mathbf{I})$, the same as IHDM [50], we can generate image samples by adding Gaussian noise ($\delta$) and executing reverse evolution through the parabolic SPDE.

Recently, there has been an active proposal of variations of diffusion models, such as generating functional data [18, 19, 23, 29, 38, 48, 49] or replacing drift coefficients with image transformation [4, 27, 50], which cannot be covered by the original SDE framework [52] formulated in the Euclidean space. Motivated by this problem, we propose a unified framework for continuous-time score-based generative models by using *stochastic evolution equations* in Hilbert spaces [6, 14, 34, 35],

$$d\mathbf{X}_t = \mathbf{B}_t(\mathbf{X}_t)dt + \mathbf{G}_t d\mathbf{W}_t, \quad d\widehat{\mathbf{X}}_t = [-\mathbf{B}_{T-t}(\widehat{\mathbf{X}}_t) + \mathbf{S}(T-t, \widehat{\mathbf{X}}_t)]dt + \mathbf{G}_{T-t} d\mathbf{W}_t, \quad (1)$$

where $(\mathbf{X}_t, \widehat{\mathbf{X}}_t)$ is a pair of forward and reverse stochastic processes with evolution operators $(\mathbf{B}_t, \mathbf{G}_t)$, $\mathbf{W}_t$ is a Wiener process with values in some Hilbert space $\mathcal{H}$, and $\mathbf{S}(t, \cdot) : [0, T] \times \mathcal{H} \to \mathcal{H}$ is a *score operator*, which generalizes the notion of score functions in the Euclidean space. Since we can choose $\mathcal{H}$ and $(\mathbf{B}_t, \mathbf{G}_t)$ in an abstract setting, stochastic equations (1) naturally extend the use of original SDEs to the infinite-dimensional setting and include stochastic partial differential equations (SPDEs), such as a parabolic equation driven by white noise (see Figure 1). Consequently, we can expand the SDE framework [52] in two aspects: sample space and evolution operator, enabling it to encompass recent variations of diffusion models in a continuous-time score-based generative model.

**Contributions**   An essential key to identifying the relationship between the forward process $\mathbf{X}_t$ and the reverse process $\widehat{\mathbf{X}}_t$ in (1) is the derivation of *time reversal formula* with *time-dependent* operators $\mathbf{B}_t$ and $\mathbf{G}_t$, which has been a challenging problem [23, 29, 38, 49] when we utilize coefficient schedulings [46, 52]. Contrary to previous approaches using semigroup theory [14], our approach is grounded in Kolmogorov equations with functional derivatives, well-studied in [2, 3, 7, 15–17], so that we can consider a wide scope of stochastic evolution equations with time-dependent operators.

- We derive a generalized time-reversal formula and unify the SDE framework [52] with stochastic equations in Hilbert spaces. Consequently, we propose a class of continuous-time score-based generative models, **H**ilbert **D**iffusion **M**odel (HDM), which enables using SDEs in infinite-dimensional settings (HDM-SDE) and parabolic SPDEs (HDM-SPDE).
- Combining with neural operators [37], HDM-SDE shows the superiority for sampling functions from functional datasets with multi-modalities compared to GP [55], NP [20], DDPM [25], and other diffusion models formulated in function spaces, NDP [18] and SP-SGM [48]. We also validate that HDM-SDE plays a crucial role in successful human motion synthesis compared to DDPM, by utilizing the Wiener process in the Hilbert space.
- We build a bridge between parabolic SPDE and diffusion models using heat dissipation [50]. HDM-SPDE improves the performance in image generation as a continuous-time version of IHDM. Also, the proposed method shows the inductive bias in controllable coarse-to-fine image generation, e.g., inpainting, deblurring, and depixelation, without additional training.

# 2 A Foundation on Time-Reversal of Stochastic Evolution Equations

We first present a preliminary theory on the stochastic evolution equations in Hilbert space and introduce the logarithmic derivatives of probability measures to propose a Hilbert space valued *score operator* (see Section 2.2) which generalizes the notion of score functions in the Euclidean space. Then we derive the *time-reversal formula* (Theorem 2.1), which is a key to obtain the time-reversal of the diffusion processes by using the Kolmogorov equations in Hilbert space.

## 2.1 Preliminaries: Stochastic Evolution Equations in Hilbert Spaces

In preliminaries, we introduce elements of stochastic evolution equations in Hilbert spaces. See B.1 or we refer [5, 6, 14, 34, 35] for basic setting.

**Gelfand Triple and Cameron-Martin Space**  Let $\mathcal{H}$ be a separable Hilbert space with the $\mathcal{H}$-norm defined by the inner product $\| \cdot \|_{\mathcal{H}} = \sqrt{\langle \cdot, \cdot \rangle_{\mathcal{H}}}$. Let $\mathcal{L}(\mathcal{H})$ denotes the set of all bounded linear operators on $\mathcal{H}$. We assume there exists a Radon centered Gaussian measure $\lambda$ on the Borel $\sigma$-field $\mathcal{B}(\mathcal{H})$ with the nonnegative, symmetric, self-adjoint, and the finite trace covariance operator $\Lambda \in \mathcal{L}(\mathcal{H})$, hence there exists eigensystem $\{(\lambda^{(\ell)}, \phi^{(\ell)}) \in \mathbb{R}_+ \times \mathcal{H} : \ell \in \mathbb{N}\}$ such that $\Lambda(\phi^{(\ell)}) = \lambda^{(\ell)} \phi^{(\ell)}$ holds and $\mathrm{Tr}(\Lambda) = \sum_{\ell=1}^{\infty} \lambda^{(\ell)} < \infty$. We define *Cameron-Martin space* by $\mathcal{H}_\lambda := \Lambda^{1/2}(\mathcal{H})$ with the inner product $\langle h_1, h_2 \rangle_{\mathcal{H}_\lambda} := \langle \Lambda^{-1/2}(h_1), \Lambda^{-1/2}(h_2) \rangle_{\mathcal{H}}$. Let $\mathcal{H}^* \subset \mathcal{H}_\lambda^*$ be the dual space of $\mathcal{H}$ and $\mathcal{H}_\lambda$, respectively. By identifying $\mathcal{H}_\lambda$ with $\mathcal{H}_\lambda^*$ via the Riesz isomorphism, we can consider a continuous inclusion map $i_{\mathcal{H}_\lambda} : \mathcal{H}^* \to \mathcal{H}_\lambda$ such that

$$\varphi(\xi) = \langle i_{\mathcal{H}_\lambda}(\varphi), \xi \rangle_{\mathcal{H}_\lambda} = \langle \Lambda^{-1/2}(i_{\mathcal{H}_\lambda}(\varphi)), \Lambda^{-1/2}(\xi) \rangle_{\mathcal{H}}, \quad \varphi \in \mathcal{H}^*, \xi \in \mathcal{H}_\lambda. \tag{2}$$

Thus, we can also identify $\mathcal{H}^*$ with $\Lambda^{1/2}(\mathcal{H}_\lambda)$ (see [8, Example 1.1]). For notational convenience, we write $i_{\mathcal{H}_\lambda}(\varphi)$ as $\Lambda(\varphi)$ by regarding $\varphi$ as an element in $\mathcal{H}$. In summary,

$$\mathcal{H}_\lambda = \Lambda^{1/2}(\mathcal{H}), \quad i_{\mathcal{H}_\lambda}(\mathcal{H}^*) = \Lambda(\mathcal{H}), \quad i_{\mathcal{H}_\lambda}(\mathcal{H}^*) \subset \mathcal{H}_\lambda \subset \mathcal{H}. \tag{3}$$

We refer to $(\mathcal{H}^*, \mathcal{H}_\lambda, \mathcal{H})$ as *Gelfand triple* which provides groundwork for the SDE theory in $\mathcal{H}$.

**Example 2.1** (Euclidean space). Let $\mathcal{H} := \mathbb{R}^d$ and $\lambda = \mathcal{N}(0, \Sigma)$ be a Gaussian measure with a positive-definite covariance matrix $\Sigma \in \mathbb{R}^{d \times d}$. Then $\left| \Sigma^{-1/2} \mathbf{x} \right| < \infty$ for $\mathbf{x} \in \mathbb{R}^d$, hence $\mathcal{H}_\lambda = \mathcal{H}$. $\square$

**Example 2.2** (Reproducing kernel Hilbert space). Let $\mathcal{H}$ denote the class of square-integrable functions on finite measure space $(\mathcal{X}, \mu_{\mathcal{X}})$ with the usual $L_2$-norm, $\|f\|_{\mathcal{H}} := \left( \int_{\mathcal{X}} |f(x)|^2 \mu_{\mathcal{X}}(\mathrm{d}x) \right)^{1/2}$. Due to the assumption on $\Lambda$, there exists symmetric and positive definite kernel function $K_\lambda$ such that

$$\Lambda(f)(\mathbf{x}) = \int_{\mathcal{X}} K_\lambda(\mathbf{x}, \mathbf{y}) f(\mathbf{y}) \mu_{\mathcal{X}}(\mathrm{d}\mathbf{y}), \quad \mathbf{x} \in \mathcal{X}, \ f \in \mathcal{H}. \tag{4}$$

Hence Cameron-Martin space $\mathcal{H}_\lambda$ is reproducing kernel Hilbert space (RKHS) with $K_\lambda$. $\square$

**Wiener Processes and Stochastic Evolution Equations in Hilbert Spaces**  For $T > 0$, let $(\mathbf{W}_t)_{t \in [0,T]}$ be $\mathcal{H}$-valued $\mathcal{F}_t$-adapted $\Lambda$-Wiener process (see Definition B.1 for details). By the Kosambi-Karhunen-Loève theorem, $\mathbf{W}_t$ has a series representation:

$$\mathbf{W}_t := \Lambda^{1/2}(\mathcal{W}_t) = \sum_{\ell=1}^{\infty} \sqrt{\lambda^{(\ell)}} W_t^{(\ell)} \phi^{(\ell)}, \quad t \in [0,T], \tag{5}$$

where $\mathcal{W}_t = \{W_t^{(\ell)} : \ell \in \mathbb{N}\}$ is *white noise* of which the components $W_t^{(\ell)}$ are independent real-valued Wiener processes. Then $\mathrm{Cov}(\mathbf{W}_t) = t\Lambda$ holds and $\mathbb{E}\langle \mathbf{W}_t, \mathbf{W}_s \rangle_{\mathcal{H}} = \min\{t, s\} \cdot \mathrm{Tr}(\Lambda) < \infty$ since $\Lambda$ has the finite trace. If $\mathcal{H}_\lambda$ is RKHS with $K_\lambda$ (Example 2.2), $\mathbf{W}_t$ preserves spatial correlation,

$$\mathbb{E}[\mathbf{W}_t(\mathbf{x}_1) \mathbf{W}_s(\mathbf{x}_2)] = \min\{t, s\} K_\lambda(\mathbf{x}_1, \mathbf{x}_2), \quad \mathbf{x}_1, \mathbf{x}_2 \in \mathcal{X}. \tag{6}$$

Let $\mathcal{L}_2(\mathcal{H})$ denote the class of Hilbert-Schmidt operator on $\mathcal{H}$. Following the usual construction of the stochastic integral with respect to $\mathbf{W}_t$ (e.g., see [14, Section 4] and [35, Section 2]), we can define the following stochastic evolution equation in $\mathcal{H}$,

$$\mathbf{X}_t = \mathbf{X}_0 + \int_0^t \mathbf{B}_s(\mathbf{X}_s) \mathrm{d}s + \int_0^t \mathbf{G}_s \mathrm{d}\mathbf{W}_s, \quad \mathbf{X}_0 \in \mathcal{H}, \quad t \in [0,T], \tag{7}$$

where $\mathbf{B}_t : [0,T] \times \mathcal{H} \to \mathcal{H}$ and $\mathbf{G}_t : [0,T] \to \mathcal{L}_2(\mathcal{H})$ are progressively measurable satisfying regular conditions [35, Section 3.2] which guarantee solvability in $\mathcal{H}$. Under these assumptions, there exists a unique $\mathcal{H}$-valued solution $\mathbf{X}_t$ to (7) for every $t \in [0,T]$ such that $\mathbf{X}_0 \in \mathcal{H}$ and $\mathbf{W}_t \in \mathcal{H}$ are independent. We assume $\mathbf{G}_t$ is constant on $\mathcal{H}$, which still recovers the SDE framework of [52].

## 2.2 Score Operator and Time-Reversal Formula

Now we consider a time-reversal $\widehat{\mathbf{X}}_t := \mathbf{X}_{T-t}$ of (7) and its corresponding diffusion process,

$$\widehat{\mathbf{X}}_t = \widehat{\mathbf{X}}_0 + \int_0^t \widehat{\mathbf{B}}_s(\widehat{\mathbf{X}}_s)\mathrm{d}t + \int_0^t \widehat{\mathbf{G}}_s\mathrm{d}\mathbf{W}_s, \quad \widehat{\mathbf{X}}_0 \sim \mathcal{N}(0,\Lambda), \quad t \in [0,T]. \tag{8}$$

A condition for the existence of the time-reversal process (8) is well-studied for diffusion processes in abstract settings (e.g., see [12]), hence we focus on identifying evolutionary operators $\widehat{\mathbf{B}}$ and $\widehat{\mathbf{G}}$.

**Logarithmic Derivative of Probability Measure and Score Operator** A score function in the Euclidean space is defined by a gradient of logarithmic probability density $\nabla_x \log p_t(x)$. For Hilbert spaces, we employ logarithmic derivatives of Fomin differentiable probability measures [8, 9]. Let us define the class of smooth cylinder functions [10], which is dense in the Sobolev space on $\mathcal{H}$, by

$$\mathcal{FC}_b^\infty := \{f_{\varphi_{1:m}} = f(\varphi_1, \ldots, \varphi_m) : \mathcal{H} \to \mathbb{R} : f \in C_b^\infty(\mathbb{R}^m), \varphi_i \in \mathcal{H}^*, m \in \mathbb{N}\}. \tag{9}$$

For $f_{\varphi_{1:m}} \in \mathcal{FC}_b^\infty$, we compute the $\mathcal{H}_\lambda$-gradient at $u \in \mathcal{H}$ in the direction of $\xi \in \mathcal{H}_\lambda$ by

$$\langle Df_{\varphi_{1:m}}(u), \xi \rangle_{\mathcal{H}_\lambda} = \sum_{i=1}^m \partial_i f\left(\varphi_1(u), \ldots, \varphi_m(u)\right) \langle \xi, \Lambda(\varphi_i) \rangle_{\mathcal{H}_\lambda}, \quad u \in \mathcal{H}, \tag{10}$$

where $Df_{\varphi_{1:m}}$ denotes a Gâteaux derivative. A differentiable measure $\mu$ on $\mathcal{H}$ along a vector $\xi \in \mathcal{H}_\lambda$ has the *partial logarithmic derivative* $\rho_\xi^\mu \in L_1(\mathcal{H},\mu)$ if the following integration by parts holds,

$$\int_{\mathcal{H}} \langle Df_{\varphi_{1:m}}(u), \xi \rangle_{\mathcal{H}_\lambda} \mu(\mathrm{d}u) = - \int_{\mathcal{H}} f_{\varphi_{1:m}}(u) \rho_\xi^\mu(u) \mu(\mathrm{d}u), \quad f_{\varphi_{1:m}} \in \mathcal{FC}_b^\infty. \tag{11}$$

If there exists a Borel map $\rho_{\mathcal{H}_\lambda}^\mu(\cdot) : \mathcal{H} \to \mathcal{H}$ such that $\langle \rho_{\mathcal{H}_\lambda}^\mu(u), \xi \rangle_{\mathcal{H}_\lambda} = \rho_\xi^\mu(u)$ for $\xi \in \mathcal{H}_\lambda$, then $\rho_{\mathcal{H}_\lambda}^\mu$ is called the *vector logarithmic derivative* of $\mu$ associated with $\mathcal{H}_\lambda$. For instance, if $\mu$ is a Gaussian measure with mean $m_\mu \in \mathcal{H}$ and covariance operator $\Lambda_\mu \in \mathcal{L}(\mathcal{H})$ satisfying $\Lambda(\mathcal{H}) \subset \Lambda_\mu(\mathcal{H})$, then $\rho_{\mathcal{H}_\lambda}^\mu$ is computed by (see [8, Example 2.5] and [14, Proposition 2.26]),

$$\rho_{\mathcal{H}_\lambda}^\mu(u) = \Lambda \circ \Lambda_\mu^{-1}(m_\mu - u), \quad u \in \mathcal{H}, \tag{12}$$

which generalizes the score function $\nabla_{\mathbf{x}} \log p(\mathbf{x})$ of a Gaussian density $p(\mathbf{x}) \sim \mathcal{N}(m_\mu, \Sigma_\mu)$. For a sequence of probability measures $(\mu_t)_{t \in [0,T]}$ of the solution to (7) with $\mu_0 = \mathbb{P}_{\mathrm{data}}$, we define a *score operator* $\mathbf{S}_\lambda(t, \cdot) : [0,T] \times \mathcal{H} \to \mathcal{H}$ by multiplying $\mathbf{G}_t\mathbf{G}_t^*$ to the vector logarithmic derivative $\rho_{\mathcal{H}_\lambda}^{\mu_t}$,

$$\mathbf{S}_\lambda(t,u) := \mathbf{G}_t\mathbf{G}_t^* \rho_{\mathcal{H}_\lambda}^{\mu_t}(u). \tag{13}$$

Hence the score operator $\mathbf{S}_\lambda(t, \cdot)$ is analogous to $g^2(t)\nabla_{\mathbf{x}} \log p_t(\mathbf{x})$ in the reverse SDE of [52].

**Kolmogorov Equations and Time-Reversal Formula** Score-based generative model through SDE is grounded on the Fokker-Planck-Kolmogorov equations [1, 24]. Following [5, 6, 8–10], we define *Kolmogorov operator* $\mathscr{L}_t$ on the class of smooth cylinder functions $\mathcal{FC}_b^\infty$ by

$$\mathscr{L}_t f_{\varphi_{1:m}}(u) := \frac{1}{2}\mathrm{Tr}_{\mathcal{H}_\lambda}(\mathbf{A}_t(u) \circ \Lambda \circ D^2 f_{\varphi_{1:m}}(u)) + \langle Df_{\varphi_{1:m}}(u), \mathbf{B}_t(u) \rangle_{\mathcal{H}_\lambda}, \tag{14}$$

where $\mathbf{A}_t := \mathbf{G}_t\mathbf{G}_t^*$ and $D^2 f_{\varphi_{1:m}}$ denotes the second-order Gâteaux derivative. The Kolmogorov operators $(\mathscr{L}_t)_{t \in [0,T]}$ describes evolution of $(\mu_t)_{t \in [0,T]}$ and becomes a *generator* of Markov diffusion process [5, 6, 24]. Now we present our main result on the time-reversal formula.

**Theorem 2.1** (Time-Reversal Formula). *Time reversal* $\widehat{\mathbf{X}}_t$ *satisfying* (8) *has generator*

$$\widehat{\mathscr{L}}_t f_{\varphi_{1:m}}(u) := \frac{1}{2}\mathrm{Tr}_{\mathcal{H}_\lambda}(\widehat{\mathbf{A}}_t \circ \Lambda \circ D^2 f_{\varphi_{1:m}}(u)) + \langle Df_{\varphi_{1:m}}(u), \widehat{\mathbf{B}}_t(u) \rangle_{\mathcal{H}_\lambda}, \tag{15}$$

*where* $\widehat{\mathbf{A}}_t$ *and* $\widehat{\mathbf{B}}_t$ *satisfy the following **time-reversal formula***

$$\widehat{\mathbf{A}}_t = \mathbf{A}_{T-t}, \quad \widehat{\mathbf{B}}_t(u) = -\mathbf{B}_{T-t}(u) + \mathbf{S}_\lambda(T-t,u). \tag{16}$$

See Appendix B.1 for the proof of Theorem 2.1. The proof is based on the Kolmogorov forward equation and the Kolmogorov backward equation with functional derivatives to show that (15) is the generator of the time-reversal diffusion process (8), which is analogous to the proof in the Euclidean space [24]. Due to (16), we can rewrite (8) by

$$\widehat{\mathbf{X}}_t = \widehat{\mathbf{X}}_0 + \int_0^t \left[ -\mathbf{B}_{T-s}(\widehat{\mathbf{X}}_s) + \mathbf{S}_\lambda(T - s, \widehat{\mathbf{X}}_s) \right] \mathrm{d}s + \int_0^t \mathbf{G}_{T-s}\mathrm{d}\mathbf{W}_s. \tag{17}$$

Hence we can generate a sample from $\widehat{\mathbf{X}}_T \sim \mathbb{P}_{\text{data}}$ by running (17) if we obtain $\mathbf{S}_\lambda(t, \cdot)$ for $t \in [0, T]$.

## 2.3 Applications of Theorem 2.1

This section presents two important applications of Theorem 2.1; SDEs in infinite-dimensional space and parabolic SPDEs in finite-dimensional space. We utilize these stochastic evolution equations to propose a unified framework of continuous-time score-based generative models in Section 3.

### 2.3.1 SDEs in Infinite-Dimensions

We first generalize the SDE framework [52] in the infinite-dimensions based on Theorem 2.1. Let $\{\lambda^{(\ell)}, \phi^{(\ell)}\}_{\ell \in \mathbb{N}}$ be an eigensystem of $\mathcal{H}$ and consider the following SDE system with values in $\mathcal{H}$,

$$\mathrm{d}X_t^{(\ell)} = b_t^{(\ell)} X_t^{(\ell)} \mathrm{d}t + \sqrt{\lambda^{(\ell)}} \sigma_t^{(\ell)} \mathrm{d}W_t^{(\ell)}, \quad t \in [0, T], \ell \in \mathbb{N}, \tag{18}$$

where $b_t^{(\ell)} : [0, T] \to \mathbb{R}$ and $\sigma_t^{(\ell)} : [0, T] \to \mathbb{R}_+$ are real-valued functions for each $\ell \in \mathbb{N}$. Set $\beta_t^{(\ell)} := \exp\left(\int_0^t b_s^{(\ell)}\mathrm{d}s\right)$ and $\Sigma_\beta^{(\ell)}(t) := \left(\sigma_t^{(\ell)}/\beta_t^{(\ell)}\right)^2$. Conditioning at $X_0^{(\ell)}$, the law of $X_t^{(\ell)}$ follows Gaussian distribution, hence we can apply [44, Theorem 5.3] to (18), which induces $\widehat{\mathbf{B}}_t(\mathbf{x})$ by

$$\widehat{\mathbf{B}}_t(\mathbf{x}) = \sum_{\ell \in \mathbb{N}} \left( -b_{T-t}^{(\ell)} x^{(\ell)} + \Sigma_\beta^{(\ell)}(T - t) \frac{\beta_{T-t}^{(\ell)} X_0^{(\ell)} - x^{(\ell)}}{\int_0^{T-t} \Sigma_\beta^{(\ell)}(s)\mathrm{d}s} \right). \tag{19}$$

Indeed, (19) is a special case of (16) (see Appendix C.1).

**Example 2.3** (VP-SDE and sub-VP-SDE in Infinite-Dimensions). Set $b_t^{(\ell)} = -\frac{1}{2}b(t)$ and $\sigma_t^{(\ell)} = \sqrt{b(t)}$ for every $\ell \in \mathbb{N}$. Then we have $\beta_t = e^{-\frac{1}{2}\int_0^t b(s)\mathrm{d}s}$ and $\Sigma_\beta(t) = b(t)e^{\int_0^t b(s)\mathrm{d}s}$. Thus,

$$m_{\mu_t|\mathbf{X}_0} = e^{-\frac{1}{2}\int_0^t b(s)\mathrm{d}s}\mathbf{X}_0, \quad \Lambda_{\mu_t|\mathbf{X}_0} = (1 - e^{-\int_0^t b(s)\mathrm{d}s})\Lambda. \tag{20}$$

Hence we can compute the conditional score function $\mathbf{S}_\lambda(t, \mathbf{X}_t|\mathbf{X}_0)$ by using (19). As a result, (20) generalizes VP-SDE [52] in infinite-dimensions. Similarly, we can recover the sub-VP-SDE in infinite-dimensions, hence (19) recovers SDEs proposed in [52] when $\Lambda = \text{Id}$ and $\mathcal{H} = \mathbb{R}^d$.

### 2.3.2 Parabolic SPDEs

We introduce parabolic SPDEs [34, 35] based on Theorem 2.1. For $k > -1$, we define

$$\mathrm{d}u_t = b(t)[\Delta u_t - \sigma u_t]\mathrm{d}t + \sqrt{2b(t)}(\sigma\mathbf{I} - \Delta)^{-\frac{k}{2}}\mathrm{d}\mathcal{W}_t, \quad t \in [0, T], \tag{21}$$

on a bounded domain $\mathcal{O}$ with a zero-Neumann boundary condition. By following [50], we use the discrete cosine transformation (DCT) from a coordinate vector $u_t$ expressed in the standard basis to a coordinate vector $\hat{u}_t$, with the eigendecomposition $\Delta \triangleq \mathbf{VDV}^\top$. Hence we can rewrite (21) as

$$\mathrm{d}\hat{u}_t = b(t)(\mathbf{D}\hat{u}_t - \sigma\hat{u}_t)\mathrm{d}t + \sqrt{2b(t)}(\sigma\mathbf{I} - \mathbf{D})^{-\frac{k}{2}}\mathrm{d}\widehat{\mathcal{W}}_t, \quad t \in [0, T], \tag{22}$$

where $\hat{u}_t = \mathbf{V}^\top u_t$ and $\widehat{\mathcal{W}}_t = \mathbf{V}^\top \mathcal{W}_t$. Since $\mathbf{V}$ is orthogonal, we have $\widehat{\mathcal{W}}_t \overset{d}{=} \mathcal{W}_t$ for each $t$, so we can regard $\widehat{\mathcal{W}}_t$ as standard space-time white noise. Thus, we can apply Theorem 2.1 to (22) with $\mathbf{B}_t = b(t)(\mathbf{D} - \sigma\mathbf{I})$ and $\mathbf{G}_t = \sqrt{2b(t)}(\sigma\mathbf{I} - \mathbf{D})^{-\frac{k}{2}}$ in $\mathbb{R}^d$ (see Appendix C.2). Therefore,

$$\widehat{\mathbf{B}}_t = b(T - t)(\sigma\mathbf{I} - \mathbf{D}) + 2b(T - t)(\sigma\mathbf{I} - \mathbf{D})^{-k}\rho_{\mathcal{H}_\lambda}^{\hat{\mu}_{T-t}}. \tag{23}$$

If we set $b(t) = 1$, $\sigma = 0$, and $k = 0$, then (21) becomes stochastic heat equation (SHE) and has a connection between the generative process of IHDM [50]. Indeed, the mild solution of SHE is

$$u_t = e^{\Delta t}u_0 + \int_0^t e^{\Delta(t-s)}\mathrm{d}\mathcal{W}_s, \quad t \in [0, T], \tag{24}$$

where $e^{\Delta t}$ is a $C_0$-semigroup, which induces the mean vector of $u_t$ as $e^{\Delta t}u_0 = \mathbf{V}\exp(\mathbf{D}t)\mathbf{V}^\top u_0$, and the remaining stochastic integration part becomes Gaussian process [14, Theorem 5.2]. If we consider a continuous-time version of IHDM with a variance scheduling, then (23) induces the appropriate reverse SPDE for sampling. See Section 4.3 for empirical results.

## 3 Hilbert Diffusion Models

This section is devoted to describing Hilbert Diffusion Models (HDMs), a class of score-based generative models based on Theorem 2.1 and stochastic evolution equations presented in Section 2.3. We introduce two versions of HDM, HDM-SDE and HDM-SPDE, which utilize SDEs in infinite-dimensions (18) and parabolic SPDEs (21), respectively.

### 3.1 Estimating Score Operators

To estimate the score operator $\mathbf{S}_\lambda(t, \cdot)$ in Hilbert space, we propose a time-dependent score model $\mathbf{s}_\theta(t, \cdot) : [0, T] \times \mathcal{H} \to \mathcal{H}$ parameterized with $\theta \in \mathbb{R}^p$, which is trained by the following objective,

$$\theta^* = \arg\min_{\theta \in \mathbb{R}^p} \mathbb{E}_{t \sim \mathcal{U}[0,T]}\mathbb{E}_{\mathbf{X}_t \sim \mu_t}\|\mathbf{s}_\theta(t, \mathbf{X}_t) - \mathbf{S}_\lambda(t, \mathbf{X}_t)\|_{\mathcal{H}}^2, \tag{25}$$

where $\mathcal{U}$ denotes the uniform distribution on $[0, T]$. Due to [38, 54], note that minimizing (25) is equivalent to minimizing the following objective,

$$\theta^* = \arg\min_{\theta \in \mathbb{R}^p} \mathbb{E}_{t \sim \mathcal{U}[0,T]}\mathbb{E}_{\mathbf{X}_0 \sim \mathbb{P}_{\text{data}}}\mathbb{E}_{\mathbf{X}_t \sim \mu_t|\mathbf{X}_0}\|\mathbf{s}_\theta(t, \mathbf{X}_t) - \mathbf{S}_\lambda(t, \mathbf{X}_t|\mathbf{X}_0)\|_{\mathcal{H}}^2, \tag{26}$$

where $\mu_t|\mathbf{X}_0$ denotes the conditional distribution of $\mathbf{X}_t$ conditioned at $\mathbf{X}_0 \sim \mathbb{P}_{\text{data}}$, and $\mathbf{S}_\lambda(t, \mathbf{X}_t|\mathbf{X}_0)$ is the corresponding score operator, which is estimated by a time-dependent neural network.

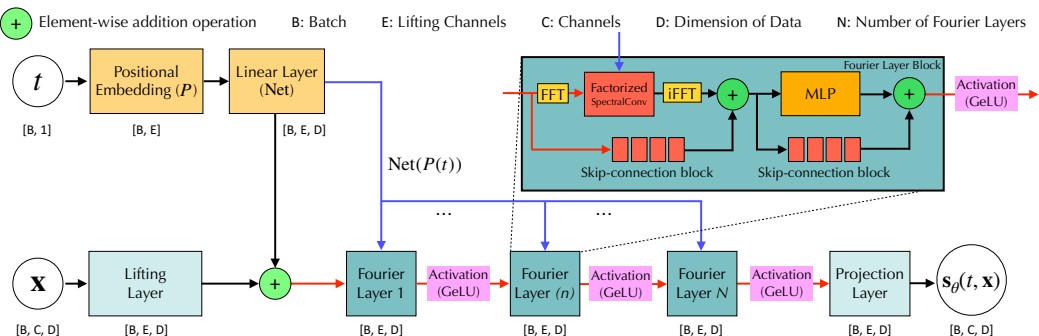

Figure 2: The architecture of time-conditioned FNO.

**Architectures of HDM** We use different architectures for modeling HDM-SDE and HDM-SPDE since the corresponding stochastic evolution equations (18) and (21) have distinct values in infinite-dimensions and Euclidean space, respectively. For HDM-SDE, same as other diffusion models in infinite-dimensions [23, 29, 38], we modify Fourier Neural Operator (FNO) [37], called *time-conditioned* FNO, by adding a positional embedding and a linear layer to embed the time variable $t$ (see Figure 2). The spatial variable $\mathbf{x}$ first passes through the lifting layer, increasing its dimensions to match the lifting channels. Then, the embedded time variable undergoes the linear layer to adjust its dimensions to match the lifted spatial variable, and is pointwisely added. The Fourier layer takes the input and applies Fast Fourier Transformation (FFT) to transform it to the frequency space, followed by spectral convolution. See Appendix C.3 for more details.

For HDM-SPDE, we use U-Net [51] similar to other diffusion models [25, 52], mostly varying the number of multipliers of resolutions and the number of residual blocks. Detailed architecture parameters according to datasets are described in Table A.3.

## 3.2 Training and Sampling

**Loss Functions**    Based on (26), both HDM-SDE and HDM-SPDE use the mean-square loss as the training loss function. Fix $\mathbf{x}_0 \sim \mathbb{P}_{\text{data}}$ and let $\mathbf{x}_t$ be a generated sample at time $t$ from $\mu_t | \mathbf{x}_0$ so that $\mathbf{x}_t \sim \mathcal{N}(m_{\mu_t | \mathbf{x}_0}, \Lambda_{\mu_t | \mathbf{x}_0})$. Then the loss functions are represented by

$$L_{\text{SDE}}(\theta) = \mathbb{E}_{t \sim \mathcal{U}(0,T]} \mathbb{E}_{\mathbf{x}_0 \sim \mathbb{P}_{\text{data}}} \mathbb{E}_{\epsilon \sim \mathcal{N}(0,\mathbf{I})} \left\| \mathbf{s}_\theta(t, \mathbf{x}_t) + \beta(t) \Lambda_{\mu_t | \mathbf{x}_0}^{1/2}(\epsilon) \right\|_2^2, \qquad (27)$$

$$L_{\text{SPDE}}(\theta) = \mathbb{E}_{t \sim \mathcal{U}[0,T]} \mathbb{E}_{\mathbf{x}_0 \sim \mathbb{P}_{\text{data}}} \mathbb{E}_{\epsilon \sim \mathcal{N}(0,\mathbf{I})} \left\| \mathbf{s}_\theta(t, \mathbf{x}_t) + \mathbf{V}\left( \beta(t) \Lambda_{\mu_t | \mathbf{x}_0}^{1/2}(\epsilon) \right) \right\|_2^2. \qquad (28)$$

Here we use the cosine beta schedule [46] with $\beta(t) = \frac{2}{1+s} \frac{\pi}{2} \tan\left( \frac{t+s}{1+s} \frac{\pi}{2} \right)$ for both HDM-SDE and HDM-SPDE for a small number $s > 0$.

**Solving Reverse Stochastic Evolution Equations**    After training a score model $\mathbf{s}_\theta(t, \cdot)$ for all $t \in [0, T]$, we can generate samples from $\mathbb{P}_{\text{data}}$ by running the reverse process (17) with the trained score model instead of $\mathbf{S}_\lambda(t, \cdot)$. Numerically, we utilize the Euler-Maruyama method to approximate a solution of (17) starting at $\widehat{\mathbf{X}}_{t_0} \sim \mathcal{GP}(0, \Lambda)$ (see Algorithms 1 and 2),

$$\widehat{\mathbf{X}}_{t_{m+1}} = \widehat{\mathbf{X}}_{t_m} + \left( -\mathbf{B}_{T-t_m}(\widehat{\mathbf{X}}_{t_m}) + \mathbf{s}_\theta(T - t_m, \widehat{\mathbf{X}}_{t_m}) \right) \Delta t + \sqrt{\Delta t} \mathbf{G}_{T-t_m} \mathcal{N}_{t_m}, \qquad (29)$$

where $\mathcal{N}_{t_m} \overset{\text{i.i.d.}}{\sim} \mathcal{GP}(0, \Lambda)$, and $\Delta t := t_{m+1} - t_m$ for $m \in \{0, \dots, M\}$ with $t_0 = 0$ and $t_M = T$.

# 4   Experiments

This section presents the experimental settings and results that apply HDM-SDE to generating function and motion and HDM-SPDE to image generation. For more details, see Appendix D.

## 4.1   Function Generation through HDM-SDE

**Datasets and Implementation**    We validate HDM-SDE for generating functions from 1D functional datasets; Quadratic, Melbourne, and Gridwatch, to follow the setting in [48]. We refer to Section D.1 for more details about datasets and settings. We tune the kernel hyperparameter of `len` parameter for 0.8 in Quadratic, 2.0 in Melbourne, and 1.8 in Gridwatch. We commonly use `gain` parameter 1.0 for all datasets. `Grid` parameter adjusted to 100 in Quadratic, 24 in Melbourne, and 288 in Gridwatch. In the Quadratic and Melbourne datasets, we train our model during 1K iterations (1.5K in Gridwatch). For sampling, we set the number of sampling steps as 1,000. For a quantitative evaluation, we calculate the power of a kernel two-sample test [56] based on Maximum Mean Discrepancy (MMD).

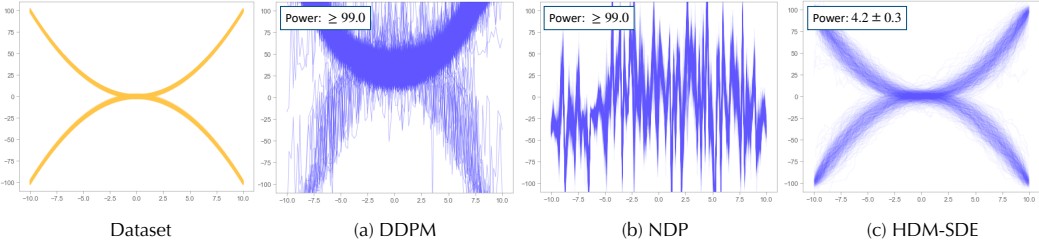

Figure 3: Comparison of functional samples generated by DDPM (VP-SDE) [25], NDP [18], and HDM-SDE (ours) on Quadratic dataset.

**Results**    Table 1 shows the results of the test power analysis, and HDM-SDE demonstrates excellence compared to Gaussian process [55], Neural process [20], DDPM (VP-SDE) [25], and other diffusion models (NDP [18] and SP-SGM [48]) formulated on function spaces. Our method records a power(%) of $4.2_{\pm 0.3}$ on Quadratic, $0.2_{\pm 0.1}$ on Melbourne, and $3.6_{\pm 0.0}$ on Gridwatch. Figures 3 and A.1 show that HDM-SDE method has outperformed results compared to the diffusion models utilizing the

|  | GP [55] | NP [20] | DDPM [25] | NDP [18] | SP-SGM [48] | HDM-SDE (Ours) |
|---|---|---|---|---|---|---|
| Quadratic | $100.0_{\pm 0.0}$ | $8.6_{\pm 1.5}$ | $\geq 99.0$ | $\geq 99.0$ | $5.4_{\pm 0.7}$ | $\mathbf{4.2_{\pm 0.3}}$ |
| Melbourne | $20.1_{\pm 4.0}$ | $10.1_{\pm 1.9}$ | $3.3_{\pm 0.2}$ | $12.8_{\pm 0.4}$ | $5.3_{\pm 0.7}$ | $\mathbf{0.2_{\pm 0.1}}$ |
| Gridwatch | $29.2_{\pm 5.5}$ | $51.8_{\pm 15.1}$ | $16.6_{\pm 1.9}$ | $16.3_{\pm 1.8}$ | $4.7_{\pm 0.5}$ | $\mathbf{3.6_{\pm 0.0}}$ |

Table 1: Power(%) of a kernel two-sample test on 1D datasets. Lower is better.

white-noise, due to their limited adaptability to variable discretizations as discussed in [38]. A trace-class diffusion model using Langevin dynamics [38] records power of $6.9_{\pm 0.5}$ on Quadratic and of $4.0_{\pm 0.3}$ on Melbourne, which is comparable to SP-SGM but are less favorable than HDM-SDE.

## 4.2 Motion Generation through HDM-SDE

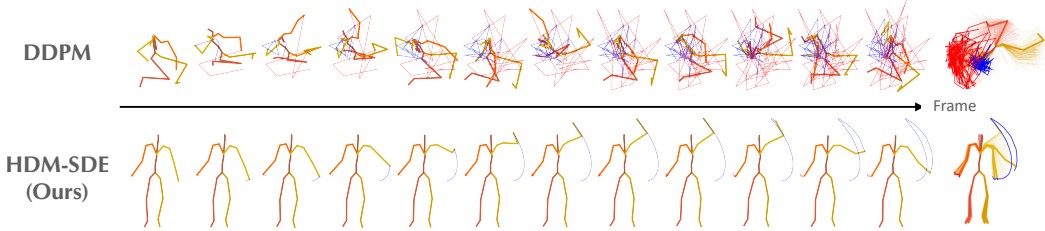

Figure 4: Generated motion by the baseline (DDPM) [25] and HDM-SDE (ours).

**Datasets and Implementation** We further investigate the performance of HDM-SDE for motion generation tasks using a HumanML3D dataset [22]. The HumanML3D dataset consists of 14,616 motions, where each motion consists of 144-dimensional trajectories, derived from 24 joints and a 6D representation and we randomly sample 100 motions for the experiment. The root poses of the skeletons are fixed to the origin, and the motion length is trimmed to 128. We compare our method with the baseline (DDPM) [25] utilizing the FNO architecture except the input and output channel sizes of FNO are changed from 1 to 144 to handle 144-dimensional trajectories. Hence, our proposed method and the baseline have identical network architectures and loss functions, and the only difference is the noise sampling procedures in forward and reverse processes. We utilize the squared-exponential kernel for generating the Wiener process in RKHS where the length parameters of each 144-dimensional trajectory are selected using a likelihood-based model selection of a Gaussian process [55]. See Appendix D.3 for details.

**Results** The synthesized motions of the proposed method are shown in Figure 4, where we overlay right and left hands trajectories with red and blue, respectively. While the proposed method successfully generates human motions, the baseline fails to generate meaningful motions. Our proposed method significantly outperforms the baseline in terms of motion synthesis as the baseline fails to generate human-like motions. This clearly highlights the advantage of utilizing $\mathcal{H}$-valued Wiener process in modeling multi-dimensional complex trajectories. The quantitative results demonstrate better performance in both hand and foot trajectories within the task space are shown in Table A.2.

## 4.3 Image Generation through HDM-SPDE

**Datasets** 2D image experiment includes MNIST [36], CIFAR10 [42], LSUN-church [58], AFHQ [13], and FFHQ [28] datasets. We adjust the image resolutions as IHDM [50], LSUN-church images are resized into 128×128; higher resolutions, including AFHQ and FFHQ, have resized into 256×256. See Appendix D.4 for implementation details.

**Evaluation** To evaluate the sampling quality, we use the clean FID [47] using InceptionV3 [53]. Clean-FID scales generated and reference images into 299×299 resolution via bicubic interpolation and compute statistics. Following IHDM [50], we sample 50,000 images with a resolution smaller than 256×256 and calculate the reference samples as the training set. Otherwise, we sample 10,000 images and use the entire dataset as reference data for evaluating metrics.

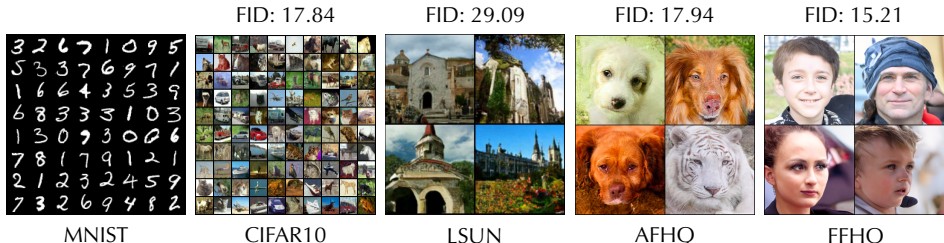

FID: 17.84    FID: 29.09    FID: 17.94    FID: 15.21

MNIST    CIFAR10    LSUN    AFHQ    FFHQ

Figure 5: Uncurated generated samples for different image datasets via HDM-SPDE.

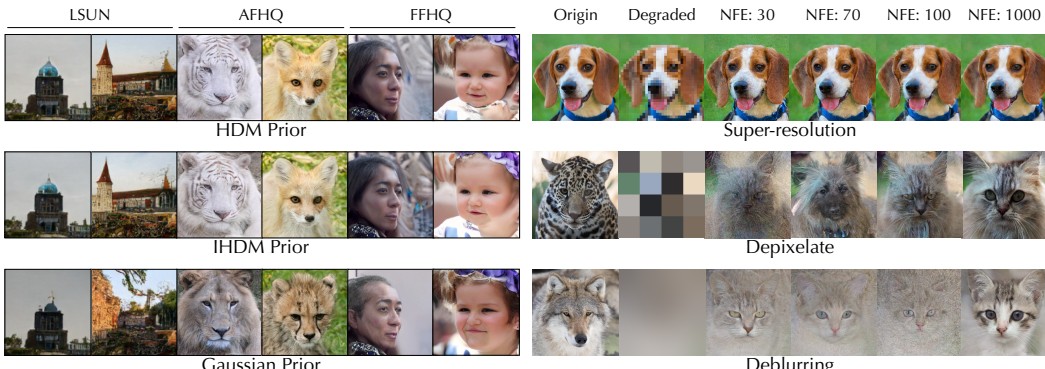

Figure 6: Sampling from HDM-SPDE prior, IHDM prior, and Gaussian prior $\delta = 0.01$ using trained HDM-SPDE.

Figure 7: Sample trajectory of HDM-SPDE on super-resolution, depixelate, and deblurring tasks.

**Results**  Figure 5 summarizes uncurated image generation results of HDM-SPDE on various image datasets. Our model records FID scores of 17.84 on CIFAR10 and 29.09 on LSUN-church, which is better than that of IHDM [50] (CIFAR: 18.96, LSUN: 45.06). Especially, our model records PRDC [45] of (P: 0.696, R: 0.633, D: 0.733, C: 0.727) on CIFAR10, which is higher than that of IHDM (P: 0.596, R: 0.767, D: 0.580, C: 0.723) except for recall. For AFHQ and FFHQ, HDM-SPDE achieves at least twice the better FID score of 17.94 on AFHQ and 15.21 on FFHQ than those of IHDM (AFHQ: 41.39, FFHQ: 64.91). Although there remain gaps compared to the state-of-the-art results using diffusion models [25, 52], the improvement in sample quality as a continuous-time version of the IHDM is significant. Thus, we conclude that the proposed framework opens the potential for exploring various evolution operators having an inductive bias in image generation.

**Connection between HDM and IHDM**  As we addressed in Section 2.3.2, SPDE and IHDM have a connection such that the mean trajectory of the SPDE is equivalent to the trajectory of the forward process of IHDM. Hence we hypothesize that the HDM can generate images by running the reverse SPDE from the prior distribution of IHDM. Figure 6 illustrates the generated images by trained HDM-SPDE provided with HDM, IHDM, and Gaussian priors. Notably, the reverse SPDE generates similar images from IHDM prior to HDM one, contrary to Gaussian prior. On AFHQ and FFHQ, HDM prior results in FID of 17.94 and 19.40, better than 19.40 and 20.89 of IHDM prior, still outperform the results reported by [50], 43.39 and 64.91, respectively.

**Controllable Coarse-to-Fine Generation**  Motivated from [4], we examine whether HDM-SPDE can leverage the learned inductive bias in controllable coarse-to-fine image generation tasks. Figures 7 and A.4 show that HDM-SPDE can successfully perform inpainting, super-resolution, depixelation, and deblurring tasks without any additional training. HDM-SPDE is capable of generating images that resemble the background color to some extent for pixelated images and also generates samples leveraging the contours of blurred images. This is possible since low frequencies are filled with the frequencies of low-resolution images, whereas the range of high frequencies is filled with generated frequencies through the learned score model. Diffusion models trained by [52] fail to generate samples that capture the color or contours of pixelated or blurred images (see Appendix D.5).

# 5    Related Works

Diffusion models in infinite-dimensional space have been actively investigated [18, 23, 29, 38, 48, 49]. [29] introduces a discrete-time denoising diffusion model in infinite-dimensional space, which provided one of the key inspirations for studying a continuous-time approach connecting with score-based methods through the SDE framework [52]. [18] is another discrete-time approach proposed to generalize neural processes [20], but it utilizes non-trace class white noise that causes an ill-posedness in the infinite-dimensional setting. See [14] for a reference of Hilbert space valued Wiener processes.

A challenging part of the continuous-time approach in Hilbert space is defining the score function without using the density function that no longer exists in the infinite-dimensional setting. [23] proposes using a discretized process to circumvent this problem and [48] suggests a continuous-time model based on spectral decomposition stemming from the Karhunen-Loève theorem; however, they do not include the general SDE framework [52] nor time-dependent operators. Based on [44], [19] only considers constant-time diffusion coefficients, which do not fully recover the desired framework requiring variance scheduling [46]. Section 2.3.1 shows the intersection between [19, 23] and our approach and how the proposed framework covers SDEs in infinite dimensions.

Concurrently, [38, 49] propose continuous-time models; however, their theoretical results are also confined to SDEs with constant-time operators since their approach is primarily grounded in semi-group theory [14], which has an intrinsic limitation when dealing with variable coefficients [39]. We note that the multiple noise scale case is revealed in [38, Section 4.4], yet it is constrained to discrete-time models. On the contrary, our approach is grounded in the study of variational approach to stochastic evolution equations [34, 35] and forward and backward Kolmogorov equations with functional derivatives [6, 15], which serves as a more established approach to identifying relations (Theorem 2.1) between time-dependent evolution operators and invariant measures of stochastic equations in Hilbert spaces [5]. It is noteworthy that proof of Theorem 2.1 (Appendix B.2) can be viewed as a generalization of the classical techniques employed by [1] and [24] in finite-dimensional spaces to general Hilbert spaces. Consequently it is able to fully recover the SDE framework proposed from [52], thereby enabling the variance scheduling [46], and building a bridge between the discrete-time model presented in IHDM [50] and the continuous-time score-based approach.

Similar to [11, 23, 29, 38], which combine diffusion models with neural operators [32, 37], we propose a modified *time-conditioned* FNO for the HDM-SDE in the functional and motion generation task, which has slightly different modules from the original neural operator. For the image generation task with the HDM-SPDE, however, we use the U-Net [51] to compare with the setting in variations of diffusion models [4, 27, 50], which utilize image transformations instead of drift coefficients.

# 6    Limitation and Conclusion

We extend the SDE framework proposed by [52] and propose a class of continuous-time score-based generative models based on stochastic evolution equations in Hilbert spaces. Our derivation of the time-reversal formula considering time-dependent coefficients enlarges the practical application of score-based generative models in function spaces and advances the performance of recent variations of diffusion models which use blurring [50] or pixelation [4]. Although performance gaps remain in image generation compared to the state-of-the-art results, we conclude that the proposed framework opens the potential for exploring various evolution operators and sample spaces in this area.

## Acknowledgments and Disclosure of Funding

This work was supported by LG AI Research. This work was also supported by Institute of Information & communications Technology Planning & Evaluation(IITP) grant funded by the Korea government(MSIT)(No. 2022-0-00612, Geometric and Physical Commonsense Reasoning based Behavior Intelligence for Embodied AI; No. 2019-0-00079, Artificial Intelligence Graduate School Program, Korea University), and National Research Foundation of Korea(NRF) funded by the Korea government(MSIT)(2021R1A4A3033149). This work was supported by Artificial intelligence industrial convergence cluster development project funded by the Ministry of Science and ICT(MSIT, Korea) & Gwangju Metropolitan City.

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
