# A Notation

- $\mathbb{N}$: the set of natural numbers
- $\mathbb{R}^d$: $d$-dimensional Euclidean space
- $\mathbb{R}_+$: the set of nonnegative real numbers ($= [0, \infty)$)
- $\mathcal{H}$: a Hilbert space
- $\mathcal{B}(\mathcal{H})$: Borel $\sigma$-field on $\mathcal{H}$
- $\mathcal{L}(\mathcal{H})$: the class of bounded linear operators on $\mathcal{H}$
- $\mathcal{L}_2(\mathcal{H})$: the class of Hilbert-Schmidt operators on $\mathcal{H}$
- $\mathcal{H}^*$: the dual space of $\mathcal{H}$
- $\Lambda \in \mathcal{L}(\mathcal{H})$: a nonnegative, symmetric, self-adjoint, and the finite trace covariance operator
- $\lambda$: Radon centered Gaussian measure on $\mathcal{H}$ with $\Lambda$
- $\mathcal{H}_\lambda$: Cameron-Martin space of the measure $\lambda$
- $\mathcal{M}(\mathcal{H})$: Banach space of all $\mathcal{H}$-valued continuous, square-integrable martingale processes
- $\mathcal{L}_\lambda(\mathcal{H})$: the class of Hilbert-Schmidt operators from $\mathcal{H}_\lambda$ into $\mathcal{H}$
- $(\mathbf{W}_t)_{t \in [0,T]}$: $\mathcal{H}$-valued $\Lambda$-Wiener process

# B A Foundation on Time-Reversal of Stochastic Evolution Equations

## B.1 Preliminaries

**Gelfand Triple and Cameron-Martin Space** Let $\mathcal{H}$ be a separable Hilbert space with the $\mathcal{H}$-norm defined by the inner product $\|\cdot\|_\mathcal{H} = \sqrt{\langle \cdot, \cdot \rangle_\mathcal{H}}$. Let $\mathcal{L}(\mathcal{H})$ denotes the set of all bounded linear operators on $\mathcal{H}$. We assume there exists a Radon centered Gaussian measure $\lambda$ on the Borel $\sigma$-field $\mathcal{B}(\mathcal{H})$ with the nonnegative, symmetric, self-adjoint, and the finite trace covariance operator $\Lambda \in \mathcal{L}(\mathcal{H})$, hence there exists $\{(\lambda^{(\ell)}, \phi^{(\ell)}) \in \mathbb{R}_+ \times \mathcal{H} : \ell \in \mathbb{N}\}$ such that $\Lambda(\phi^{(\ell)}) = \lambda^{(\ell)} \phi^{(\ell)}$ holds and $\mathrm{Tr}(\Lambda) = \sum_{\ell=1}^\infty \lambda^{(\ell)} < \infty$. We define $\mathcal{H}_\lambda := \Lambda^{1/2}(\mathcal{H})$ by

$$\mathcal{H}_\lambda = \{h \in \mathcal{H} : \|h\|_{\mathcal{H}_\lambda} < \infty\}, \quad \langle h_1, h_2 \rangle_{\mathcal{H}_\lambda} := \langle \Lambda^{-1/2}(h_1), \Lambda^{-1/2}(h_2) \rangle_\mathcal{H}. \tag{30}$$

$\mathcal{H}_\lambda$ is called *Cameron-Martin space*. Note that $\mathcal{H}_\lambda$ continuously embedded into $\mathcal{H}$. Let $\mathcal{H}^* \subset \mathcal{H}_\lambda^*$ be the dual space of $\mathcal{H}$ and $\mathcal{H}_\lambda$, respectively. By identifying $\mathcal{H}_\lambda$ with $\mathcal{H}_\lambda^*$ via the Riesz isomorphism, we can consider a continuous inclusion map $i_{\mathcal{H}_\lambda} : \mathcal{H}^* \to \mathcal{H}_\lambda$ such that

$$\varphi(\xi) = \langle i_{\mathcal{H}_\lambda}(\varphi), \xi \rangle_{\mathcal{H}_\lambda} = \langle \Lambda^{-1/2}(i_{\mathcal{H}_\lambda}(\varphi)), \Lambda^{-1/2}(\xi) \rangle_\mathcal{H}, \quad \varphi \in \mathcal{H}^*, \xi \in \mathcal{H}_\lambda. \tag{31}$$

By the Riesz map $\mathcal{R}_\mathcal{H} : \mathcal{H}^* \to \mathcal{H}$, there exists unique $\mathcal{R}_\mathcal{H}(\varphi) \in \mathcal{H}$ such that $\varphi(\xi) = \langle \mathcal{R}_\mathcal{H}(\varphi), \xi \rangle_\mathcal{H}$. Since $\Lambda$ is self-adjoint, we get $i_{\mathcal{H}_\lambda}(\varphi) = \Lambda(\mathcal{R}_\mathcal{H}(\varphi))$ via (31) so that $i_{\mathcal{H}_\lambda}(\mathcal{H}^*) = \Lambda(\mathcal{H})$. Thus, we can identify $\mathcal{H}^*$ with $\Lambda^{1/2}(\mathcal{H}_\lambda)$ (see [8, Example 1.1]). For notational convenience, we write $i_{\mathcal{H}_\lambda}(\varphi)$ as $\Lambda(\varphi)$ instead of $\Lambda(\mathcal{R}_\mathcal{H}(\varphi))$ by regarding $\varphi$ as an element in $\mathcal{H}$. In summary,

$$\mathcal{H}_\lambda = \Lambda^{1/2}(\mathcal{H}), \quad i_{\mathcal{H}_\lambda}(\mathcal{H}^*) = \Lambda(\mathcal{H}), \quad i_{\mathcal{H}_\lambda}(\mathcal{H}^*) \subset \mathcal{H}_\lambda \subset \mathcal{H}. \tag{32}$$

We refer to $(\mathcal{H}^*, \mathcal{H}_\lambda, \mathcal{H})$ as *Gelfand triple* which provides groundwork for the SDE theory in $\mathcal{H}$.

**Stochastic Integration in Hilbert Spaces** Let $(\Omega, \mathcal{F}, (\mathcal{F}_t)_{t \in [0,T]}, \mathbb{P})$ be a filtered probability space.

**Definition B.1** ($\mathcal{H}$-valued $\Lambda$-Wiener process[3])**.** A $\mathcal{H}$-valued stochastic process $(\mathbf{W}_t, t \in [0, T])$ is called $\Lambda$-Wiener process if

1. $\mathbf{W}_0 = 0$ (a.s.),

2. $\mathbf{W}$ has continuous trajectories $[0, T] \to \mathcal{H}$ for each $\omega \in \Omega$,

---

[3]In general, one can use different Hilbert space $\mathcal{H}_0 \neq \mathcal{H}$ to define $\Lambda_0$-Wiener process $\mathbf{W}_t \in \mathcal{H}_0$ in (7) with covariance operator $\Lambda_0 \in \mathcal{L}(\mathcal{H}_0)$ and $\mathbf{G} \in \mathcal{L}(\mathcal{H}_0, \mathcal{H}_\lambda)$. In this paper, we focus on $\mathcal{H}_0 = \mathcal{H}_1$ and $\Lambda_0 = \Lambda$ case.

3. $\mathbf{W}_t$ is $\mathcal{F}_t$-adapted and has independent increments such that $\mathbf{W}_t - \mathbf{W}_s$ is independent of $\mathcal{F}_s$ and $\mathbf{W}_t - \mathbf{W}_s \sim \mathcal{N}(0, (t-s)\Lambda)$ for $0 \leq s \leq t \leq T$.

By Definition B.1, for any $u, v \in \mathcal{H}$ and $t \in [0, T]$, it holds that

$$\langle \mathbf{W}_t, u \rangle_{\mathcal{H}} \sim \mathcal{N}(0, t\langle \Lambda(u), u \rangle_{\mathcal{H}}), \quad \mathbb{E}\left[\langle \mathbf{W}_t, u \rangle_{\mathcal{H}} \langle \mathbf{W}_t, v \rangle_{\mathcal{H}}\right] = t\langle \Lambda(u), v \rangle_{\mathcal{H}}. \tag{33}$$

Thus, $\mathrm{Cov}(\mathbf{W}_t) = t\Lambda$ holds. Recall $\Lambda(\phi^{(\ell)}) = \lambda^{(\ell)}\phi^{(\ell)}$ for $\ell \in \mathbb{N}$. Then,

$$\mathbb{E}\left[\langle \mathbf{W}_t, \phi^{(\ell)} \rangle_{\mathcal{H}} \langle \mathbf{W}_s, \phi^{(\ell')} \rangle_{\mathcal{H}}\right] \tag{34}$$

$$= \mathbb{E}\left[\langle \mathbf{W}_t - \mathbf{W}_s, \phi^{(\ell)} \rangle_{\mathcal{H}} \langle \mathbf{W}_s, \phi^{(\ell')} \rangle_{\mathcal{H}}\right] + \mathbb{E}\left[\langle \mathbf{W}_s, \phi^{(\ell)} \rangle_{\mathcal{H}} \langle \mathbf{W}_s, \phi^{(\ell')} \rangle_{\mathcal{H}}\right] \tag{35}$$

$$= s\langle \Lambda(\phi^{(\ell)}), \phi^{(\ell')} \rangle_{\mathcal{H}} = s\langle \Lambda^{1/2}(\phi^{(\ell)}), \Lambda^{1/2}(\phi^{(\ell')}) \rangle_{\mathcal{H}} = s\sqrt{\lambda^{(\ell)}\lambda^{(\ell')}}\delta_{\ell\ell'} \tag{36}$$

By the Kosambi-Karhunen-Loève theorem (see [14, Proposition 4.3]), $\mathbf{W}_t$ has a series representation:

$$\mathbf{W}_t = \Lambda^{1/2}(\mathcal{W}_t) := \sum_{\ell=1}^{\infty} \sqrt{\lambda^{(\ell)}} W_t^{(\ell)} \phi^{(\ell)}, \quad t \in [0, T], \tag{37}$$

where $\mathcal{W}_t = \{W_t^{(\ell)} : \ell \in \mathbb{N}\}$ is *white noise* of which the components $W_t^{(\ell)}$ are independent real-valued Wiener processes. Note that $\mathcal{W}_t \notin \mathcal{H}$ in general; hence $\Lambda^{1/2}(\mathcal{W}_t)$ is an informal use of the notation, but the series (37) is valid and almost surely uniformly convergent on $[0, T]$. Furthermore,

$$\mathbb{E}\left[\langle \mathbf{W}_t, \mathbf{W}_s \rangle_{\mathcal{H}}\right] = \sum_{\ell,\ell'} \sqrt{\lambda^{(\ell)}\lambda^{(\ell')}} \mathbb{E}\left[W_t^{(\ell)} W_s^{(\ell')}\right] \langle \phi^{(\ell)}, \phi^{(\ell')} \rangle_{\mathcal{H}} \tag{38}$$

$$= \sum_{\ell,\ell'} \sqrt{\lambda^{(\ell)}\lambda^{(\ell')}} \min\{t, s\}\delta_{\ell,\ell'} \tag{39}$$

$$= \min\{t, s\} \sum_{\ell \in \mathbb{N}} \lambda^{(\ell)} = \min\{t, s\} \cdot \mathrm{Tr}(\Lambda) < \infty, \tag{40}$$

since $\Lambda$ has the finite trace. Now we introduce how to construct the stochastic integration in $\mathcal{H}$ with respect to $\Lambda$-Wiener process $\mathbf{W}_t$. A complete investigation of the theory of stochastic integration is beyond the purpose of the paper, we discuss only necessary components which we use in the sequel. Let $\mathcal{M}(\mathcal{H})$ be the space of all $\mathcal{H}$-valued continuous, square-integrable martingale processes on $t \in [0, T]$. Then $\mathcal{M}(\mathcal{H})$ is a Banach space equipped with the norm

$$\|\mathbf{X}\|_{\mathcal{M}(\mathcal{H})} := \sup_{t \in [0,T]} \left(\mathbb{E}\|\mathbf{X}_t\|_{\mathcal{H}}^2\right)^{1/2}. \tag{41}$$

Note that $\mathbf{W} \in \mathcal{M}(\mathcal{H})$. Indeed,

$$\|\mathbf{W}\|_{\mathcal{M}(\mathcal{H})} = \sup_{t \in [0,T]} \left(\mathbb{E}\|\mathbf{W}_t\|_{\mathcal{H}}^2\right)^{1/2} = \sup_{t \in [0,T]} (t \cdot \mathrm{Tr}(\Lambda))^{1/2} = \sqrt{T \cdot \mathrm{Tr}(\Lambda)} < \infty. \tag{42}$$

Let $\mathcal{E}$ be the class of $\mathcal{L}(\mathcal{H})$-valued elementary processes satisfying usual conditions (see [14, Section 4.2]). Then we can define a linear map $\mathfrak{Int} : \mathcal{E} \to \mathcal{M}(\mathcal{H})$ such that for any $\mathbf{G}_0 \in \mathcal{E}$, $\mathfrak{Int}(\mathbf{G}_0)$ induces a natural stochastic integration with the following Itō-isometry property

$$\mathfrak{Int}(\mathbf{G}_0)(t) = \int_0^t \mathbf{G}_0(s)\mathrm{d}\mathbf{W}_s, \quad \|\mathfrak{Int}(\mathbf{G}_0)\|_{\mathcal{M}(\mathcal{H})} = \left(\mathbb{E}\left[\int_0^T \|\mathbf{G}_0(s) \circ \Lambda^{1/2}\|_{\mathcal{L}_2(\mathcal{H})}^2 \mathrm{d}s\right]\right)^{1/2}. \tag{43}$$

Therefore, we can extend the domain of $\mathfrak{Int}$ from the class $\mathcal{E}$ to $\mathcal{L}_\lambda(\mathcal{H}) := \mathcal{L}_2(\mathcal{H}_\lambda, \mathcal{H})$-valued predictable processes satisfying the following condition (see [14, Proposition 4.22]):

$$\left(\mathbb{E}\left[\int_0^T \|\mathbf{G}_s \circ \Lambda^{1/2}\|_{\mathcal{L}_\lambda(\mathcal{H})}^2\right]\right)^{1/2} < \infty. \tag{44}$$

Thus, the stochastic integration in $\mathcal{H}$ for $\mathbf{G}_t : [0, T] \to \mathcal{L}_\lambda(\mathcal{H})$ is well-defined.

## B.2 Proof of Theorem 2.1

This section is devoted to the derivation of the time-reversal formula (16) in Theorem 2.1, which identifies the evolution operators $\widehat{\mathbf{B}}$ and $\widehat{\mathbf{G}}$ of (8). A condition for the existence of the time-reversal process (8) is well-studied for diffusion processes in abstract settings (e.g., see [12]); however, the existence of a time-reversal process does not imply the exact time-reversal formula, and vice versa [10, 57]. This topic is beyond the scope of this paper; hence we focus on the proof of (16). Note that we derive (16) when $\mathbf{G}_t$ is constant with respect to $u \in \mathcal{H}$. It is possible to get a more generalized formula through the same approach assuming non-constant operator $\mathbf{G}_t$ (e.g. see [6, 43]). However, we omit the case since Theorem 2.1 sufficiently recovers SDEs used in [52].

To show (16), we need to define a Kolmogorov operator for (8) by

$$\widehat{\mathscr{L}_\tau} f_{\varphi_{1:m}}(u) := \frac{1}{2} \mathrm{Tr}_{\mathcal{H}_\lambda} (\widehat{\mathbf{A}}_t \circ \Lambda \circ D^2 f_{\varphi_{1:m}}(u)) + \left( D f_{\varphi_{1:m}}(u), \widehat{\mathbf{B}}_t(u) \right) \tag{45}$$

$$= \frac{1}{2} \sum_{i,j} \partial_{ij}^2 f(\varphi_1(u), \dots, \varphi_m(u)) \left\langle \widehat{\mathbf{A}}_t \Lambda(\varphi_i), \Lambda(\varphi_j) \right\rangle_{\mathcal{H}_\lambda} \tag{46}$$

$$+ \sum_i \partial_i f(\varphi_1(u), \dots, \varphi_m(u)) \langle \widehat{\mathbf{B}}_t(u), \Lambda(\varphi_i) \rangle_{\mathcal{H}_\lambda} \tag{47}$$

where $\widehat{\mathbf{A}}_t := \widehat{\mathbf{G}}_t \widehat{\mathbf{G}}_t^*$. We modify the proof of [24, Theorem 2.1] which shows the desired result for the finite-dimensional case. It is well-known that $\widehat{\mathbf{X}}_t$ is already a Markov diffusion process; hence we only need to show that $\widehat{L}_t$ is its generator. Recall that $\mathcal{FC}_b^\infty$ is dense in the Sobolev space on $\mathcal{H}$. Fix $\varphi_i \in \mathcal{H}^*$ for $1 \le i \le m$ where $m \in \mathbb{N}$. For notational convenience, we omit $\varphi_{1:m}$ and $\varphi_1(u), \dots, \varphi_m(u)$ without confusion. For $0 \le s \le t \le T$ and $f \in \mathcal{FC}_b^\infty$, we will show

$$\mathbb{E}\left[ f(\widehat{\mathbf{X}}_t) - f(\widehat{\mathbf{X}}_s) - \int_s^t \widehat{\mathscr{L}_\tau} f(\widehat{\mathbf{X}}_\tau) d\tau \, \Big| \widehat{\mathbf{X}}_r : 0 \le r \le s \right] = 0 \tag{48}$$

or, equivalently, for any $g \in \mathcal{FC}_b^\infty$,

$$\mathbb{E}\left[ \left( f(\widehat{\mathbf{X}}_t) - f(\widehat{\mathbf{X}}_s) - \int_s^t \widehat{\mathscr{L}_\tau} f(\widehat{\mathbf{X}}_\tau) d\tau \right) g(\widehat{\mathbf{X}}_s) \right] = 0. \tag{49}$$

By the change of variable and $T - s \to t$ and $T - t \to s$ (so $0 \le s \le t \le T$), this is equivalent to

$$\mathbb{E}\left[ \left( f(\mathbf{X}_t) - f(\mathbf{X}_s) - \int_s^t \widetilde{\mathscr{L}_\tau} f(\mathbf{X}_\tau) d\tau \right) g(\mathbf{X}_t) \right] = 0 \tag{50}$$

where

$$\widetilde{\mathscr{L}_\tau} f(x) = -\widehat{\mathscr{L}_{T-\tau}} f(x) \tag{51}$$

$$= -\frac{1}{2} \mathrm{Tr}_{\mathcal{H}_\lambda} (\widehat{\mathbf{A}}_{T-\tau} \circ \Lambda \circ D^2 f(u)) - \left( D f(u), \widehat{\mathbf{B}}_{T-\tau}(u) \right) \tag{52}$$

$$= -\frac{1}{2} \mathrm{Tr}_{\mathcal{H}_\lambda} (\mathbf{A}_\tau \circ \Lambda \circ D^2 f(u)) + \left( D f(u), \mathbf{B}_\tau(u) - \mathbf{A}_\tau \rho_{\mathcal{H}_\lambda}^{\mu_\tau} \right) \tag{53}$$

$$= -\frac{1}{2} \sum_{i,j} \partial_{ij}^2 f \langle \mathbf{A}_\tau \circ \Lambda(\varphi_i), \Lambda(\varphi_j) \rangle_{\mathcal{H}_\lambda} + \sum_i \partial_i f \langle \mathbf{B}_\tau - \mathbf{A}_\tau \rho_{\mathcal{H}_\lambda}^{\mu_\tau}, \Lambda(\varphi_i) \rangle_{\mathcal{H}_\lambda}. \tag{54}$$

Let $\langle f, \mu \rangle = \int_{\mathcal{H}} f(u)\mu(du)$ denote dual pairing between a function and a measure. Then $(\mu_t)_{t \in [0,T]}$ satisfies the Kolmogorov forward equation in weak sense [6, Section 5.3.1],

$$\frac{\partial}{\partial t} \langle f_{\varphi_{1:m}}, \mu_t \rangle = \langle \mathscr{L}_t f_{\varphi_{1:m}}, \mu_t \rangle, \quad t \in [0,T], \ f_{\varphi_{1:m}} \in \mathcal{FC}_b^\infty. \tag{55}$$

For $s < t$, we define an evolution operator $\mathcal{A}_{s,t}$ for $\mathcal{B}(\mathcal{H})$-measurable function as follows:

$$(\mathcal{A}_{s,t} f_{\varphi_{1:m}})(v) = \mathbb{E}[f_{\varphi_{1:m}}(\mathbf{X}_t) | \mathbf{X}_s = v], \quad v \in \mathcal{H} \tag{56}$$

and its adjoint operator $\mathcal{A}_{s,t}^*$ for measure on $\mathcal{H}$ such that $\langle f_{\varphi_{1:m}}, \mathcal{A}_{s,t}^* \mu \rangle = \langle \mathcal{A}_{s,t} f_{\varphi_{1:m}}, \mu \rangle$. By the definition of $(\mu_t)_{t \in [0,T]}$ and the tower property, we have $\mathcal{A}_{s,t}^* \mu_s = \mu_t$ in weak sense:

$$\langle \mathcal{A}_{s,t} f_{\varphi_{1:m}}, \mu_s \rangle = \int_{\mathcal{H}} \mathbb{E}[f_{\varphi_{1:m}}(\mathbf{X}_t) | \mathbf{X}_s = v] \, \mathbb{P}(\mathbf{X}_s \in dv) = \mathbb{E}[f_{\varphi_{1:m}}(\mathbf{X}_t)] = \langle f_{\varphi_{1:m}}, \mu_t \rangle. \tag{57}$$

Then by the Itō formula for Hilbert space valued processes [33, Theorem 3.1], $\zeta_s := \mathcal{A}_{s,t} f_{\varphi_{1:m}}$ satisfies the Kolmogorov backward equation [6, Section 5.1.4],

$$\frac{\partial \zeta_s}{\partial s} + \mathscr{L}_s \zeta_s = 0, \quad s \in (0, t). \tag{58}$$

Recall the Kolmogorov forward equation (55). For $0 \le s \le t \le T$ and $v \in \mathcal{H}$, set

$$\zeta_s(v) := (\mathcal{A}_{s,t} g)(v) = \mathbb{E}[g(\mathbf{X}_t) | \mathbf{X}_s = v] \tag{59}$$

hence

$$\mathbb{E}[f(\mathbf{X}_s)g(\mathbf{X}_t)] = \mathbb{E}[f(\mathbf{X}_s)\mathbb{E}[g(\mathbf{X}_t)|\mathbf{X}_s]] = \mathbb{E}[f(\mathbf{X}_s)\zeta_s(\mathbf{X}_s)] = \langle f\zeta_s, \mu_s \rangle. \tag{60}$$

By (58), $\zeta_s$ satisfies the Kolmogorov backward equation:

$$\frac{\partial \zeta_s}{\partial s} + \mathscr{L}_s \zeta_s = 0, \quad \zeta_t = g, \quad s \in (0, t). \tag{61}$$

Then by (60), (55), and (61), we have

$$\mathbb{E}[f(\mathbf{X}_s)g(\mathbf{X}_t)] = \langle f\zeta_s, \mu_s \rangle = \langle f\zeta_t, \mu_t \rangle - \int_s^t \frac{\partial}{\partial \tau} \langle f\zeta_\tau, \mu_\tau \rangle d\tau \tag{62}$$

$$= \mathbb{E}[f(\mathbf{X}_t)g(\mathbf{X}_t)] - \int_s^t \left( \langle \mathscr{L}_\tau(f\zeta_\tau), \mu_\tau \rangle + \langle f \frac{\partial}{\partial \tau} \zeta_\tau, \mu_\tau \rangle \right) d\tau \tag{63}$$

$$= \mathbb{E}[f(\mathbf{X}_t)g(\mathbf{X}_t)] - \int_s^t \int_{\mathcal{H}} (\mathscr{L}_\tau(f\zeta_\tau) - f\mathscr{L}_\tau \zeta_\tau) \, \mu_\tau(du) d\tau \tag{64}$$

To compute the integrand, observe that

$$\mathscr{L}_\tau(f\zeta_\tau) - f\mathscr{L}_\tau \zeta_\tau = \frac{1}{2}\mathrm{Tr}_{\mathcal{H}_\lambda} \left( \widehat{\mathbf{A}}_\tau \circ \Lambda \circ D^2(f\zeta_\tau)(u) \right) + \left( D(f\zeta_\tau)(u), \widehat{\mathbf{B}}_t(u) \right) \tag{65}$$

$$- f \left[ \frac{1}{2}\mathrm{Tr}_{\mathcal{H}_\lambda} \left( \widehat{\mathbf{A}}_\tau \circ \Lambda \circ D^2\zeta_\tau(u) \right) + \left( D\zeta_\tau(u), \widehat{\mathbf{B}}_t(u) \right) \right] \tag{66}$$

$$= \zeta_\tau \mathscr{L}_\tau f + \sum_{i,j} \partial_i \zeta_\tau \partial_j f \left\langle \widehat{\mathbf{A}}_\tau \circ \Lambda(\varphi_i), \circ\Lambda(\varphi_j) \right\rangle_{\mathcal{H}_\lambda} \tag{67}$$

Note that (11) can be restated as

$$\int_{\mathcal{H}} \langle Df_{\varphi_{1:m}}(u), \xi \rangle_{\mathcal{H}_\lambda} \mu(\mathrm{d}u) = - \int_{\mathcal{H}} f_{\varphi_{1:m}}(u) \langle \rho_{\mathcal{H}_\lambda}^\mu(u), \xi \rangle_{\mathcal{H}_\lambda} \mu(\mathrm{d}u). \tag{68}$$

Due to the integration by parts formula (68), we get

$$\int_{\mathcal{H}} \sum_{i,j} \partial_i \zeta_\tau \left\langle \partial_j f \mathbf{A}_\tau \circ \Lambda(\varphi_i), \Lambda(\varphi_j) \right\rangle_{\mathcal{H}_\lambda} \mu_\tau(du) \tag{69}$$

$$= \sum_j \int_{\mathcal{H}} (D\zeta_\tau(u), \partial_j f \mathbf{A}_\tau \circ \Lambda(\varphi_j)) \, \mu_\tau(du) \tag{70}$$

$$= - \sum_j \int_{\mathcal{H}} \zeta_\tau(u) \langle \rho_{\mathcal{H}_\lambda}^{\mu_t}(u), \partial_j f \mathbf{A}_\tau \circ \Lambda(\varphi_j) \rangle_{\mathcal{H}_\lambda} \mu_\tau(du) \tag{71}$$

$$- \sum_{i,j} \int_{\mathcal{H}} \zeta_\tau(u) \partial_{ij}^2 f \left\langle \mathbf{A}_\tau \circ \Lambda(\varphi_i), \Lambda(\varphi_j) \right\rangle_{\mathcal{H}_\lambda} \mu_\tau(du). \tag{72}$$

Observe that

$$\sum_j \int_{\mathcal{H}} \zeta_\tau(u) \langle \rho_{\mathcal{H}_\lambda}^{\mu_t}(u), \partial_j f \mathbf{A}_\tau \circ \Lambda(\varphi_j) \rangle_{\mathcal{H}_\lambda} \mu_\tau(du) \tag{73}$$

$$= \int_{\mathcal{H}} \zeta_\tau(u) \sum_j \partial_j f \langle \mathbf{A}_\tau \rho_{\mathcal{H}_\lambda}^{\mu_t}(u), \Lambda(\varphi_j) \rangle_{\mathcal{H}_\lambda} \mu_\tau(du) \tag{74}$$

$$= \int_{\mathcal{H}} \zeta_\tau(u) \left( Df(u), \mathbf{A}_\tau \rho_{\mathcal{H}_\lambda}^{\mu_t} \right) \mu_\tau(du) \tag{75}$$

and

$$\int_{\mathcal{H}} \zeta_\tau(u)\partial_{ij}^2 f \langle \mathbf{A}_\tau \circ \Lambda(\varphi_i), \Lambda(\varphi_j)\rangle_{\mathcal{H}_\lambda} \mu_\tau(du) \tag{76}$$

$$= \int_{\mathcal{H}} \zeta_\tau(u) \sum_{i,j} \partial_{ij}^2 f \langle \mathbf{A}_\tau \circ \Lambda(\varphi_i), \Lambda(\varphi_j)\rangle_{\mathcal{H}_\lambda} \mu_\tau(du) \tag{77}$$

$$= \int_{\mathcal{H}} \zeta_\tau(u)\mathrm{Tr}_{\mathcal{H}_\lambda}\left(\mathbf{A}_\tau \circ \Lambda \circ D^2 f(u)\right) \mu_\tau(du). \tag{78}$$

Therefore, combining the above results, we have

$$\sum_{i,j} \int_{\mathcal{H}} \partial_i\zeta_\tau \partial_j f \langle \mathbf{A}_\tau \circ \Lambda(\varphi_i), \Lambda(\varphi_j)\rangle_{\mathcal{H}_\lambda} \mu_\tau(du) \tag{79}$$

$$= -\int_{\mathcal{H}} \zeta_\tau(u)\left[\mathrm{Tr}_{\mathcal{H}_\lambda}\left(\mathbf{A}_\tau \circ \Lambda \circ D^2 f(u)\right) + \left(Df(u), \mathbf{A}_\tau\rho_{\mathcal{H}_\lambda}^{\mu_\tau}\right)\right] \mu_\tau(du). \tag{80}$$

Hence

$$\int_s^t \int_{\mathcal{H}} \left(\mathscr{L}_\tau(f\zeta_\tau) - f\mathscr{L}_\tau\zeta_\tau\right) \mu_\tau(du)d\tau \tag{81}$$

$$= \int_s^t \int_{\mathcal{H}} \left[\mathscr{L}_\tau f(u) - \mathrm{Tr}_{\mathcal{H}_\lambda}\left(\mathbf{A}_\tau \circ \Lambda \circ D^2 f(u)\right) - \left(Df(u), \mathbf{A}_\tau\rho_{\mathcal{H}_\lambda}^{\mu_\tau}\right)\right] \zeta_\tau(u)\mu_\tau(du)d\tau \tag{82}$$

$$= \int_s^t \int_{\mathcal{H}} \left[-\frac{1}{2}\mathrm{Tr}_{\mathcal{H}_\lambda}\left(\mathbf{A}_\tau \circ \Lambda \circ D^2 f(u)\right) + \left(Df(u), \mathbf{B}_\tau(u) - \mathbf{A}_\tau\rho_{\mathcal{H}_\lambda}^{\mu_\tau}\right)\right] \zeta_\tau\mu_\tau(du)d\tau \tag{83}$$

$$= \int_s^t \int_{\mathcal{H}} \widetilde{\mathscr{L}_\tau} f(u)\zeta_\tau(u)\mu_\tau(du)d\tau = \int_s^t \mathbb{E}\left[\widetilde{\mathscr{L}_\tau} f(\mathbf{X}_\tau)\zeta_\tau(\mathbf{X}_\tau)\right] d\tau. \tag{84}$$

Therefore,

$$\mathbb{E}[f(\mathbf{X}_s)g(\mathbf{X}_t)] = \mathbb{E}[f(\mathbf{X}_t)g(\mathbf{X}_t)] - \int_s^t \mathbb{E}\left[\widetilde{\mathscr{L}_\tau} f(\mathbf{X}_\tau)\zeta_\tau(\mathbf{X}_\tau)\right] d\tau \tag{85}$$

$$= \mathbb{E}[f(\mathbf{X}_t)g(\mathbf{X}_t)] - \mathbb{E}\left[\int_s^t \widetilde{\mathscr{L}_\tau} f(\mathbf{X}_\tau)g(\mathbf{X}_t)d\tau\right] \tag{86}$$

$$= \mathbb{E}\left[\left(f(\mathbf{X}_t) - \int_s^t \widetilde{\mathscr{L}_\tau} f(\mathbf{X}_\tau)d\tau\right) g(\mathbf{X}_t)\right]. \tag{87}$$

Thus, we prove (50) and obtain the desired result. The theorem is proved.

## C  Hilbert Diffusion Models

### C.1  Derivation of (19) via Theorem 2.1

*Proof.* Note that (18) can be rewritten as an infinite-dimensional stochastic differential system

$$\sum_{\ell\in\mathbb{N}} X_t^{(\ell)}\phi^{(\ell)} = \sum_{\ell\in\mathbb{N}} \left(X_0^{(\ell)} + \int_0^t b_s^{(\ell)} X_s^{(\ell)}ds + \sqrt{\lambda^{(\ell)}} \int_0^t \sigma_s^{(\ell)}dW_s^{(\ell)}\right)\phi^{(\ell)}, \quad t\in[0,T]. \tag{88}$$

By the Itō formula, we can derive a solution of SDE system (88) for each $\ell\in\mathbb{N}$ so that

$$X_t^{(\ell)} = \beta_t^{(\ell)} X_0^{(\ell)} + \beta_t^{(\ell)} \int_0^t \sqrt{\lambda^{(\ell)}\Sigma_\beta^{(\ell)}(s)}dW_s^{(\ell)}, \quad t\in[0,T], \quad \ell\in\mathbb{N}. \tag{89}$$

Then the law of $X_t^{(\ell)}$ conditioned at $X_0^{(\ell)}$ follows Gaussian distribution as follows

$$\mathbb{E}[X_t^{(\ell)}|X_0^{(\ell)}] = \beta_t^{(\ell)} X_0^{(\ell)}, \quad \mathrm{Cov}(X_t^{(\ell)}|X_0^{(\ell)}) = \lambda^{(\ell)}\left(\beta_t^{(\ell)}\right)^2 \int_0^t \Sigma_\beta^{(\ell)}(s)ds. \tag{90}$$

Therefore, we can compute the conditional score function for $X_t^{(\ell)}$ by

$$\partial_{x^{(\ell)}} \log p_t \left( x^{(\ell)} | X_0^{(\ell)} \right) = \frac{\beta_t^{(\ell)} X_0^{(\ell)} - x^{(\ell)}}{\lambda^{(\ell)} \left( \beta_t^{(\ell)} \right)^2 \int_0^t \Sigma_\beta^{(\ell)}(s) \mathrm{d}s}. \tag{91}$$

Thus, we can apply [44, Theorem 5.3] to (18) so we obtain (19) as follows

$$\widehat{\mathbf{B}}_t(\mathbf{x}) = \sum_{\ell \in \mathbb{N}} \left( -b_{T-t}^{(\ell)} x^{(\ell)} + \left( \sqrt{\lambda^{(\ell)}} \sigma_{T-t}^{(\ell)} \right)^2 \partial_{x^{(\ell)}} \log p_{T-t}(x^{(\ell)} | X_0^{(\ell)}) \right) \phi^{(\ell)} \tag{92}$$

$$= \sum_{\ell \in \mathbb{N}} \left( -b_{T-t}^{(\ell)} x^{(\ell)} + \Sigma_\beta^{(\ell)}(T-t) \frac{\beta_{T-t}^{(\ell)} X_0^{(\ell)} - x^{(\ell)}}{\int_0^{T-t} \Sigma_\beta^{(\ell)}(s) \mathrm{d}s} \right) \phi^{(\ell)}. \tag{93}$$

Now we prove the above formula is a special case of Theorem 2.1. Let us set *diagonal* operators

$$\mathbf{B}_t = \mathrm{diag}\left( b_t^{(\ell)} \right)_{\ell \in \mathbb{N}}, \quad \mathbf{G}_t = \mathrm{diag}\left( \sigma_t^{(\ell)} \right)_{\ell \in \mathbb{N}}, \quad \Upsilon_{s,t} = \mathrm{diag}\left( \beta_t^{(\ell)} / \beta_s^{(\ell)} \right)_{\ell \in \mathbb{N}}, \tag{94}$$

or equivalently, for each $u = (u^{(\ell)})_{\ell \in \mathbb{N}} \in \mathcal{H}$,

$$\mathbf{B}_t(u) = \sum_{\ell \in \mathbb{N}} b_t^{(\ell)} u^{(\ell)} \phi^{(\ell)}, \quad \mathbf{G}_t(u) = \sum_{\ell \in \mathbb{N}} \sigma_t^{(\ell)} u^{(\ell)} \phi^{(\ell)}, \quad \Upsilon_{s,t}(u) = \sum_{\ell \in \mathbb{N}} \frac{\beta_t^{(\ell)}}{\beta_s^{(\ell)}} u^{(\ell)} \phi^{(\ell)}, \tag{95}$$

where $\Upsilon_{s,t} \in \mathcal{L}(\mathcal{H})$ and $\mathbf{G}_t \in \mathcal{L}_2(\mathcal{H})$ for each $0 \le s < t \le T$. Since $\mathbf{G}_t$ and $\Upsilon_{s,t}$ are symmetric,

$$\Upsilon_{s,t} \mathbf{G}_s \Lambda \mathbf{G}_s^* \Upsilon_{s,t}^* = \mathrm{diag}\left( \left( \beta_t^{(\ell)} \right)^2 \Sigma_\beta^{(\ell)}(s) \right)_{\ell \in \mathbb{N}} \circ \Lambda. \tag{96}$$

Hence the conditional mean vector and covariance operator is derived by

$$m_{\mu_t|u_0} = \Upsilon_{0,t} u_0 = \sum_{\ell \in \mathbb{N}} \beta_t^{(\ell)} u_0^{(\ell)} \phi^{(\ell)}, \tag{97}$$

$$\Lambda_{\mu_t|u_0} = \int_0^t \Upsilon_{s,t} \mathbf{G}_s \Lambda \mathbf{G}_s^* \Upsilon_{s,t}^* \mathrm{d}s = \mathrm{diag}\left( \left( \beta_t^{(\ell)} \right)^2 \int_0^t \Sigma_\beta^{(\ell)}(s) \mathrm{d}s \right)_{\ell \in \mathbb{N}} \circ \Lambda. \tag{98}$$

Since $\Upsilon_{s,t}$, $\mathbf{G}_s$, and $\Lambda$ are symmetric linear operators, $(\Upsilon_{s,t} \mathbf{G}_s \Lambda^{1/2})^* = \Lambda^{1/2} \mathbf{G}_s \Upsilon_{s,t}$ holds. Then

$$\Lambda_{\mu_t|\mathbf{X}_0}(u) = \sum_{\ell \in \mathbb{N}} \left[ \left( \beta_t^{(\ell)} \right)^2 \lambda^{(\ell)} u^{(\ell)} \int_0^t \Sigma_\beta^{(\ell)}(s) \mathrm{d}s \right] \phi^{(\ell)}, \quad u \in \mathcal{H}. \tag{99}$$

By the assumption, $\Lambda_{\mu_t|\mathbf{X}_0}(\mathcal{H}) = \Lambda(\mathcal{H})$ holds for $t \in [0, T]$ and we can compute $\rho_{\mathcal{H}_\lambda}^{\mu_t|\mathbf{X}_0}$ from (12),

$$\mathbf{S}_\lambda(t, \mathbf{X}_t|\mathbf{X}_0) = \mathbf{A}_t \rho_{\mathcal{H}_\lambda}^{\mu_t|\mathbf{X}_0}(\mathbf{X}_t) = \sum_{\ell \in \mathbb{N}} \left( \Sigma_\beta^{(\ell)}(t) \frac{\beta_t^{(\ell)} X_0^{(\ell)} - X_t^{(\ell)}}{\int_0^t \Sigma_\beta^{(\ell)}(s) \mathrm{d}s} \right) \phi^{(\ell)}. \tag{100}$$

Therefore, the time-reversal formula (16) of Theorem 2.1 and (19) induce the same conclusion.

$\square$

## C.2 Derivation of (23) via Theorem 2.1

*Proof.* Let us consider the transformed stochastic equation (22):

$$\mathrm{d}\hat{u}_t = b(t)(\mathbf{D} - \sigma \mathbf{I})\hat{u}_t \mathrm{d}t + \sqrt{2b(t)}(\sigma \mathbf{I} - \mathbf{D})^{-\frac{k}{2}} \mathrm{d}\mathcal{W}_t, \tag{101}$$

where $\Lambda = \mathbf{I}$ and evolution operators $\mathbf{B}_t = -b(t)(\sigma \mathbf{I} - \mathbf{D})$ and $\mathbf{G}_t = \sqrt{2b(t)}(\sigma \mathbf{I} - \mathbf{D})^{-\frac{k}{2}}$ are matrices acting on $\mathbb{R}^d$. To induce the score operator $\mathbf{S}_\lambda(t, \hat{u}|\hat{u}_0)$ conditioned at $\hat{u}_0$, it is sufficient

to derive the mean vector $m_{\hat{\mu}_t|\hat{u}_0}$ and the covariance matrix $\Lambda_{\hat{\mu}_t|\hat{u}_0}$ from (101). Note that the mild solution of (101) is derived by

$$\hat{u}_t = \mathcal{T}_{0,t}(\hat{u}_0) + \int_0^t \mathcal{T}_{s,t}\mathbf{G}_s d\mathcal{W}_s \tag{102}$$

where

$$\mathcal{T}_{s,t}(v) = \exp\left(-\int_s^t b(r)(\sigma\mathbf{I} - \mathbf{D})dr\right)(v), \quad v \in \mathbb{R}^d. \tag{103}$$

Thus, $m_{\hat{\mu}_t|\hat{u}_0} = \mathcal{T}_{0,t}(\hat{u}_0)$ Since $\mathbf{D}$ is a symmetric matrix and $\Lambda = (\sigma\mathbf{I} - \mathbf{D})^{-1}$ in this case,

$$\Lambda_{\hat{\mu}_t|\hat{u}_0} \tag{104}$$

$$= \int_0^t \mathcal{T}_{s,t}\mathbf{G}_s\Lambda\mathbf{G}_s^\top\mathcal{T}_{s,t}^\top ds \tag{105}$$

$$= \int_0^t 2b(s)\mathcal{T}_{s,t}(\sigma\mathbf{I} - \mathbf{D})^{-k}\mathcal{T}_{s,t} ds \tag{106}$$

$$= \int_0^t 2b(s)\exp\left(-\int_s^t b(r)(\sigma\mathbf{I} - \mathbf{D})dr\right)(\sigma\mathbf{I} - \mathbf{D})^{-k}\left(-\int_s^t b(r)(\sigma\mathbf{I} - \mathbf{D})dr\right)ds. \tag{107}$$

By the integration by parts,

$$\Lambda_{\mu_t|\hat{u}_0} = \int_0^t 2b(s)(\sigma\mathbf{I} - \mathbf{D})^{-k} ds. \tag{108}$$

Thus, $\rho_{\mathcal{H}_\lambda}^{\mu_t|\hat{u}_0}$ is computed by

$$\rho_{\mathcal{H}_\lambda}^{\mu_t|\hat{u}_0}(u) \tag{109}$$

$$= \left(\int_0^t 2b(s)(\sigma\mathbf{I} - \mathbf{D})^{-k} ds\right)^{-1}\left(m_{\mu_t|\hat{u}_0} - u\right) \tag{110}$$

$$= \left(\int_0^t 2b(s)(\sigma\mathbf{I} - \mathbf{D})^{-k} ds\right)^{-1}\left(\exp\left(\int_0^t b(s)(\mathbf{D} - \sigma\mathbf{I})ds\right)\hat{u}_0 - u\right). \tag{111}$$

Therefore, we can compute the score operator conditioned at $\hat{u}_0$ using Theorem 2.1,

$$\mathbf{S}_\lambda(t, \hat{u}|\hat{u}_0) = \mathbf{G}_t\mathbf{G}_t^*\rho_{\mathcal{H}_\lambda}^{\mu_t|\hat{u}_0}. \tag{112}$$

$\square$

### C.3  Score Operators in HDM-SDE and Wiener Processes

Let a score operator $\mathbf{S}_\lambda(t, \mathbf{x})$ be given. By [32, Lemma 22], for any $\varepsilon > 0$, there exists a number $L(\varepsilon) \in \mathbb{N}$ and continuous linear maps $F_L : [0, T] \times \mathcal{H} \to \mathbb{R}^L, G_L : \mathbb{R}^L \to \mathcal{H}$ as well as $M \in C(\mathbb{R}^L, \mathbb{R}^L)$ such that

$$\sup_{(t,\mathbf{x})\in[0,T]\times\mathcal{H}} \|\mathbf{S}_\lambda(t, \mathbf{x}) - (G_L \circ M \circ F_L)(t, \mathbf{x})\|_{\mathcal{H}} \leq \varepsilon. \tag{113}$$

Here $F_N$ has the form $F_L(t, \mathbf{x}) = (w_1(t, \mathbf{x}), \ldots, w_L(t, \mathbf{x}))$ where $w_1(t, \cdot), \ldots, w_L(t, \cdot) \in \mathcal{H}^*$ for $(t, \mathbf{x}) \in [0, T] \times \mathcal{H}$, and $G_L(\mathbf{v}) = \sum_{j=1}^L v_j\phi^{(j)}$ for $\mathbf{v} \in \mathbb{R}^L$ for a given basis $\{\phi^{(l)}\}_{l=1}^\infty$ of $\mathcal{H}$. Also, by following [32, Appendix B], there exists a truncated score operator $\mathbf{S}_\lambda^L(t, \mathbf{x}) = \sum_{l=1}^L a_l(t, \mathbf{x})\phi^{(l)}$ satisfying $\sup_{(t,\mathbf{x})\in[0,T]\times\mathcal{H}} \|\mathbf{S}_\lambda(t, \mathbf{x}) - \mathbf{S}_\lambda^L(t, \mathbf{x})\|_{\mathcal{H}} \leq \varepsilon$. Therefore, the actual target that the score model learns is the function $M$ satisfying $M \circ F(t, \mathbf{x}) = (a_1(t, \mathbf{x}), \ldots, a_L(t, \mathbf{x}))$ corresponding to $\phi^{(l)}$ in a finite-dimensional space with the given basis as the coordinate basis. Let the function approximating $M$ denotes $\tilde{M} \circ F = (\tilde{a}_1, \ldots, \tilde{a}_L)$. $F_L$ transforms $(t, \mathbf{x})$ into a finite-dimensional coordinate vector $\mathbf{w}_t = (w_1(t, \mathbf{x}), \ldots, w_L(t, \mathbf{x})) \in \mathbb{R}^{L(\varepsilon)}$ for each $t \in [0, T]$. Using $\tilde{M}(\cdot)$, the coefficient vector $\mathbf{w}_t$ corresponding to each $\phi^{(l)}$ are calculated. By combining $M(\mathbf{w}_t)$ into $\phi^{(l)}$, we obtain $\mathbf{y} = \sum_{l=1}^L \tilde{a}_l(t, \mathbf{w})\phi^{(l)} \in \mathcal{H}$ that approximate the value of the score operator. In this way, FNO learns $\tilde{M}$ as a kernel integral operator transformed by Fourier transformation with respect to a fixed basis. We summarize the code block focusing on the used components without writing the full code. We denote denotes MLP as multi-linear perceptron and conv as spectral convolution.

```
1  import torch.fft.ifftn as ifftn
2
3  def conv(x, temb, weight, act, mode_sizes):
4      x = rfftn(x, norm='ortho', dim=(2,3))
5      x = x + Dense(act(temb))[:,:,None,None]
6      x = torch.einsum('bchw, chw->bchw', x, weight)
7      return ifftn(x, s=(mode_sizes), norm='ortho')
8
9  # Motified version of FNO
10 def forward(self, x, t):
11     x = lifting(x)
12     temb = Dense(get_timestep_embedding(t, lifting_channels))
13     x = x + temb[:,:,None]
14     x_fno = conv(x, temb) + fno_skips(x)
15     x_fno = MLP(x_fno, temb)
16     x_fno = projection(x_fno)
17     return x_fno
```

Listing 1: The added forward process of Time-conditioned FNO for 1d examples

**Generating $\mathcal{H}$-valued $\Lambda$-Wiener Processes**    For HDM-SDE, we use a Gaussian square kernel to set the covariance operator $\Lambda$ to implement $\Lambda$-Wiener processes for HDM-SDE (see Example 2.2). The kernel, denoted as $K(\mathbf{x}_1, \mathbf{x}_2) = K(\mathbf{x}_1, \mathbf{x}_2; \texttt{len}, \texttt{gain})$, is expressed as

$$K(\mathbf{x}_1, \mathbf{x}_2; \texttt{len}, \texttt{gain}) = \texttt{gain} \times \exp\left(-\|(\mathbf{x}_1 - \mathbf{x}_2)/\texttt{len}\|^2\right), \tag{114}$$

and the hyperparameters involved are the $\texttt{len}$ and $\texttt{gain}$. We tune the hyperparameters for each dataset; the $\texttt{gain}$ is commonly set to 1 while the length scale $\texttt{len}$ is adjusted between 0.1 and 1. To obtain $\mathcal{H}$-valued Wiener process $\mathbf{W}$, we need to approximate eigenfunctions of the covariance operator $\Lambda$ based on a given train data $D_{\text{train}} = \{\mathbf{x}_i\}_{i=1}^N$. To estimate eigenfunctions, we first compute the eigenvalues $\{\lambda^{(\ell)} > 0\}_{\ell=1}^N$ and eigenvectors $\{\Phi^{(\ell)} \in \mathbb{R}^N\}_{\ell=1}^N$ due to the eigendecomposition of the Gram matrix $\mathbf{K} \in \mathbb{R}^{N \times N}$

$$\mathbf{K} = \Phi \mathbf{D} \Phi^\top, \quad \mathbf{D} = \text{diag}(\{\lambda^{(\ell)}\}), \quad \Phi = [\Phi^{(1)}, \dots, \Phi^{(N)}], \tag{115}$$

where the Gram matrix $\mathbf{K}_{ij} = K(\mathbf{x}_i, \mathbf{x}_j)$ is evaluated on $D_{\text{train}}$. Then by multiplying $\Phi^\top$ and $\mathbf{D}^{\frac{1}{2}}$ to an $N$-dimensional Gaussian random variable $\mathbf{Z} \sim \mathcal{N}(0, \mathbf{I})$ yields the $\Lambda$-Wiener process evaluated at $D_{\text{train}}$ as follows

$$\mathbf{W} = \mathbf{Z} \mathbf{D}^{\frac{1}{2}} \Phi^\top. \tag{116}$$

The detailed code follows:

```
1  from scipy import distance
2  from torch.linalg import eigh
3
4  def kernel(x, y, hyp_gain=1.0, hyp_len=1,  metric=metric):
5      D = distance.cdist(x/hyp_len,y/hyp_len,'sqeuclidean')
6      K = hyp_gain*np.exp(-D)
7      return torch.from_numpy(K).to(torch.float32)
8
9  def sample_wiener_process_on_grid(hyp_gain=1.0, hyp_len=1, size):
10     x = y = torch.linspace(0.0,1.0,grid).reshape((-1,1))
11     K = kernel(x, y, hyp_len, hyp_gain, metric=metric)
12     g_dim = K.shape[0]
13     eig_val, eig_vec = eigh(K+1e-6*torch.eye(g_dim, g_dim))
14     D = torch.sqrt(torch.diag(eig_val))
15     M = torch.matmul(eig_vec, D)
16     size = list(size) # batch * grid
17     x_0 = torch.randn(size)
18     output = x_0 @ M.transpose(0,1)
19     return output # batch * grid
```

Listing 2: Generating Wiener processes with kernel $K$

Since our goal is to infer the value at an arbitrary $\mathbf{x}_* \in D_{\text{test}}$, which is unseen in the training phase, we can evaluate $\mathbf{W}(\mathbf{x}_*)$ as follows:

$$\mathbf{W}(\mathbf{x}_*) = \mathbf{Z}\mathbf{D}^{-\frac{1}{2}}\mathbf{\Phi}^{\top}\mathbf{K}_{\mathbf{x}_*}. \tag{117}$$

where $[\mathbf{K}_{x_*}]_{i1} = \mathbf{K}(\mathbf{x}_i, \mathbf{x}_*)$ for unseen data $\mathbf{x}_*$.

## C.4 Evolution Operators of HDM-SPDE

The eigenvector $\phi_{n,m}(x,y)$ and eigenvalue $\lambda_{n,m}$ of the negative Laplacian operator satisfying the zero Neumann boundary condition are given by the following equation:

$$\nabla^2 \phi_{n,m}(x,y) = -\lambda_{n,m}\phi_{n,m}(x,y) \tag{118}$$

$$\frac{\partial \phi_j(x,y)}{\partial x} = \frac{\partial \phi_j(x,y)}{\partial y} = 0 \tag{119}$$

where $n$ and $m$ are indices indicating the mode shapes and frequencies of the eigenvectors and eigenvalues, respectively.

$$\phi_{n,m}(x,y) \sim \cos\left(\frac{\pi n x}{W}\right)\cos\left(\frac{\pi m y}{H}\right) \tag{120}$$

$$\lambda_{n,m} = \pi^2 \left(\frac{n^2}{W^2} + \frac{m^2}{H^2}\right). \tag{121}$$

Then the negative Laplacian $-\Delta$ can be represented as a eigenvalue decomposition $-\Delta \triangleq \mathbf{V}\mathbf{D}\mathbf{V}^{\top}$, where $\mathbf{V}$ is the matrix with column vectors of the finite-dimensional eigenvectors of $-\Delta$ and $\mathbf{D}$ is the diagonal matrix consisting of eigenvalues of $-\Delta$.

## C.5 General Setting of HDM

**Forward and Reverse Evolution of HDM-SDE** The forward process employs a beta schedule, expressed in the following equation.

$$d\mathbf{X}_t = -\frac{\beta(t)}{2}\mathbf{X}_t dt + \sqrt{\beta(t)}\Lambda^{\frac{1}{2}}d\mathbf{W}_t. \tag{122}$$

We denote $\sigma$ as `sigma`, $k$ as `index` and $\tau(t) = \exp(-\int_0^t \frac{\beta(s)}{2}ds)$ as `tau` in the below listing.

```python
import math
import torch
s0 = 0.0008
def beta(self, t):
    beta = math.pi / 2 * 2 * (s0 + 1) * torch.tan((t + s0)/(1 + s0) *
    math.pi / 2)
    return beta

def marginal_log_mean_coeff(t):
    cosine_log_alpha_0 = math.log(math.cos(s0/ (1. + s0) * math.pi /
    2.))
    log_alpha_fn = lambda s: torch.log(torch.cos((s + s0) / (1. + s0)
    * math.pi / 2.))
    log_alpha_t =  log_alpha_fn(t) - log_alpha_fn(0)
    return log_alpha_t

def diffusion_coeff(t):
    return torch.exp(marginal_log_mean_coeff(t))

def marginal_std(t):
    return torch.pow(1. - diffusion_coeff(t)**2, 1 /2)
```

Listing 3: Computing the solution for HDM-SDE

Then, the solution of this forward process can be represented as:

$$\mathbf{X}_t = m_{\mu_t|\mathbf{X}_0} + \Lambda_{\mu_t|\mathbf{X}_0}(\epsilon) \tag{123}$$

where $\epsilon \sim \mathcal{N}(0, \mathbf{I})$. We can simulate the forward process as follow:

```
def forward_process(x,t,e):
    sigma = marginal_std(t)
    x_coeff = diffusion_coeff(t)
    x_t = x0 * x_coeff[:,None,None,None] + e * sigma[:,None,None,None]
    return x_t
```

Listing 4: Simulating the forward process for HDM-SDE

For sampling, the Euler-Maruyama method is utilized to do the reverse process corresponding to the forward process. The reverse process corresponding to the forward process follows below.

$$d\mathbf{X}_t = \left[ \frac{\beta(t)}{2}\mathbf{X}_t + \mathbf{s}_\theta(t, \mathbf{x}_t) \right] dt + \sqrt{\beta(t)}\Lambda^{\frac{1}{2}}d\mathbf{W}_t. \tag{124}$$

When we apply the Euler-Maruyama method to the reverse process, we can get a sequence $(\mathbf{x}_t)$ that converges to data in the data space.

$$\mathbf{x}_t = \mathbf{x}_s + \left[ \frac{\beta(s)}{2}\mathbf{x}_s + \mathbf{s}_\theta(s, \mathbf{x}_s) \right] \Delta t + \sqrt{\beta(s)\Delta t}\Lambda^{\frac{1}{2}}(\mathbf{W}), \quad t < s. \tag{125}$$

where $\mathbf{W} \sim \mathcal{GP}(0, \Lambda)$ and $\Delta t = s - t$. The implementation of the reverse sampling follows:

```
def sampling(x,y,model,resolution_grid):
    def sde_score_update(x, s, t):
        score_s = model(x,t) * torch.pow(marginal_std(s), -1)[:,None,
    None,None]
        beta_step = sde.beta(s) * (s - t)
        x_coeff = 1 + beta_step / 2
        noise_coeff = torch.pow(beta_step, 1 / 2)
        score_coeff = beta_step
        if resolution_grid == None:
            e = sample_wiener_process_on_grid(x.shape)
        else:
            e = sample_wiener_process(resolution_grid)
        x_t = x_coeff[:, None, None, None] * x + score_coeff[:, None,
    None, None] * score_s + noise_coeff[:, None, None,None] * e

        return x_t

    # Sampling
    timesteps = torch.linspace(T,eps, steps + 1)
    with torch.no_grad():
        for i in range(steps):
            vec_s, vec_t = torch.ones((x.shape[0],)) * timesteps[i],
    torch.ones((x.shape[0],)) * timesteps[i + 1]
            x = sde_score_update(x, vec_s, vec_t)
    return x
```

Listing 5: Reverse sampling for HDM-SDE

**Forward and Reverse Evolution of HDM-SPDE**    Recall the eigendecomposition of the negative Laplacian $-\Delta = \mathbf{V}\mathbf{D}\mathbf{V}^\top$ due to the DCT, and we write $\hat{U}(t) = \mathbf{V}^\top \mathbf{X}(t)$. The forward process of HDM-SPDE employs the beta scheduling [46], expressed in the following equation.

$$d\hat{U}(t) = -\frac{\beta(t)}{2}(\sigma\mathbf{I} + \mathbf{D})\hat{U}(t)dt + \sqrt{\beta(t)}(\sigma\mathbf{I} + \mathbf{D})^{-\frac{k}{2}}d\mathcal{W}_t. \tag{126}$$

where $\mathcal{W}_t$ is a space-time white noise. Important parameters are sigma, $\sigma$ and k, $k \in [0, \infty)$. We usually set index $k \in \{0, 0.1\}$ and $\sigma = 1$ for the image generation task. In the 2D image experiments, the forward process includes the new term $(\sigma\mathbf{I} + \mathbf{D})$ in both the drift term and diffusion term, unlike the 1D experiments.

```
1  from torch_dct import dct_2d, idct_2d
2  k = image_size
3  freqs = np.pi*torch.linspace(0, k-1, k)/k
4  D = freqs[None,None,:, None]**2 + freqs[None,None,None, :]**2
5  s0 = 0.0008
6  def beta(t):
7      beta = math.pi / 2 * 2 / (s0 + 1) * torch.tan((t + s0)/(1 + s0) *
       math.pi / 2)
8      return beta
9
10 def marginal_log_mean_coeff(t):
11     log_alpha_fn = lambda s: torch.log(torch.cos((s + s0) / (1. + s0)
       * math.pi / 2.))
12     log_alpha_t =  log_alpha_fn(t) - log_alpha_fn(0)
13     return log_alpha_t
14
15 def Lambda(t):
16     t = - 2*marginal_log_mean_coeff(t)
17     a = 1-torch.exp(-t[:,None,None,None]*(sigma+D))
18     b = sigma+D
19     return a/b
20
21 def square_Lambda(self,t):
22     return torch.sqrt(self.Lambda(t))
23
24 def diffusion_coeff(x,t):
25     t = - marginal_log_mean_coeff(t)
26     k = x.shape[-1]
27     dct_coefs = dct_2d(x, norm='ortho')
28     dct_coefs = dct_coefs * torch.exp(- (sigma+D)* t[:,None,None,None
       ])
29     return idct_2d(dct_coefs, norm='ortho')
30
31 def marginal_std(x,t,index):
32     dct_coefs =dct_2d(x, norm='ortho')
33     c = square_Lambda(t)
34     c *= torch.pow((sigma+D),-index/2)
35     dct_coefs = dct_coefs * c
36     return idct_2d(dct_coefs, norm='ortho')
37
38 def inverse_marginal_std(x,t):
39     dct_coefs = dct_2d(x, norm='ortho')
40     c = torch.pow(Lambda(t)+1e-5,-1)
41     dct_coefs = dct_coefs * c
42     return idct_2d(dct_coefs, norm='ortho')
43
44 def laplacian(x,t,s):
45     t = (t-s)
46     k = x.shape[-1]
47     freqs = np.pi * torch.linspace(0, k-1, k) / k
48     dct_coefs = dct_2d(x, norm='ortho')
49     dct_coefs = dct_coefs * ((sigma+D) * t[:,None,None,None])
50     return idct_2d(dct_coefs, norm='ortho')
51
52 def marginal_laplacian(x,t,s, index):
53     t = (t-s)
54     k = x.shape[-1]
55     dct_coefs = dct_2d(x, norm='ortho')*torch.sqrt(torch.abs(t))[:,
       None,None,None]
56     dct_coefs = dct_coefs.to(x.device) * torch.pow(torch.abs((sigma+D
       )), index/2)
57     return idct_2d(dct_coefs, norm='ortho')
```

Listing 6: Computing the solution for HDM-SPDE

Since $\mathbf{D}$ is a diagonal matrix, the corresponding exact formula of the forward process is represented by a more complex form than the 1D case. Let $\tau(t) = \int_0^t \frac{\beta(u)}{2} du$. The solution of this forward process can be represented as:

$$\hat{U}(t) = e^{-\tau(t)(\sigma\mathbf{I}+\mathbf{D})}\hat{U}(0) + \int_0^t e^{-\frac{\tau(t)}{\tau(u)}(\sigma\mathbf{I}+\mathbf{D})}(\sigma\mathbf{I}+\mathbf{D})^{-k/2}\sqrt{\beta(u)}d\mathcal{W}_u. \quad (127)$$

Then

$$\hat{U}(t) \stackrel{d}{=} m_{\mu_t|\hat{U}_0} + \Lambda_{\mu_t|\mathbf{X}_0}^{\frac{1}{2}}(\epsilon), \quad \epsilon \sim \mathcal{N}(0,\mathbf{I}), \quad (128)$$

where

$$m_{\mu_t|\hat{U}_0} = \exp(-\tau(t)(\sigma\mathbf{I}+\mathbf{D}))\hat{U}_0, \quad \Lambda_{\mu_t|\hat{U}_0}^{\frac{1}{2}} = (\mathbf{I} - m_{\mu_t|\mathbf{X}_0})^{\frac{1}{2}}(\sigma\mathbf{I}+\mathbf{D})^{-\frac{k}{2}}. \quad (129)$$

```
def tau(t): # depend on the choice of beta schedules
    return weighted_time

def sqaure_Lambda(self, t):
    t = tau(t)
    a = 1 - torch.exp(-t[:,None,None,None]*(sigma+D))
    b = D + sigma
    return torch.sqrt(a/b)

def mean_evolution(x,t):
    t = tau(t)
    k = x.shape[-1]
    dct_coefs = dct_2d(x, norm='ortho')
    dct_coefs = dct_coefs * torch.exp(-(sig+D) * t[:,None,None,None])
    return idct_2d(dct_coefs,norm='ortho')

def noise_evolution(x,t,index):
    dct_coefs = dct_2d(x, norm='ortho')
    c = square_Lambda(t)
    c *= torch.pow((sigma+D),-(index)/2)
    dct_coefs = dct_coefs * c
    return idct_2d(dct_coefs, norm='ortho')

def evolution_operator(model, x, e,t,index):
    x_mean = mean_evolution(x, t)
    noise = noise_evolution(e, t)
    return x_mean + noise
```

Listing 7: Evolution operator algorithm for HDM-SPDE

For sampling, the Euler-Maruyama method is utilized to do the reverse process corresponding to the forward process. The reverse process corresponding to the forward process follows below.

$$d\hat{U}(t) = \left((\sigma\mathbf{I}+\mathbf{D})\frac{\beta(t)}{2}\hat{U}(t) + \mathbf{V}^\top s_\theta(t,\mathbf{V}\hat{U}(t))\right)dt + \sqrt{\beta(t)}(\sigma\mathbf{I}+\mathbf{D})^{-\frac{k}{2}}d\mathcal{W}_t. \quad (130)$$

When we apply the Euler-Maruyama method to the reverse process, we can get a sequence $(\mathbf{x}_t)$ that converges to data in the data space.

$$\begin{aligned} \mathbf{V}\hat{U}(t) = \mathbf{V}\hat{U}(s) + \left[\mathbf{V}(\sigma\mathbf{I}+\mathbf{D})\frac{\beta(s)}{2}\hat{U}(s) + s_\theta(t,\mathbf{V}\hat{U}(s))\right]\Delta t \\ + \mathbf{V}\sqrt{\beta(s)\Delta t}(\sigma\mathbf{I}+\mathbf{D})^{-\frac{k}{2}}\epsilon \end{aligned} \quad (131)$$

where $\Delta t = s - t$ and $\epsilon \sim \mathcal{N}(0,I)$.

```
def sampling(x,model):
    def sde_score_update(x, s, t):
        score_s = inverse_marginal_std(model(x,s),s)
        e = torch.randn(x.shape)
```

```
5        x_t = -beta(s)[:,None,None,None]*laplacian(x,t,s)/2+x
6        x_t += -beta(s)[:,None,None,None]*score_s*(t-s)[:,None,None,
    None]
7        x_t += torch.sqrt(beta(s))[:,None,None,None]*
    marginal_laplacian(e,t,s)
8        return x_t
9    # Sampling steps
10   timesteps = torch.linspace(T, eps, steps + 1)
11   with torch.no_grad():
12       for i in tqdm.tqdm(range(steps)):
13           vec_s = torch.ones((x.shape[0],)) * timesteps[i]
14           vec_t = torch.ones((x.shape[0],)) * timesteps[i + 1]
15           x = sde_score_update(x, vec_s, vec_t)
16   return x
```

Listing 8: Reverse sampling for HDE-SPDE

**Sampling Algorithms via Euler-Maruyama** in HDM-SDE and HDM-SPDE as follows. We summarize the algorithms for reverse sampling

---

**Algorithm 1:** Sampling for HDM-SDE

1: $\mathbf{W} \sim \mathcal{GP}(0, \Lambda)$
2: $\mathbf{x}_T \leftarrow \mathbf{W}$
3: For $t < s$:
4: $\quad \mathbf{W} \sim \mathcal{GP}(0, \Lambda)$
5: $\quad \Delta t = s - t$
6: $\quad \mathbf{x}_t \leftarrow \mathbf{x}_s + \frac{\beta(s)}{2}\mathbf{x}_s \Delta t + \mathbf{s}_\theta(s, \mathbf{x}_s)\Delta t$
7: $\quad \mathbf{x}_t \leftarrow \mathbf{x}_t + \sqrt{\beta(s)\Delta t}\mathbf{W}$
8: $\quad \mathbf{x}_s \leftarrow \mathbf{x}_t$

---

**Algorithm 2:** Sampling for HDM-SPDE

1: $\epsilon \sim \mathcal{N}(0, I)$
2: $\mathbf{x}_T = \epsilon$
3: For $t < s$:
4: $\quad \epsilon \sim \mathcal{N}(0, I)$
5: $\quad \Delta t = s - t$
6: $\quad \mathbf{x}_t \leftarrow \mathbf{x}_s + \frac{\beta(s)}{2}\mathbf{V}(\sigma\mathbf{I} + D)\mathbf{V}^\top(\mathbf{x}_s)\Delta t$
7: $\quad \mathbf{x}_t \leftarrow \mathbf{x}_t + s_\theta(s, \mathbf{x}_s)\Delta t$
8: $\quad \mathbf{x}_t \leftarrow \mathbf{x}_t + \mathbf{V}\left(\sqrt{\beta(s)\Delta t}(\sigma\mathbf{I} + \mathbf{D})^{-\frac{k}{2}}\epsilon\right)$
9: $\quad \mathbf{x}_s \leftarrow \mathbf{x}_t$

---

# D   Experiment and Empirical Results

## D.1   Function Generation in Quadratic, Melbourne, and Gridwatch Datasets

**Datasets**   We use the Quadratic, Melbourne, and Gridwatch datasets for experiments.

**Quadratic** The quadratic dataset is a synthetic dataset that represents bimodality. The sample consists of a functional evaluation of $f(x; a, b) := ax^2 + b$, where $a$ is randomly chosen from $\{-1, 1\}$, and $b \sim \mathcal{N}(0, 1)$. We use an equally spaced grid of 100 points at $[-10, 10]$ as inputs of training data $\mathcal{X}_{\text{train}}$. We extract 1,000 $(a, b)$ pairs to construct a training data of functionals $D_{\text{train}}$ as follows

$$D_{\text{train}} = \{(x, f(x; a, b)) : x \in \mathcal{X}_{\text{train}}, a \sim \{-1, 1\}, b \sim \mathcal{N}(0, 1)\}. \tag{132}$$

To generate $\mathcal{H}$-valued Wiener processes, we first sample

$$\mathcal{X}_{\text{noise}} = \{\mathbf{x}_j = (x_{j,1}, \ldots, x_{j,100}) \in \mathbb{R}^{100} : j = 1, \ldots, 1000\}, \tag{133}$$

where $x_{j,i} \sim \mathcal{U}(-10, 10)$ for each $j = 1, \ldots, 1000$ and $i = 1, \ldots, 100$, and then construct $\mathbf{W}$ according to (117) by computing the Gram matrix $K_\lambda(\mathbf{x}_j, \mathbf{x}_{j'})$ on $\mathbf{x}_j, \mathbf{x}_{j'} \in \mathcal{X}_{\text{noise}}$ with $N = 100$. To apply the kernel test [56], we use the test data $D_{\text{test}}$, which is constructed by following the same procedure for constructing $D_{\text{train}}$ by setting $\mathcal{X}_{\text{test}} = \mathcal{X}_{\text{train}}$. By setting $\mathcal{X}_{\text{test}} \neq \mathcal{X}_{\text{noise}}$, we can verify that our model can generate functionals by reversing HDM-SDE which uses Wiener processes with values in Hilbert space.

**Melbourne**[4] The Melbourne dataset is a real-world dataset that contains the number of pedestrians recorded different locations. Each sample records measurements taken hourly for 24 hours. Before the experiments, we removed the recordings with unknown values and normalized that each recording have zero mean and unit variance. We use 1,855 data for training and use 232 data for testing.

**Gridwatch**[5] The gridwatch dataset is a real-world dataset that documents the energy demand of the UK grid. We preprocess the dataset by selecting the Demand field. We obtain the preprocessing step

---

[4]Source: http://www.timeseriesclassification.com/description.php?Dataset=MelbournePedestrian
[5]Source: https://www.gridwatch.templar.co.uk/download.php

as follows. First of all, select a date on which measurements were taken at exactly 5 minutes. Second, remove days containing the difference between two consecutive readings every 5 minutes beyond the 99.5th percentile of the difference. In addition, we remove days where continuous readings do not change for more than 30 minutes. Moreover, we center the dataset so that each sample has a mean of 0. Finally, we adjust the dataset so that each sample has a unit variance.

**Implementation Details** We tune the length scale (len) parameter of kernel $K$ to generate better sampling quality using the Wiener process, 0.8 in Quadratic, 2.0 in Melbourne, and 1.8 in Gridwatch, which is mentioned in C.3. We commonly use gain parameter 1.0 for all datasets. Grid parameter adjusted to 24 in Melbourne, 288 in Gridwatch, and 100 in Quadratic. In VP-SDE, we used cosine beta schedule [46], which is mentioned in C.5. We tune the architecture of FNO [37] which is mentioned in Section 3; the lifting and projection channels to 256, the number of layers is 4, and MLP's expansion and dropout value is 4.0 and 0.0, respectively. We use gelu function for the activation function and the group normalization. Moreover, we use the pre-activation parameter and separable as True, and fft normalization parameter is ortho. We train our model on each data over 1,000 iterations, except for the Gridwatch data, on which we train 1,500 iterations until the convergence. We set the same iteration numbers for training our model, DDPM [25], and NDP [18] for a fair comparison. We use a single RTX 2080 GPU for each experiment, which takes approximately 330 seconds for training. For sampling, the experiment takes approximately 90 seconds. The number of sampling steps (NFE) is 1,000. For a quantitative evaluation, we calculate the power of a kernel two-sample test [56] based on Maximum Mean Discrepancy (MMD), which attempts to discriminate between models and samples in the functional data. We clamp the range of generated functions as post-processing to stabilize the results of kernel tests. We perform 30 times two-sample tests and report the power with 95%-confidence intervals.

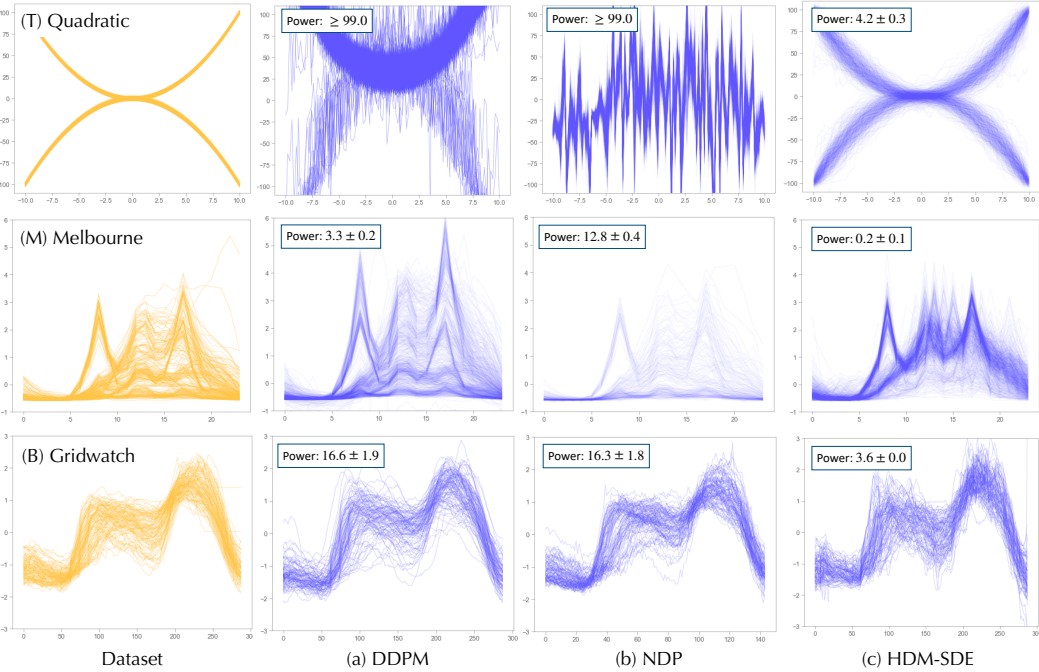

Figure A.1: Comparison of functional samples generated by DDPM (VP-SDE) [25], NDP [18], and HDM-SDE (ours) in different 1D datasets; (top) Quadratic, (middle) Melbourne, and (bottom) Gridwatch.

## D.2 Function Generation in AEMET Dataset

We also experiment with our method on the real-world dataset AEMET[6], a renowned dataset in functional data analysis, to compare it with the Functional DDPM, presented in [29]. We utilize the SE

---

[6]Source: https://www.aemet.es/es/portada

kernel for HDM-SDE instead of the Mátern kernel used in [29], while the rest of the hyperparameter settings are the same. The power(%) of two-sample test for HDM-SDE records a value of $0.48_{\pm 0.95}$, implying that the statistical difference between the generated and the true dataset is negligible. The quantitative comparison between the Functional DDPM [29] and the HDM-SDE in the AEMET dataset is reported in Table A.1.

| | Mean | Variance | Autocorr |
|---|---|---|---|
| Functional DDPM [29, Table 3] (the best result) | 0.0152 | 1.0748 | 2.551e-06 |
| HDM-SDE (ours) | **0.0001** | **0.0293** | **1.127e-06** |

Table A.1: Comparison between the Functional DDPM [29] and the proposed method in the AEMET dataset. Following [29], we compute the average pointwise mean, pointwise variance, and average autocorrelation of both the real and generated functions and evaluate the Mean-Square Error (MSE) between the statistics of generated samples and of the true data. Lower is better.

## D.3 Motion Generation

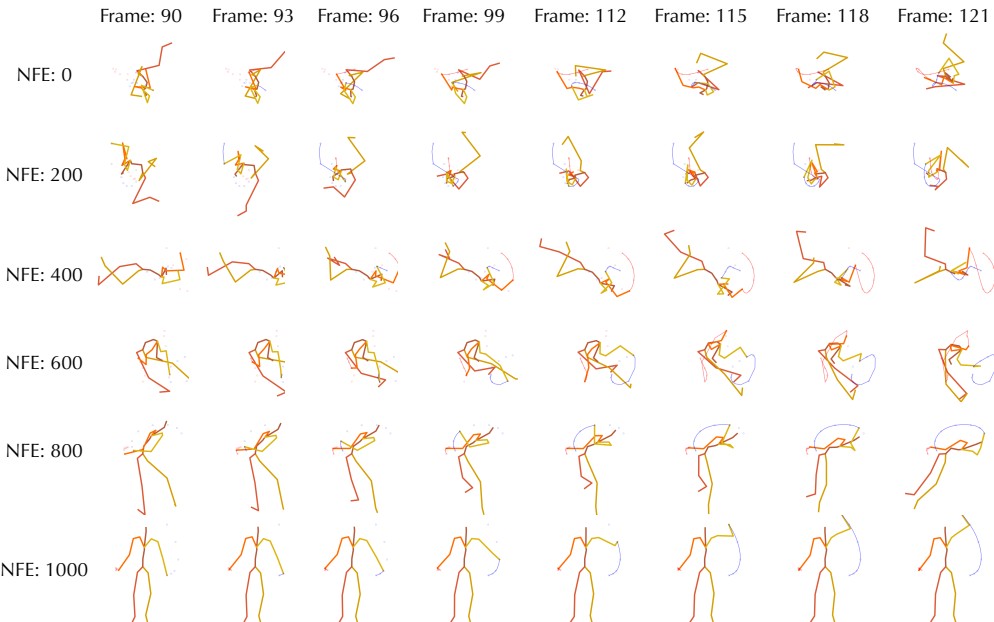

Figure A.2: Example of sampling process of motion through HDM-SDE.

**Data Preprocessing** For the motion generation task, we utilize the HumanML3D [22] dataset, which contains motions from AMASS [41] and HumanAct12 [21] datasets with additional text labels. In particular, it contains 14,616 unique motions. Among all motions, 100 motions are randomly selected with 20 FPS, and we trim the motions to have 128 frames. All the motion sequences are rotated and translated so that the pelvis joint of the first frame faces the $Z$ direction and is located at the origin. When constructing the test data, 100 motions are selected from among the unique motions that have similarities with training samples; precisely, we restrict the motion of test data to having the power of a kernel two-sample test based on MMD less than 0.05 compared with the training data.

**Motion Representation** A motion is represented as a sequence of skeletons where each skeleton follows a skinned multi-person linear model (SMPL) [40] format that has 24 joints. Each joint represents a rotation in three-dimensional space where 6D representation [59] is used, making the

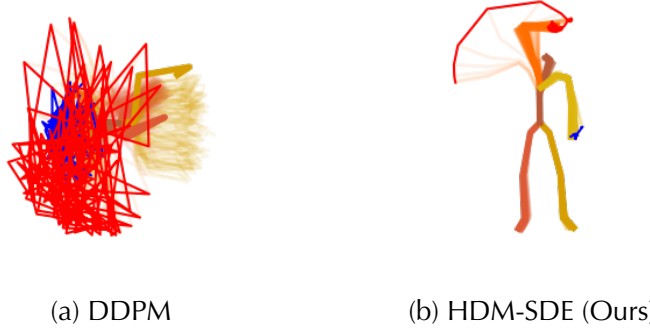

(a) DDPM       (b) HDM-SDE (Ours)

Figure A.3: Generated motions of (a) DDPM and (b) HDM-SDE (ours).

dimension of each skeleton to be $24 \times 6 = 144$. Specifically, a motion sequence is denoted by $M = [m_1, \ldots, m_{128}] \in \mathbb{R}^{128 \times 144}$.

**Length Parameter Selection** To generate Wiener processes with values in Hilbert space for each dimension, we first optimize the length parameters of 144-dimensional motion data. In particular, we utilize the squared exponential kernel function (114) $K_d(\mathbf{x}, \mathbf{x}'; \texttt{len}_d)$, and tune the length parameter $\texttt{len}_d$ for each dimension $d$ due to the maximum likelihood estimation of a Gaussian Process [55],

$$\texttt{len}_d^* = \operatorname*{argmax}_{\texttt{len}_d} \left( -\frac{1}{2} m_d^\top (\mathbf{K}_d + \sigma_n^2 \mathbf{I})^{-1} m_d - \frac{1}{2} \log |\mathbf{K}_d + \sigma_n^2 \mathbf{I}| - \frac{n}{2} \log 2\pi \right), \qquad (134)$$

where $\mathbf{K}_d \in \mathbb{R}^{128 \times 128}$ is the Gram matrix, $\sigma_n^2 = 10^{-6}$ is the expected measurement noise, and $n = 128$ is the motion length.

**Implementation Details** We use the HDM-SDE, including multi-dimensional FNO. The input and output channel sizes of the FNO are changed from 1 to 144 to handle 144-dimensional trajectories. The number of the Fourier layer is 4 and the expansion number of the MLP is 4. We use `gelu` activation and group normalization. The input is normalized by $1/\sqrt{n}$ in FFT. Cosine scheduling [46] is used for beta scheduling. We compare our method with DDPM (VP-SDE) [25], replacing U-Net with the FNO architecture using the same setting as ours. Thus, the baseline and our approach utilize identical network architectures and loss function $L_{\text{SDE}(\theta)}$, and the only difference is the covariance operators used in generating Wiener processes in forward and reverse processes. Precisely, DDPM (VP-SDE) uses space-time white noise, i.e., $\Lambda = \mathbf{I}$, whereas HDM-SDE uses a correlated Wiener process based on the kernel function $K_d$ (114) related to a certain RKHS (see Example 2.2). We train our model over 2,000 epochs with a single RTX 3090 24GB GPU. The number of NFE is 1,000.

**Experimental Results** Sampling a motion sequence through the proposed method is illustrated in Figure A.2. The synthesized motions of the proposed method and DDPM (VP-SDE) are shown in Figure A.3, where we overlay and left hands trajectories with red and blue, respectively. While the proposed method successfully generates human motions, DDPM (VP-SDE) fails to generate meaningful motions. We sample 3,000 motions with the baseline and HDM-SDE, then measure the mean with 95%-confidence intervals of the MMD in 30 times trials between every 100 samples and 100 evaluation data. Since the task space trajectory is expressed as $x, y, z$, the average of the individual values is reported for each joint. Quantitative results are shown in Table A.2, where the proposed method significantly outperforms the baseline in all joints.

### D.4 Image Generation

**Datasets** We follow the same dataset configurations as described in IHDM [50] on image experiments: MNIST [36], CIFAR-10 [42], LSUN-Churches [58], AFHQ [13] and FFHQ dataset [28]. The resolution has not been changed on MNIST and CIFAR-10 datasets where LSUN-Churches images are resized into 128×128. For higher resolution datasets, AFHQ and FFHQ, we resized the image into 256×256. While training a model, we apply random horizontal flips with a probability of 0.5 on all datasets.

|  | DDPM (VP-SDE) [25] | HDM-SDE (Ours) |
|---|---|---|
| Right Hand | $0.388_{\pm 0.065}$ | $\mathbf{0.071}_{\pm 0.031}$ |
| Left Hand | $0.218_{\pm 0.051}$ | $\mathbf{0.037}_{\pm 0.024}$ |
| Right Foot | $0.614_{\pm 0.067}$ | $\mathbf{0.059}_{\pm 0.018}$ |
| Left Foot | $0.764_{\pm 0.057}$ | $\mathbf{0.054}_{\pm 0.018}$ |

Table A.2: A quantitative comparison between DDPM (VP-SDE) [25] the HDM-SDE in terms of the average of measured MMD between generated motions and evaluation data. Lower is better.

| Dataset (Resolution) | Num res-blocks | Layer multipliers | Self-attention resolution | Dropout probability (%) |
|---|---|---|---|---|
| MNIST (28×28) | 1 | [1, 2, 4, 8] | [32, 64, 128, 256], [256, 128, 2, 1] | 10 |
| CIFAR (32×32) | 2 | [1, 2, 2, 2] | 16 | 10 |
| LSUN-church (128×128) | 2 | [1, 1, 2, 2, 4, 4] | 16 | 0 |
| AFHQ (256×256) | 2 | [1, 1, 2, 2, 4, 4] | 16 | 10 |
| FFHQ (256×256) | 2 | [1, 1, 2, 4, 4] | 16 | 10 |

Table A.3: U-Net model architecture parameters for 2D image datasets followed by [52].

**Implementation Details** All the experiments are done with 8 NVIDIA V100 GPUs in a distributed setting. We use the 2D U-Net model [51] followed by [52]. Detailed architecture parameters are described in Table A.3. We use cosine scheduling [46] for beta-scheduling with $\beta_0 = 0.01$ and $\beta_1 = 20$. We set $T = 0.9946$ to prevent numerical instability. We train our model on the CIFAR-10 dataset during 50,000 iterations with 256 batch sizes, which takes approximately 9 hours for training. We train our model on the LSUN-Churches dataset during 125,000 iterations with 256 batch sizes, which takes approximately 50 hours. On AFHQ and FFHQ datasets, we train our model with 64 batch sizes during 200,000 and 450,000 iterations, respectively, which requires approximately 60 and 92 hours. Table A.4 summarizes our training configurations.

**Evaluation** We use clean FID [47] with pre-trained InceptionV3 [53] to compare sample quality. Following IHDM [50], we compute the FID score with the generated sample images and the original dataset. Precisely, on CIFAR-10 and LSUN-church datasets, we use the train set as a reference dataset while the entire set is used for AFHQ and FFHQ datasets, and we use 50,000 generated samples on CIFAR10 and LSUN-church datasets whereas 10,000 samples are generated on AFHQ and FFHQ datasets. The total number of function evaluations (NFE) is set to 1,000. To compare IHDM and the proposed model in terms of diversity, we retrain IHDM model on CIFAR-10 dataset and report PRDC [45] of IHDM and HDM-SPDE.

**Connection between HDM and IHDM** A prior from IHDM is a low-dimensional blur image generated at the frequency domain with reasonably small Gaussian noise controlled with $\delta$. We generate an IHDM prior through DCT blurring using the same configuration as IHDM[7]: $K = 200$, $\sigma_0 = 0.5$ and $\sigma_K = 24$ is set for CIFAR10 dataset; $K = 400$, $\sigma_0 = 0.5$, and $\sigma_K = 64$ is set for LSUN-church; and $K = 200$, $\sigma_0 = 0.5$, and $\sigma_K = 128$ is set for AFHQ and FFHQ. We fix $\delta = 0.01$ for scheduling Gaussian noise on IHDM prior. Since HDM-SPDE is trained with the noise scheduling process, scaling the prior generated by IHDM is required to fit in our noise space. We use the factor of 0.5 on LSUN-Churches and 0.01 on AFHQ and FFHQ datasets to reduce the prior range. Without retraining our model on the noise space of IHDM, adjusting the prior ranges enables our model to sample similar images as HDM does.

### D.5 Controllable Coarse-to-Fine Generation

Motivated from [4], we present how HDM-SPDE can generate high-fidelity images from degraded images. To perform coarse-to-fine generation tasks from degraded images that contain low-frequency information, HDM-SPDE can reconstruct high-frequency images lost in degraded images since HDM-SPDE is trained on the frequency domain. Inpainting can be done by leveraging high frequencies on the masked portion and low frequencies on the unmasked image. Super-resolution including depixelate and deblurring can be done by masking a portion of data to fill out high frequencies of data and translating generated samples from the frequency domain into the image domain. Consequently,

---

[7]Source: https://github.com/AaltoML/generative-inverse-heat-dissipation

| Dataset (Resolution) | Learning rate | Gradient clipping | Sigma | index | batch size | iterations |
|---|---|---|---|---|---|---|
| MNIST ($28 \times 28$) | $1 \times 10^4$ | ✓ | 0.1 | 1 | 128 | 100,000 |
| CIFAR ($32 \times 32$) | $1 \times 10^4$ | ✓ | 1 | 0 | 256 | 50,000 |
| LSUN-church ($128 \times 128$) | $1 \times 10^5$ | ✓ | 1 | 0.1 | 256 | 125,000 |
| AFHQ ($256 \times 256$) | $1 \times 10^5$ | ✓ | 1 | 0.1 | 64 | 200,000 |
| FFHQ ($256 \times 256$) | $1 \times 10^5$ | ✓ | 1 | 0.1 | 64 | 450,000 |

Table A.4: U-Net model training parameters for 2D image datasets followed by [52].

HDM-SPDE can generate high-resolution images from degraded images as [4] do without the need for extra training on a conditional generation as shown in Figure A.4.

Image inpainting involves restoring masked portions or randomly scattered parts of an image where our model is able to fill out finer information by inpainting at high frequency. We use reverse sampling of $\mathbf{x}_s$ until reaching time $t$, and the masked data $\mathbf{x}^{\text{data}}$ is passed through the forward process into $\mathbf{x}_t^{\text{data}}$ and concatenated with $\mathbf{x}_t$ to create $[\mathbf{x}_t, \mathbf{x}_t^{\text{data}}]$. Finally, this concatenated input is fed into the score model.

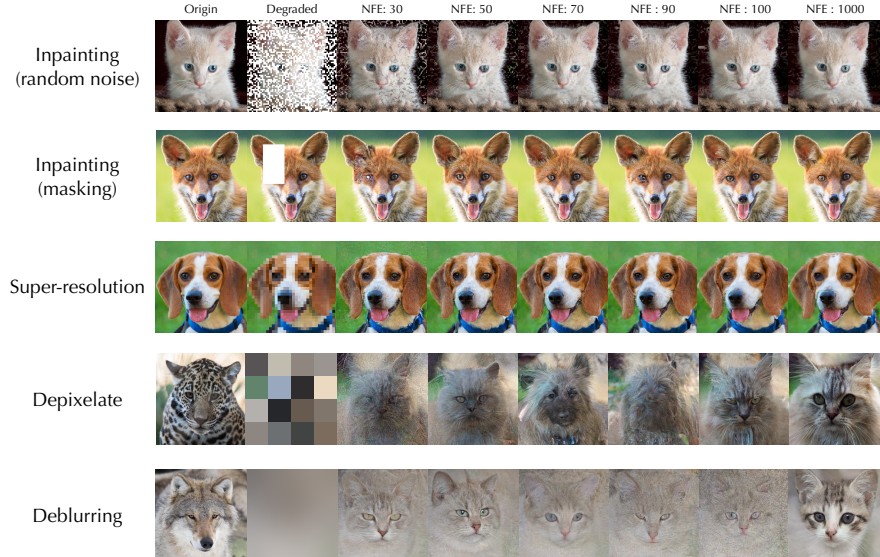

Figure A.4: Examples of Controllable Coarse-to-Fine Generation using HDM-SPDE.

Low-resolution images are made by downsampling $256 \times 256$ images to $32 \times 32$, then increasing the pixel size to create low-resolution images of size $256 \times 256$. In the method we use for inpainting on frequency size, the portion of low frequencies is filled with the frequencies of low-resolution images, and the portion of high frequencies is filled with the high-resolution image in the frequency domain. HDM-SPDE is also able to generate samples from blurred or pixelated images, where pixelate refers to the deduction of image details by replacing individual pixels with larger, rectangular pixels. The method of depixelate is similar to super-resolution in that it also fills out high-frequencies conditioning on low-frequencies.

Here is a code for inpainting and super-resolution including depixelate.

```
def inpainted_noise(data, noise, mask, t):
    e = torch.randn(x.shape).to(device)
    n = sde.marginal_std(e,t)
    x_mean = diffusion_coeff(data,t)
    return (x_mean + n) * (mask) + noise * (1-mask)

def sde_score_update_imputation(data, mask, x, s, t):
    score_s = inverse_marginal_std(model(x,s) + 1e-5, s)
```

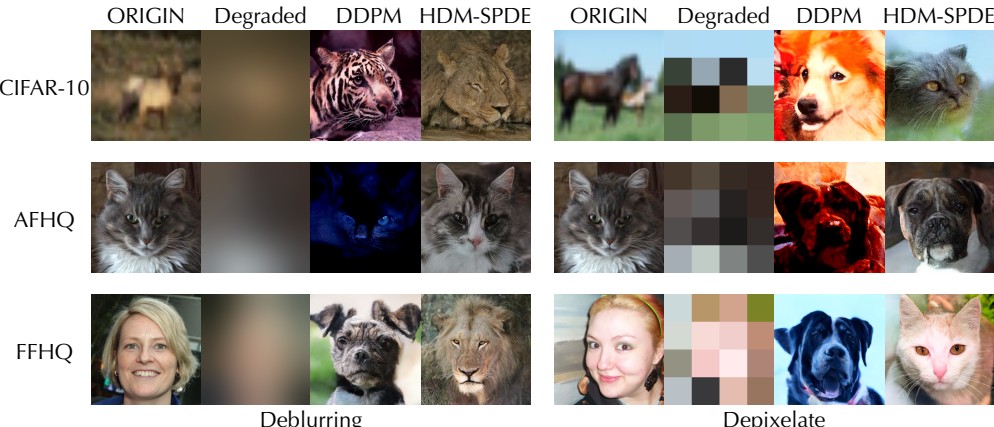

Figure A.5: Comparison of deblurring and depixelate on CIFAR-10, AFHQ, and FFHQ between DDPM (VP-SDE) [52] and HDM-SPDE trained on AFHQ.

```
9       e = torch.randn(x.shape)
10      x_t = -beta(s)[:,None,None,None] * laplacian(x,t,s) / 2+x
11      x_t += -beta(s)[:,None,None,None] * score_s * (t-s)[:,None,None,
        None]
12      x_t += torch.sqrt(beta(s))[:,None,None,None]*marginal_laplacian(e,
        t,s)
13      return inpainted_noise(data, x_t, mask, t)
```

Listing 9: Inpainitng

```
1   def create_circular_mask(h, w, center, radius):
2       Y, X = np.ogrid[:h, :w]
3       distance = np.sqrt((X - center[0])**2 + (Y-center[1])**2)
4       mask = (distance <= radius)
5       return mask
6
7   def forward_process(data,t):
8       e = torch.randn(x.shape)
9       noise = marginal_std(e, t)
10      x_mean = diffusion_coeff(data, t)
11      return x_mean + noise
12
13  def sde_score_update_super_resolution_or_depixelate(data, mask, x, s,
        t, muliplier):
14      score_s = inverse_marginal_std(score_model(x,s)+1e-5,s)
15      e= torch.randn(x.shape)
16      x_t = -beta(s)[:,None,None,None]*laplacian(x,t,s)/2+x
17      x_t += -beta(s)[:,None,None,None]*score_s*(t-s)[:,None,None,None]
18      x_t += torch.sqrt(beta(s))[:,None,None,None]*marginal_laplacian(e,
        t,s)
19      b,c,h,w = data.shape
20      data = forward_process(data,t)
21      x = dct_2d(x_t, norm='ortho')
22      mask = create_circular_mask(h,h,(0,0),int(h/multiplier))
23      y = dct_2d(data, norm='ortho')
24      a1 = x * (~mask)
25      a2 = y * mask
26      x = idct_2d(a1+a2, norm='ortho')
27      return x
```

Listing 10: Super-resolution and Depixelate

# E    Miscellaneous

HDM-SPDE can generate samples that resemble the background color to some extent for pixelated images and also generates samples following the contours of blurred images, while DDPM fails without additional training. Our deblurring or depixelate can be applied not only to AFHQ images but also to images such as MNIST or CIFAR10. Here are examples of deblurring and depixelate applied on MNIST, CIFAR-10, AFHQ and FFHQ using HDM-SPDE trained on AFHQ dataset.

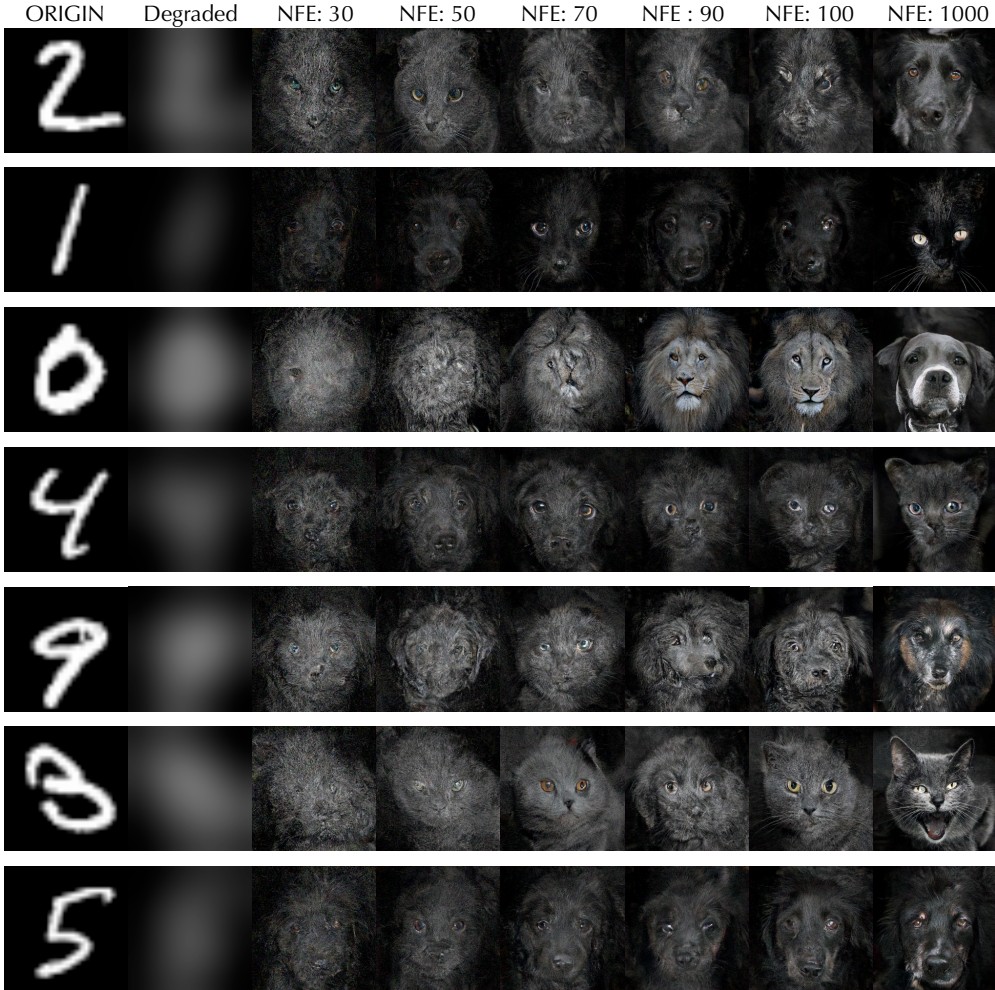

Figure A.6: Examples of deblurring on MNIST using HDM-SPDE trained on AFHQ

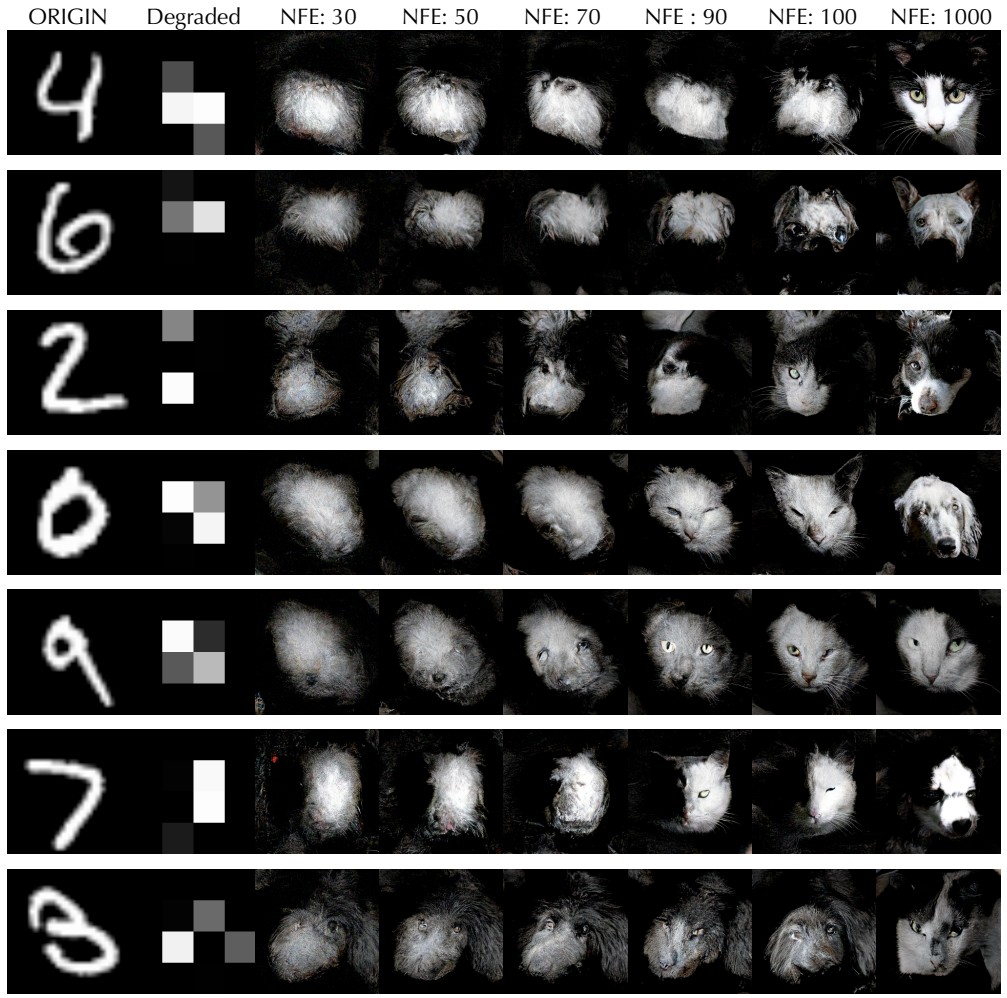

Figure A.7: Examples of depixelate on MNIST using HDM-SPDE trained on AFHQ

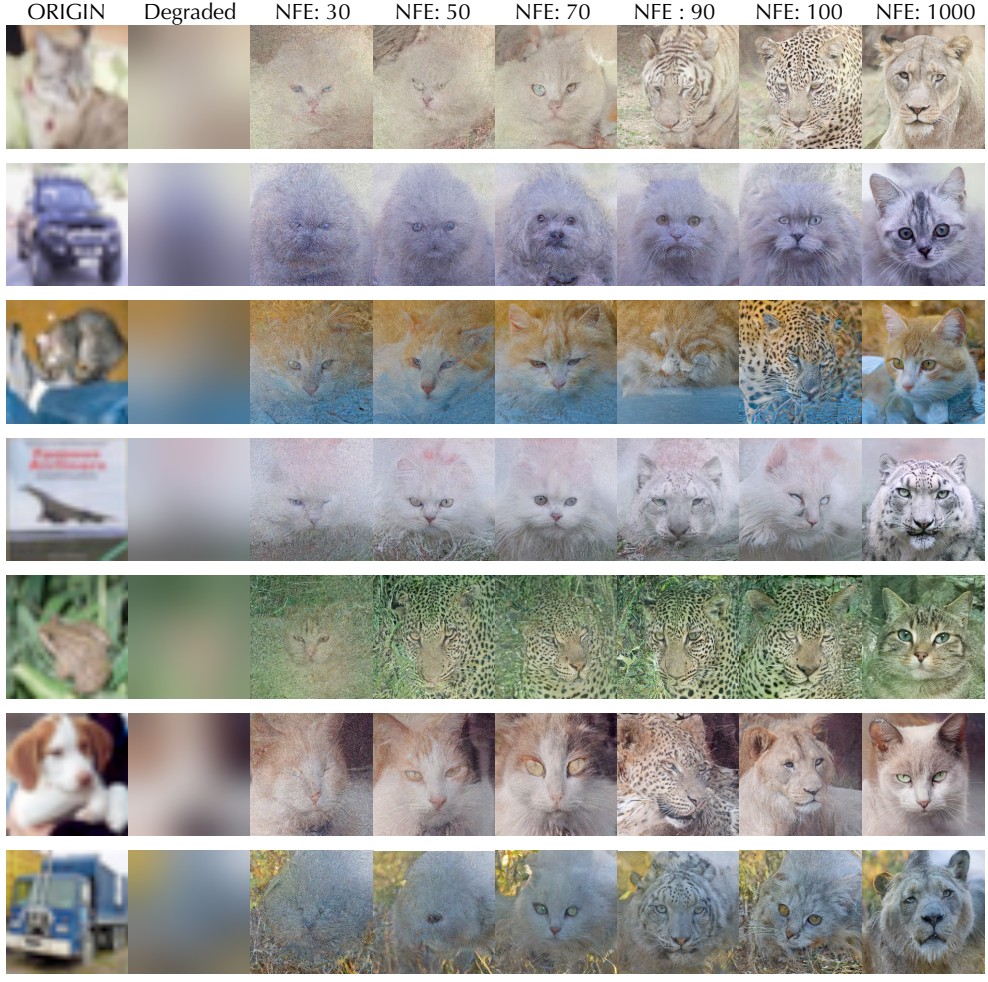

Figure A.8: Examples of deblurring on CIFAR10 using HDM-SPDE trained on AFHQ

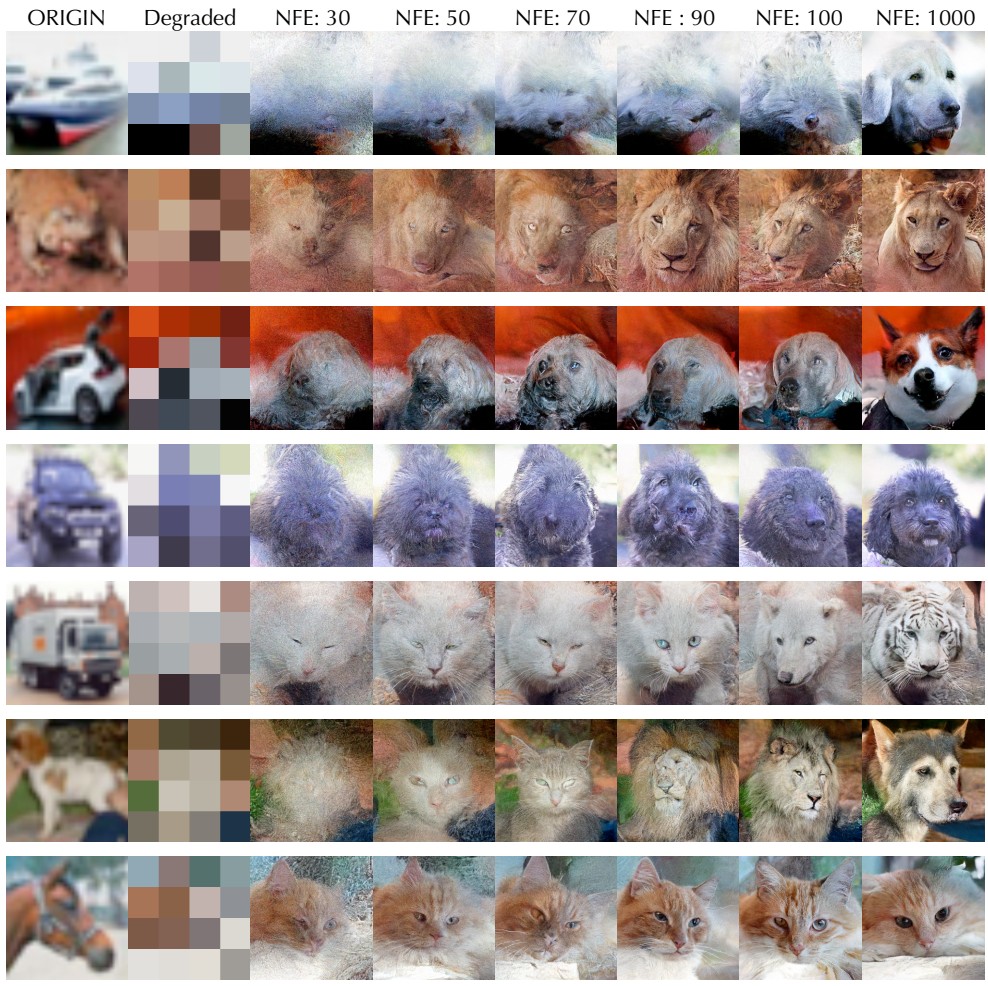

Figure A.9: Examples of depixelate on CIFAR10 using HDM-SPDE trained on AFHQ

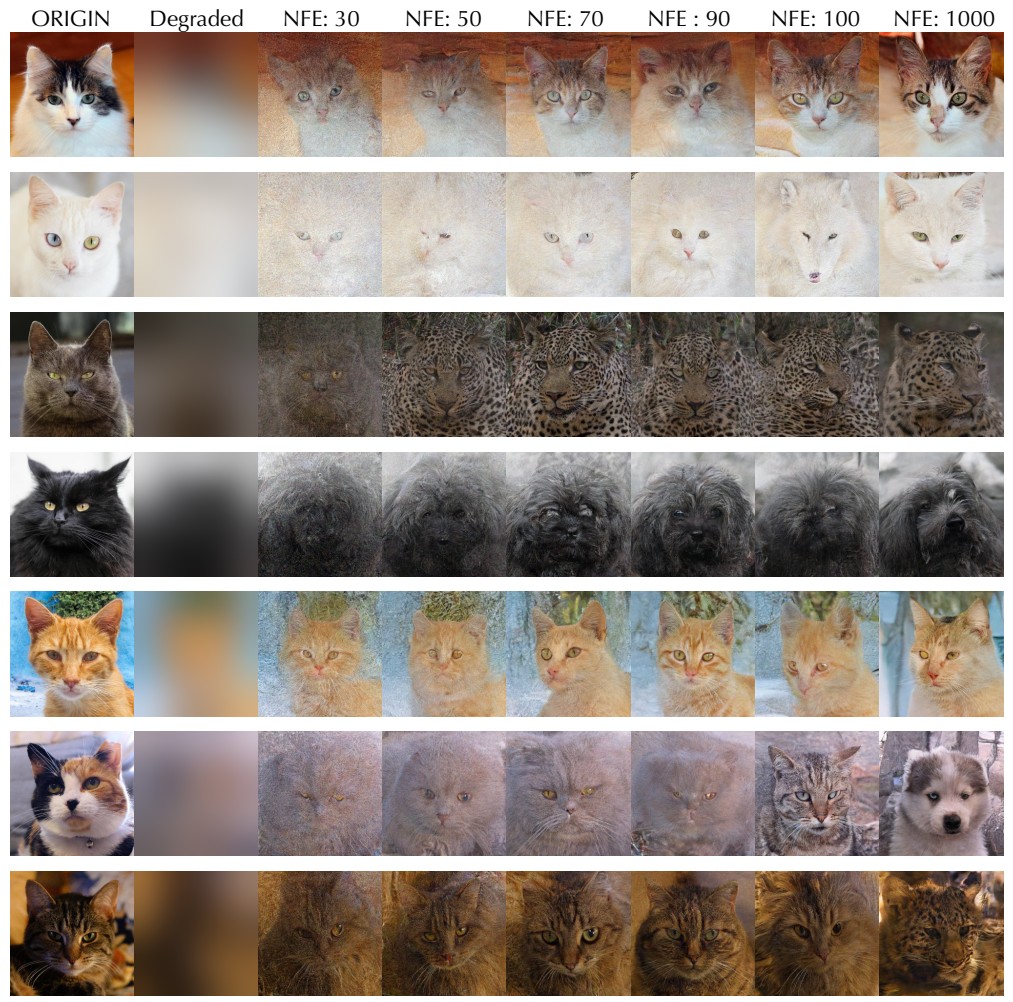

Figure A.10: Examples of deblurring on AFHQ using HDM-SPDE trained on AFHQ

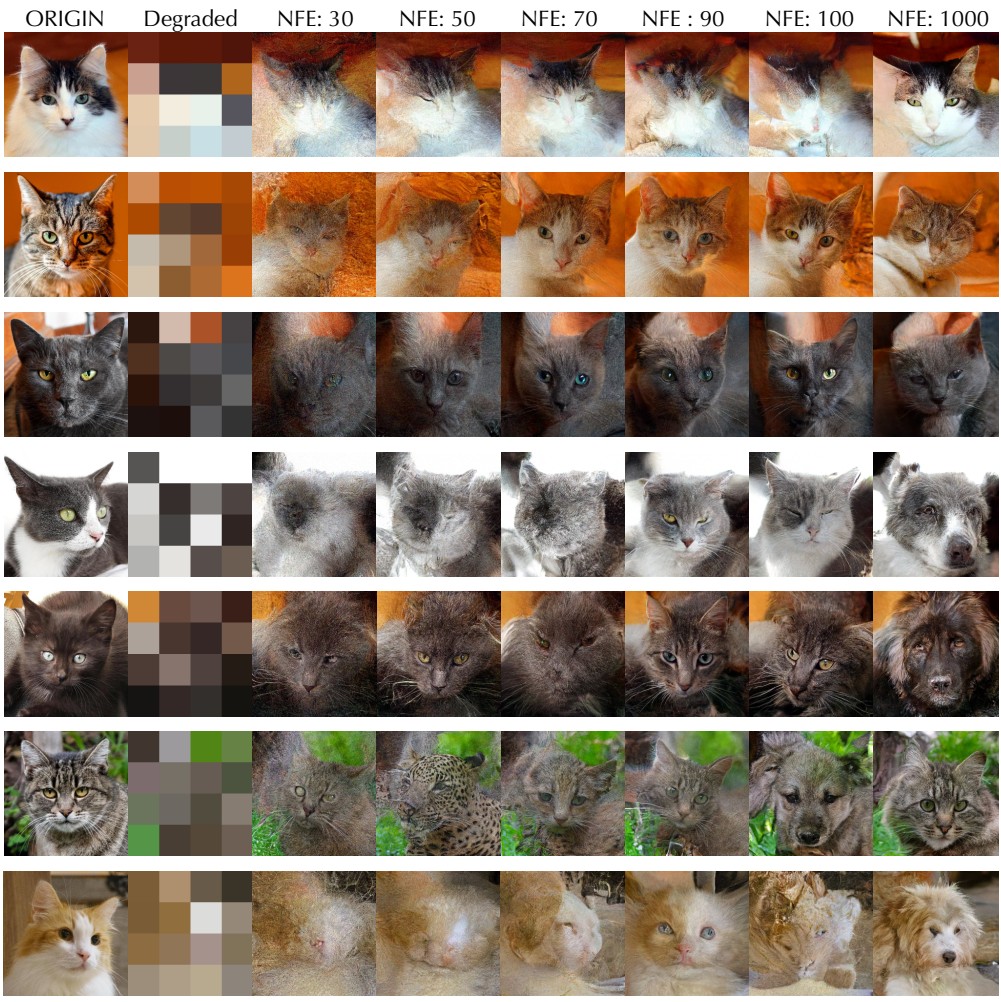

Figure A.11: Examples of depixelate on AFHQ using HDM-SPDE trained on AFHQ

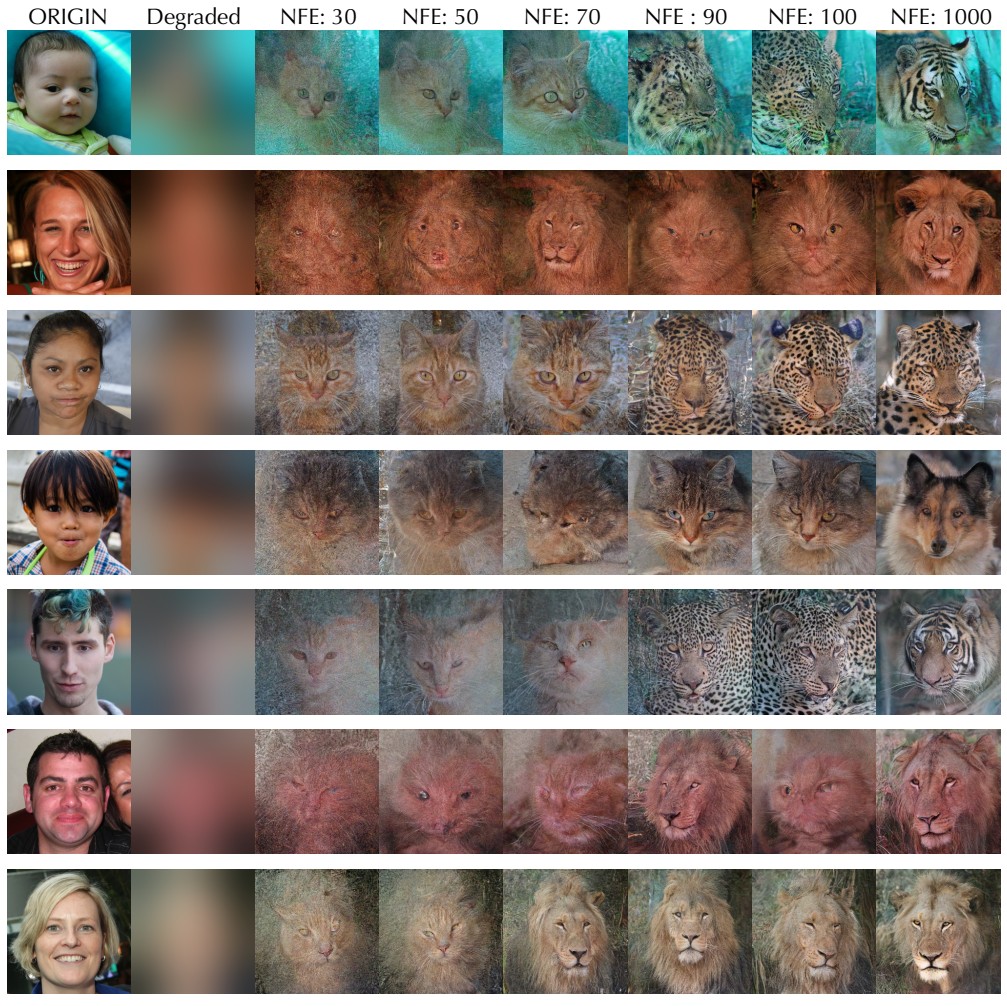

Figure A.12: Examples of deblurring on FFHQ using HDM-SPDE trained on AFHQ

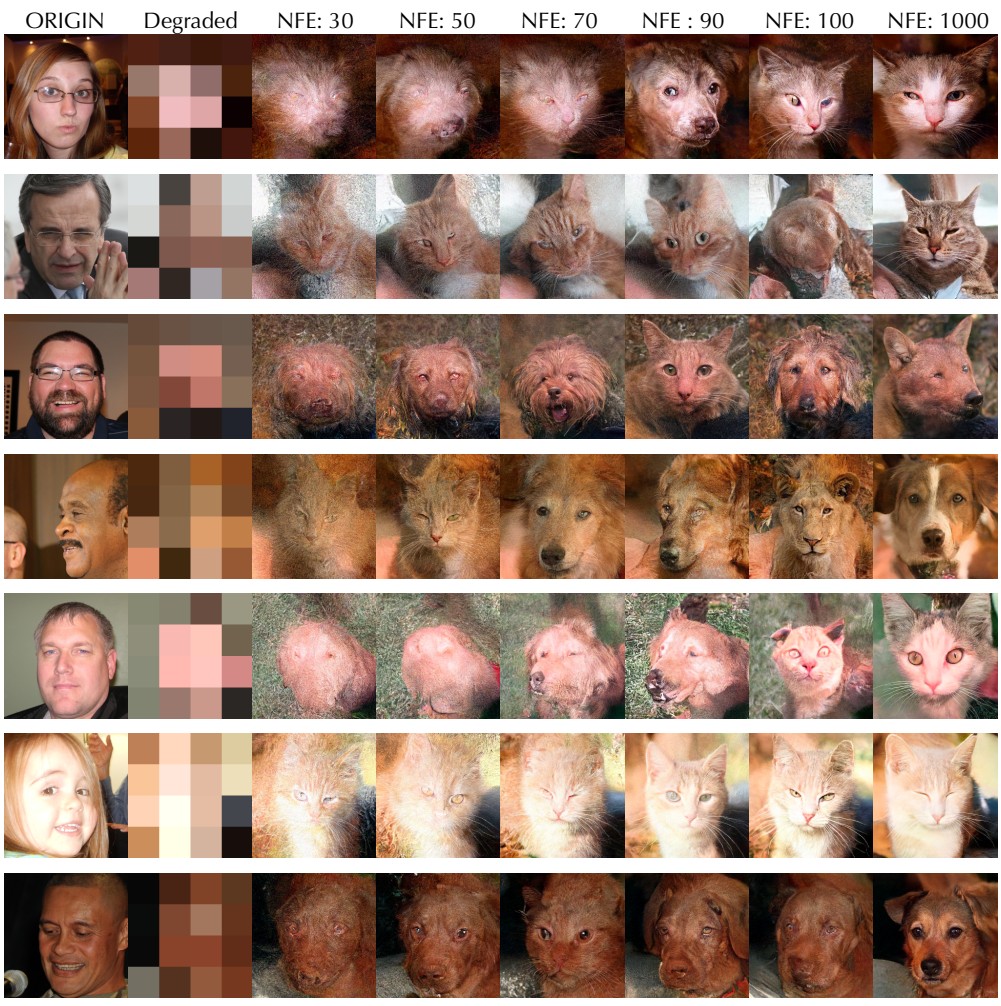

Figure A.13: Examples of depixelate on FFHQ using HDM-SPDE trained on AFHQ