# OpenReview forum: "Score-based Generative Modeling through Stochastic Evolution Equations in Hilbert Spaces"
_NeurIPS.cc/2023/Conference — NeurIPS 2023 spotlight_

### Official Review · Reviewer_8erp · 2023-07-05

**Soundness:** 3 good
**Presentation:** 3 good
**Contribution:** 2 fair
**Rating:** 7
**Confidence:** 3

**Summary:**

In this paper, the authors introduce Hilbert Diffusion Model (HDM). This generative model is a generalization of diffusion models to infinite dimensional state space (in this case Hilbert spaces) through the use of generalized SDEs. The authors extend the notion of score and time-reversal to this setting. Their framework can be applied to infinite dimensional diffusion models and to Stochastic Partial Differential Equations (SPDEs). They propose to use Fourier Neural Operator (FNO) as a basis of the architecture of the neural network to learn the score, similar to other infinite dimensional diffusion models. They showcase the efficiency of their model on function and image generation tasks.

**Strengths:**

* The paper presents a strong theoretical framework for the theory of Hilbert diffusion models. The ideas are very well-presented (even though I think the authors could give more details about their SPDE application, see my Questions).

* The experiments are convincing. In particular, the model seems to perform better than other models which are not based on a noising perturbation but on other degradations such as [1].

[1] Rissanen et al. (2023) -- Generative modelling with inverse heat dissipation

**Weaknesses:**

* One of my main remark is that there is little discussion on the relation between the current work and the existing literature on diffusion models. For instance, what is the novelty in the derived theorem 2.1? It seems to me that [1, Section 3.4] already studied the time-reversal of infinite diffusion models. Similarly, the framework of Cameron-Martin theory to derive infinite dimensional Brownian motion and Gaussian measures was already considered in [1,2]. What is the difference between the current work and the works [1,2,4,6] from a theoretical point of view? From a practical point of view, the authors should also specify how different their method is from existing works. As they point out the Fourier Neural Operator (FNO) was already used in several works [2,4,5]. I think the authors omit to discuss [3] which not only use the FNO but found that Galerkin operators worked well. At the end of the day what are the practical differences with [2,4,5]? Finally, could the authors discuss the relationship between their model and other infinite dimensional approaches such as [7,8]?

* My second remark is more general and has to do with the motivation of the work. It is not clear to me why, from a practical perspective, would anyone want to use infinite dimensional diffusion models? When it comes to implementation the space is going to be discretised and advantages that are usually cited to be given by the infinite dimensional setting (infinite resolution) can also be incoporated in existing finite dimensional diffusion models (by interpolating on a sub-pixel grid for instance). For example the fact that we consider continuous-time diffusion models could also be criticized since everything is then discretized at implementation time but it turns out that this formulation yields some advantages (gives ideas for better samplers, allows for simple integration of the Ornstein-Uhlenbeck to sample without Markov chains, allows for log-likelihood computation using ideas from continuous normalizing flows). Could the authors present such motivation in the case of infinite dimensional diffusion model? I think coming up with an example where infinite dimensional diffusion models are needed would be key to motivate the study.

* My third remark concerns the motivation behind other choices of perturbation than a Gaussian noising. Could the authors explain why not using a Gaussian noising might be beneficial? While it is possible to do so I have seen little practical justification of this so far.

[1] Pidstrigach et al. (2023) -- Infinite-dimensional diffusion models for function space

[2] Kerrigan et al. (2022) -- Diffusion generative models in infinite dimensions

[3] Bond-Taylor and Willcocks (2023) -- Infinite Diff: Infinite Resolution Diffusion with Subsampled Mollified States

[4] Lim et al. (2023) -- Score-based diffusion models in function space

[5] Hagemann et al. (2023) -- Multilevel diffusion: Infinite dimensional score-based diffusion models for image generation

[6] Franzese et al. (2023) -- Continuous-time functional diffusion processes

[7] Dutordoir et al. (2022) -- Neural Diffusion Processes

[8] Philips et al. (2022) -- Spectral Diffusion Processes

**Questions:**

* In Section 2.1 can the authors define what is a Gaussian measure in a Hilbert space

* We define Cameron-Martin space --> We define "the" Cameron-Martin space (?)

* In Section 2.1 it seems that there is no need for the operator $\Lambda$ to be positive. However, the authors take its inverse. Am I missing something here? In Example 2.1 what happens if $\Sigma$ is not invertible?

* The authors state "the time-reversal process is well-studied for diffusion processes in abstract settings" and cite [1]. Then, could the authors simply use the results from [1] to establish Theorem 2.1? [1] uses only the carre du champs operator and so, since the authors also describe things in terms of infinitesimal generator, it seems to me that their main result could be applied. Again, I might be missing something.

* For parabolic SPDE what is the motivation behind Equation (21)? It is not clear to me what is the role of the power of $k$ on the terms in front of the white noise Brownian motion.  Also in Equation (22) it seems that $1+k/2$ has turned into $k/2$. In Equation (23) where is the score? I think the authors should give more explanations in that section.

* The losses in Equation (25) and Equation (26) could be improved by using weighting terms or using $v$-prediction [2]. Have the authors considered this?

* It is not clear to me why the authors do not choose a FNO for the SPDE application, considering that the Laplacian is diagonalized in a Fourier basis.

* I am bit confused by the experiments on HumanML3D. DDPM seems to be performing extremely poorly in that setting. I am quite surprised by that and especially the discrepancy with the model proposed by the authors. Is there any good reason for that?

* I don't understand the "Controllable Coarse-to-FIne Generation" paragraph. How do authors solve inverse problems with a model trained for generative modeling? Is there a guidance function somewhere? Are the degraded images passed as input to the diffusion model where the forward has been chosen appropriately? More details are needed.

[1] Cattiaux et al. (2021) -- Time reversal of difusion processes under finite entropy condition

[2] Salimans et al. (2022) -- Progressive Distillation for fast sampling of diffusion models

**Limitations:**

Adressed in the conclusion.

---

> ### Author Rebuttal · Authors · 2023-08-09
>
> First of all, [Bond-Taylor & Willcocks, 2023] is a paper we had not cited at the time of submission, and we are very grateful to the reviewer for pointing it out. For the first main remark, Please see the **Author Rebuttal by Authors** for more details.
>
> For the second remark, real applications can be proposed in two main scopes in the context of infinite-dimensional tasks. Firstly, motivated from a Functional Data Analysis (FDA), proposing a generative model in function space is valid from a sample efficiency perspective. Suppose the functional sample should be smooth or periodic. In that case, we can imagine a probability measure defined on the corresponding function spaces with desired properties. As suggested in Section 4.1, the proposed method demonstrates better sampling efficiency from functional data spaces with multi-modality. The reviewer also can see this in the motion generation task requiring smooth trajectory sampling in higher dimensional settings, as shown in Section 4.2.
>
> Also, we promise to fix some errata below.
>
> > ***In Section 2.1 it seems that there is no need for the operator to be positive. However, the authors take its inverse.***
>
> Thanks for pointing that out. I think it was accidentally omitted due to space limitations. The theory presented in this paper assumes non-negativity, but Example 2.1 requires to be positive-definite. We will fix it in the revised manuscript.
>
> > ***The authors state the time-reversal process is well-studied for diffusion processes in abstract settings and cite [1]. Then, could the authors simply use the results from [1] to establish Theorem 2.1?***
>
> Thanks for the wonderful question. In fact, it may be possible to partially derive the result of Theorem 2.1 by applying Theorem 3.17 in [Cattiaux et al., 2021], since their approach is also based on the Integration by Part (IbP) formula. However, to specifically complete the dual-drift formula for the time-dependent operator, we need to compute the exact score operator using the calculations in Appendix B.1. For time-independent operators, this derivation is relatively trivial, but for time-dependent operators, we need the Itō’s formula in Hilbert space. Therefore, we must discuss the functional derivative, which naturally leads to the proposed Kolmogorov operators. In this sense, we claim that Theorem 2.1 can be viewed as a generalization of [B.D.O Anderson, 1982] and [Haussmann & Pardoux, 1987] in finite-dimensional spaces to general Hilbert spaces.
>
> > ***For parabolic SPDE what is the motivation behind Equation (21)? It is not clear to me what is the role of the power of on the terms in front of the white noise Brownian motion.***
>
> Thanks for pointing out this crucial point. It seems that an errata occurred while fixing the formula due to a lack of space. First, the role of $k$ is the order of the Bessel potential operator that transforms the Wiener process into smooth function space (Sobolev space). For $d \geq 2$, parabolic SPDEs become an ill-posed problem without such an operator and suffer regularity problems [N. V. Krylov, 1999]. Please also see [Section 4.2, Lim et al. 2023].
>
> We need to modify eq. (22) as follows:
>
> $$ \text{d}\hat{u}_t = -b(t)\left(\sigma\mathbf{I}-\mathbf{D}\right)\hat{u}_t \text{d}t+\sqrt{2b(t)}(\sigma\mathbf{I}-\mathbf{D})^{-\frac{k}{2}}\text{d}\widehat{\mathcal{W}}_t $$
>
> Therefore, eq. (23) also needs to be modified by adding the vector logarithmic derivative. The derivation of the exact formula is derived in Appendix C.3 (some minor errata therein, and we will fix them in the camera ready-version). Again, we deeply appreciate for pointing errata.
>
> > ***The losses in Equation (25) and Equation (26) could be improved by using weighting terms or using $v$-prediction [2]. Have the authors considered this?***
> >
>
> Thanks for the suggestion. We did not use $v$-prediction nor weighting terms in the experiments to compare with other approaches in functional generation. It is expected to bring performance gain in the image generation task compared with IHDM [Rissanen et al., 2023].
>
> > ***It is not clear to me why the authors do not choose a FNO for the SPDE application, considering that the Laplacian is diagonalized in a Fourier basis.***
> >
>
> Although HDM-SPDE experiments with image generation with the U-Net architecture for comparison with IHDM in a finite-dimensional setting, as reviewer mentioned, it can be converted to an infinite-dimensional setting using FNO architectures, similar to HDM-SDE. This direction is planned to be proposed as future research.
>
> > ***I am bit confused by the experiments on HumanML3D. Is there any good reason for that?***
> >
>
> One of the main advantages of the proposed method is its ability to determine the solution space of stochastic equations driven by a Hilbert space-valued Wiener process formulated by a specific kernel function, which can contain the characteristics of target functionals such as smoothness or curvature information by optimizing the kernel function from the training data, e.g., a model selection method for a Gaussian process, to benefit from sampling efficiency. On the contrary, white-noise DDPM is based on white noise which ignores correlated information, which may deteriorate the sample efficiency when modeling smooth functions.
>
> > ***I don't understand the "Controllable Coarse-to-FIne Generation" paragraph. More details are needed.***
> >
>
> Indeed, we have explained the parts of the coarse-to-fine generation in Appendix D.4. To solve the inverse problems, we apply the inpainting technique described in Appendix I.2 of (Song et al., ICLR 2021] to restore masked parts of the given image using the pre-trained score model without a guidance function. We will supplement more details in the revised manuscript.
>
> ---
>
> [Cattiaux et al., 2021], Time reversal of diffusion processes under finite entropy condition, 2021.
>
> [N. V. Krylov, 1999], An analytic approach to SPDEs, 1999.

---

> > ### Comment · Reviewer_8erp · 2023-08-12
> > **Updated score**
> >
> > I thank the authors for their detailed rebuttal. I am satisfied with most of the answers provided and have upgraded my score.
> >
> > I still have a few questions left.
> >
> > > largely grounded at theory of linear semigroups [Da Prato & Zabczyk, 2014], which is applicable to a wide range of differential equations but has a limitation when dealing with variable coefficients.
> >
> > What is the intrisic limitation of this approach when dealing with variable coefficients ?
> >
> > > Next, the works of [Lim et al., 2023; Pidstrigach et al., 2023] focus on continuous-time models, contrary to [Kerrigan et al., 2023]. Both methods and our approach start on a standard theoretical setting, utilizing Cameron-Martin space and Gaussian measure theory on Hilbert spaces. It is crucial to recognize, however, that the theoretical results in [Lim et al., 2023; Pidstrigach et al., 2023] are confined to SDEs with constant-time coefficients [...]
> >
> > So to summarize, the main theoretical contribution of this work is to extend [1] to the continuous-time setting. It seems that compared to other works such as [2,3,4] the main advantage of the current work is to be able to deal with the case of time-dependent coefficients, is that correct? I think it would be really beneficial to clarify this in the revised version of the paper.
> >
> > > On the contrary, our approach is grounded in the study of variational approach to stochastic evolution equations [Krylov & Rozovskii, 1981, 2007] and forward and backward Kolmogorov equations with functional derivatives [Dalecky & Fomin, 1991; Belopolskaya & Dalecky, 2012], which serves as a more established approach to identifying relations between time-dependent evolution operators and invariant measures of stochastic equations in Hilbert spaces [Y. Belopolskaya, 2020].
> >
> > Similarly to my previous comment it is not clear to me why that framework is "more established [...] to identifying relations between time-dependent evolution operators". Is there something specific to the stochastic evolution approach justifying this increased flexibility? At the end of the day what framework should be used if one wants to analyse infinite-dimensional SDE?
> >
> > > I am bit confused by the experiments on HumanML3D. DDPM seems to be performing extremely poorly in that setting. I am quite surprised by that and especially the discrepancy with the model proposed by the authors. Is there any good reason for that?
> >
> > I have read the rebuttal of the authors for this question but I am still surprised by how badly the DDPM model performs.
> > It clearly fails to learn something meaningful and the samples generated are not even close to be in the data distribution.
> > While I would expect the HDM to train faster and perform better, I would not expect the DDPM to fail completely. For me this reveals that maybe the DDPM was not trained for long enough or that maybe the chosen FNO architecture is adversarial when considering white noise.
> >
> > [1] Kerrigan et al. (2022) -- Diffusion generative models in infinite dimensions
> >
> > [2] Lim et al. (2023) -- Score-based diffusion models in function space
> >
> > [3] Pidstrigach et al. (2023) -- Infinite-dimensional diffusion models for function space
> >
> > [4] Franzese et al. (2023) -- Continuous-time functional diffusion processes

---

> > > ### Author Response · Authors · 2023-08-17
> > > **Re: Updated score**
> > >
> > > We sincerely appreciate reviewer 8erp for additional feedback and for updating the score. Please see below for answers to additional questions.
> > >
> > > > ***What is the intrisic limitation of this approach when dealing with variable coefficients ?***
> > >
> > > To apply the semigroup theory [Da Prato & Zabczyk, 2014], it needs to have a linear operator in the drift part, which has to generate a $C_0$-semigroup, which is not required in the variational approach [Liu & Röckner, 2010]. For instance, due to [Appendix A, Da Prato & Zabczyk, 2014], it is necessary to assume the condition, $S(t)S(s) = S(t+s)$. This condition holds when the leading operator is time-constant $\mathbf{A}$ so that $S(t)=\exp(\mathbf{A} t)$. On the other hand, this condition hardly handle general time-dependent operators $\mathbf{B}_{t}$ and $S(t) = \exp\left(\int_0^t \mathbf{B}_r \text{d}r\right)$ as we used, because
> > > $$
> > > S(t)S(s)=\exp\left(\int_0^t \mathbf{B}_r\text{d}r + \int_0^s \mathbf{B}_r\text{d}r\right) \neq \exp\left(\int_0^{t+s} \mathbf{B}_r\text{d}r\right)=S(t+s)
> > > $$
> > > We expect that the discussion above addresses the reviewer's question.
> > >
> > > [Liu & Röckner] SPDE in Hilbert space with locally monotone coefficients. *Journal of Functional Analysis*, 2010.
> > >
> > > > ***So to summarize, the main theoretical contribution of this work is to extend [1] to the continuous-time setting. It seems that compared to other works such as [2,3,4] the main advantage of the current work is to be able to deal with the case of time-dependent coefficients, is that correct?***
> > > >
> > >
> > > Yes, you are correct. We promise that it will be clarified in the revised manuscript.
> > >
> > > > ***Similarly to my previous comment it is not clear to me why that framework is "more established [...] to identifying relations between time-dependent evolution operators". Is there something specific to the stochastic evolution approach justifying this increased flexibility? At the end of the day what framework should be used if one wants to analyse infinite-dimensional SDE?***
> > > >
> > >
> > > Historically, the variational approach is a technique developed by researchers to overcome the theoretical limitations of the semigroup approach [Liu & Röckner, 2010]. Stochastic equation researchers have developed various advanced analytic tools that can be used in Hilbert spaces to create breakthroughs (Kolmogorov equations in Hilbert spaces [Belopolskaya & Dalecky, 2012], smooth cylinder functions [Bogachev et al., 2022], etc.). The proof proposed in this study is established using such analytic tools.
> > >
> > > From a mathematical perspective, the variational approach boasts numerous advantages; however, for the ML community, choosing a framework from a practical perspective is recommended. If the leading operator of SDE is time-constant, then the semigroup approach would be sufficient, as in previous studies. Yet, as we proposed, if the goal is to expand the SDE framework introduced by [Song et al., 2021] to incorporate Hilbert spaces, the variational method stands out as the preferable choice. This comes with a caveat: diving into this approach demands a grasp of the analytical tools in the Hilbert space, which may present challenges for a broader ML audience.
> > >
> > > [Belopolskaya & Dalecky, 2012] Stochastic equations and differential geometry, 2012.
> > >
> > > [Bogachev et al., 2022] Fokker–Planck–Kolmogorov Equations, *American Mathematical Society*, 2022.
> > >
> > > > ***I have read the rebuttal of the authors for this question but I am still surprised by how badly the DDPM model performs. It clearly fails to learn something meaningful and the samples generated are not even close to be in the data distribution. While I would expect the HDM to train faster and perform better, I would not expect the DDPM to fail completely. For me this reveals that maybe the DDPM was not trained for long enough or that maybe the chosen FNO architecture is adversarial when considering white noise.***
> > >
> > > Thanks for your suggestion. The DDPM fails to generate proper human motions even in the case of longer training. In particular, we trained the model five times longer (2K to 10K epochs). However, increasing the feature size of the FNO with DDPM was shown to be helpful. Specifically, we increased the feature size of the FNO with DDPM four times (256 to 1,024), and the generated human motions are reasonably okay, and the generated skeletons maintain human-like shapes.
> > >
> > > Nevertheless, its quality was inferior to the proposed HDM in that it shows more jittery movements and fails to capture diverse behaviors in the dataset. The quantitative results also show that it still underperforms the proposed method. We evaluate MMD (mean maximum discrepancy) for measuring the quality of the generated motion. The average MMD scores of DDPM decrease from 0.496 to 0.375 as we increase the feature size from 256 to 1,024. However, it is still worse than the MMD of the proposed method (HDM), 0.055, with a smaller feature size of 256.
> > >
> > > We will supplement the above results in the revised manuscript.

---

### Official Review · Reviewer_gdSN · 2023-07-05

**Soundness:** 4 excellent
**Presentation:** 3 good
**Contribution:** 3 good
**Rating:** 7
**Confidence:** 4

**Summary:**

The authors propose a framework called the Hilbert Diffusion Model (HDM), which generalizes and unifies the time-reversal diffusion generative models with stochastic differential equations (SDEs) in Hilbert spaces. This model enables SDEs to be used in infinite-dimensional settings and to include stochastic partial differential equations (SPDEs) in Euclidean settings. The HDM's performance is verified against several datasets, outperforming existing diffusion models formulated in function spaces. It's particularly successful in human motion synthesis, where it applies the Wiener process in the Hilbert space.

**Strengths:**

This paper effectively aims to expand the time-reversal diffusion generative model into Hilbert space. The authors also provide a comprehensive background and overview of Stochastic Differential Equations (SDEs) in Hilbert space, as illustrated in sections 2.1 and 2.2 before the time-reversal theorem.

The paper is very well-structured, featuring clear and logical derivations. The authors deserve commendation for their efforts in:

- Establishing the time-reversal diffusion property in Hilbert space (as shown in Theorem 2.1 and Appendix B.2) through the use of infinitesimal generators;
- Introducing and unifying the time-reversal diffusion in Hilbert space (as seen in Appendix C.2);
- Employing parabolic SPDEs as functional examples and presenting the connection between parabolic SPDEs and the stochastic process used in IHDM (refer to Section 2.3.2).

The mathematical rigor demonstrated throughout the paper is thorough and cohesive.

**Weaknesses:**

Major Points:

- This work is not the first to extend time-reversal diffusion models to infinite-dimensional space, and the concept of time-reversal diffusion on Hilbert space is already addressed in [1][2][3]. As these are mentioned in your citations, it would be beneficial to delve into the relationship between HDM and these models. For instance, the authors might discuss whether they are particular cases of HDM. This further elaboration could help clarify how HDM contributes to the field beyond what has been covered in these previous works.

- It would be advantageous to incorporate an evaluation comparison with [1][2][3], given their direct relevance. I would be interested in seeing, for example, how your model compares to [2] using the AEMET dataset[2] or how their model performs on the datasets you're presenting.

Minor Points:

- For the convenience of readers, it would be more effective if the authors could present the experimental results of image generation in tables instead of embedding them in the text.

- In terms of image generation, the performance of SPDE does not match up to DDPM or VE-SDE. However, since this paper primarily concentrates on generation in Hilbert space - an issue already noted in the limitations - it may not be necessary to highlight this discrepancy in this context.


[1] Pidstrigach, J., Marzouk, Y., Reich, S., & Wang, S. (2023). Infinite-dimensional diffusion models for function spaces. arXiv preprint arXiv:2302.10130.

[2] Kerrigan, G., Ley, J., & Smyth, P. (2022). Diffusion generative models in infinite dimensions. arXiv preprint arXiv:2212.00886.

[3] Lim, J. H., Kovachki, N. B., Baptista, R., Beckham, C., Azizzadenesheli, K., Kossaifi, J., ... & Anandkumar, A. (2023). Score-based diffusion models in function space. arXiv preprint arXiv:2302.07400.

**Questions:**

See weaknesses.

**Limitations:**

The authors already discussed the current limitations of their model (e.g., limited performance on image generation).

---

> ### Author Rebuttal · Authors · 2023-08-08
>
> We sincerely appreciate the reviewer’s valuable feedback.
>
> > ***This work is not the first to extend time-reversal diffusion models to infinite-dimensional space, and the concept of time-reversal diffusion on Hilbert space is already addressed in [1][2][3]. As these are mentioned in your citations, it would be beneficial to delve into the relationship between HDM and these models. For instance, the authors might discuss whether they are particular cases of HDM. This further elaboration could help clarify how HDM contributes to the field beyond what has been covered in these previous works.***
>
> We appreciate the reviewer's suggestion. There is a theoretical difference between the proposed method and the existing methodologies [Kerrigan et al., 2023, Lim et al., 2023, Pidstrigach et al., 2023]. We promise to elaborate on it in more detail. Please see the **Author Rebuttal by Authors** for more details. We emphasize that the proof of Theorem 2.1 (Appendix B.2) can be viewed as a generalization of the classical techniques employed by [B.D.O Anderson, 1982] and [Haussmann & Pardoux, 1987] in finite-dimensional spaces to general Hilbert spaces; consequently it is able to fully recover the SDE framework proposed from [Song et al., 2021].
>
> > ***It would be advantageous to incorporate an evaluation comparison with [1][2][3], given their direct relevance. I would be interested in seeing, for example, how your model compares to [2] using the AEMET dataset[2] or how their model performs on the datasets you're presenting.***
>
> Thanks for your suggestion. We experiment with our method on the AEMET dataset presented in [Kerrigan et al., 2023] for the further comparison. For HDM-SDE, we use the SE kernel as presented in the main paper instead of the Mátern kernel used in [Kerrigan et al., 2023], while the rest of the hyperparameter settings are the same. The MSE performance of HDM-SDE is reported as follows:
>
> - [Kerrigan et al., 2023] (Best result): Mean: 0.0152, Variance: 1.0748, Autocorr: 2.551e-06
> - HDM-SDE: **Mean: 0.0001**, **Variance: 0.0293**, **Autocorr: 1.1272e-06**
>
> Also, the power(%) of two-sample test for HDM-SDE records a value of $0.48\pm0.95$, implying that the statistical difference between the generated and target datasets is negligible and hardly discernable. We believe these experimental results may compare an empirical difference between the proposed and existing methods [Kerrigan et al., 2023].
>
> In our analysis, we examined the performance of the trace-class Wiener process, specifically considering **constant-time coefficients** (i.e., $\mathbf{B}_t = I$ and $\mathbf{G}_t = \sqrt{2}I$, as discussed in [Section 4.1, Lim et al., 2023]), aligns closely with the results of SP-SGM [Phillips et al., 2022]. To further elucidate the comparison, we conducted additional 1D function generation experiments, utilizing the infinite-dimensional Langevin dynamics method as proposed by [Lim et al., 2023]. The power(%) of a kernel two-sample test on the Quadratic and Melbourne datasets are reported as follows:
>
> - Quadratic: $6.9 \pm 0.5$, Melbourne: $4.0 \pm 0.3$
>
> The above result is comparable to SP-SGM (Quadratic: $5.4 \pm 0.7$, Melbourne: $5.3 \pm 0.7$) but are generally less favorable than HDM-SDE (Quadratic: $4.2 \pm 0.3$, Melbourne: $0.2 \pm 0.1$). We claim that this discrepancy in performance is attributable to the scheduling strategy; hence our results underscore the significance of the dual drift formula presented in Theorem 2.1.
>
> However, it should be noted that the above experiment does not undervalue previous works since [Lim et al., 2023; Kerrigan et al., 2023] also present discrete-time models considering time scheduling, which recovers the DDPM framework [Ho et al., 2020]. A key difference between previous works and ours is that we consider a continuous-time model, which fully recover SDEs (VP-SDE and sub-VP-SDE) from [Song et al., 2021] and can utilize time-dependent coefficients, thereby enabling cosine-beta scheduling.
>
> > ***For the convenience of readers, it would be more effective if the authors could present the experimental results of image generation in tables instead of embedding them in the text.***
> >
>
> Thanks for the suggestion. We will try to make the changes in the revised manuscript.
>
> > ***In terms of image generation, the performance of SPDE does not match up to DDPM or VE-SDE. However, since this paper primarily concentrates on generation in Hilbert space - an issue already noted in the limitations - it may not be necessary to highlight this discrepancy in this context.***
> >
>
> As mentioned by the reviewer, there is room for improvement in the 2D image generation task, so we also noted in the limitation section. The proposed HDM-SPDE focuses on building a bridge between the discrete-time model presented in IHDM [Rissanen et al., 2023] and the continuous-time models, so it is different from the generative models used in DDPM [Ho et al., 2020], and VP-SDE [Song et al., 2023]. We will address these issues in future research.
>
> ---
>
> [B.D.O Anderson, 1982] Reverse time diffusion equation models. *Stochastic Processes and their Applications,* 1982.
>
> [Haussmann & Pardoux, 1987] Time reversal of diffusions. *Annals of Probability*, 1987.
>
> [Ho et al., 2020] Denoising diffusion probabilistic models. *NeurIPS*, 2020.
>
> [Kerrigan et al., 2023] Diffusion generative models in infinite dimensions. *AISTATS*, 2023.
>
> [Lim et al., 2023] Score based diffusion models in function space. *arXiv*, 2023.
>
> [Pidstrigach et al., 2023] Infinite-dimensional diffusion models for function spaces. *arXiv*, 2023.
>
> [Rissanen et al., 2023] Generative modelling with inverse heat dissipation. *ICLR*, 2023.
>
> [Song et al., 2021] Score-based generative modeling through stochastic differential equations. *ICLR*, 2021.

---

> > ### Comment · Reviewer_gdSN · 2023-08-21
> > **Reply to authors**
> >
> > Thank you for your rebuttal. In this case, I would still like to deem this as a more general framework that can unify previous works even considering time continuity.
> >
> > Also, thanks for your experiment updates, and I hope this result will appear in your revision. I update the score accordingly.

---

> ### Comment · Area_Chair_eqPS · 2023-08-20
> **Reminder from AC**
>
> Dear reviewer,
>
> The author-reviewer discussion period ends in 2 days. Please review the authors' rebuttal and engage with them if you have additional questions or feedback. Your input during the discussion period is valued and helps improve the paper.
>
> Thanks, Area Chair

---

### Official Review · Reviewer_yicF · 2023-07-06

**Soundness:** 4 excellent
**Presentation:** 3 good
**Contribution:** 3 good
**Rating:** 7
**Confidence:** 4

**Summary:**

This work studies continuous-time diffusion processes in Hilbert spaces, and their applications to generative modeling. In particular, a time-reversal formula is derived for Hilbert-space SDEs, allowing the continuous-time diffusion model framework to be generalized to infinite-dimensional Hilbert spaces. The proposed methodology is evaluated on several tasks, including generation of images and time-series.


**Strengths:**

- The exposition is clear and the paper is generally well-written throughout.
- This work adds to the growing literature on function-space diffusion models, and lays a solid theoretical foundation for future work in this area. The results are likely to be of significant interest to the community.
- Section 2.3 proposes a novel continuous-time generalization of the IHDM model (Riassanen et al., 2023) which shows strong empirical performance (Section 4) compared to the discrete-time model.
- The proposed method shows favorable performance on a functional generation task when compared to several existing functional baselines.
- To the best of my knowledge the proofs are correct, and the empirical claims are well-supported by the provided evidence.

**Weaknesses:**

- The paper could be better contextualized within the existing literature on this topic, i.e. there is no related work section. E.g. the authors briefly mention several closely related works on generating functional data (Line 30) but do not detail the differences between this submission and those existing works. See also the question below regarding time-reversals of infinite dimensional diffusions.

**Questions:**

- Time reversals of infinite-dimensional diffusions have been studied previously [1-2] -- how does the main result (Theorem 2.1) relate to these existing works?
- Could you elaborate on why you believe the DDPM model performed so poorly in Table 1, given that previous works [3-4] have successfully developed function-space generalizations of this model? E.g. it was unclear if a white-noise DDPM model was used or a trace-class DDPM model was used (as discussed  in [3-4]).

[1] Time reversal for infinite-dimensional diffusions. Annie Millet, David Nualart, Marta Sanz, 1989.

[2] Time reversal of infinite-dimensional diffusions. H. Follmer, A. Wakolbinger, 1986.

[3] Diffusion generative models in infinite dimensions. Gavin Kerrigan, Justin Ley, Padhraic Smyth, 2023.

[4] Score based diffusion models in function space. Jae Hyun Lim et al., 2023.

**Limitations:**

The limitations of the work are appropriately addressed within the paper.

---

> ### Author Rebuttal · Authors · 2023-08-08
>
> We greatly appreciate the reviewer’s constructive feedback.
>
> > ***The paper could be better contextualized within the existing literature on this topic, i.e. there is no related work section.***
>
> In fact, we compare our approach to the existing literature in Appendix C.1, but we do not have a corresponding section in the main text due to lack of space. We promise to elaborate on it in more detail in the revised manuscript.
>
> > ***Time reversals of infinite-dimensional diffusions have been studied previously [1-2] -- how does the main result (Theorem 2.1) relate to these existing works?***
>
> The work of [Follmer & Wakolbinger, 1986] considers a sequence of independent standard Wiener processes, utilizing an identity operator as the diffusion coefficient. This study represents a specific instance of the more general framework outlined in [Millet et al., 1989]. In Section 2.3.1 of our paper, we reference Theorem 5.3 from [Millet et al., 1989], which leverages the theory of stochastic calculus of variations to identify the drift operator (eq. 19) for SDEs in infinite dimensions. Additionally, in Appendix C.2, we demonstrate that this result can be viewed as a special case of our main result (eq. 16).
>
> It is also worth noting that although [Millet et al., 1989] offer a satisfactory time-reversal theory for SDEs in infinite dimensions, it does not readily lend itself to the creation of a unified framework capable of encompassing diffusion models based on stochastic PDEs (eq. 21) described in Section 2.3.2. In contrast, our approach is grounded in the study of variational approach to stochastic evolution equations [Krylov & Rozovskii, 1981, 2007] and forward and backward Kolmogorov equations with functional derivatives [Dalecky & Fomin, 1991; Belopolskaya & Dalecky, 2012], which serves as a more established approach to identifying relations between time-dependent evolution operators and invariant measures of stochastic equations in Hilbert spaces [Y. Belopolskaya, 2020]. It is noteworthy that proof of Theorem 2.1 (Appendix B.2) can be viewed as a generalization of the classical techniques employed by [B.D.O Anderson, 1982] and [Haussmann & Pardoux, 1987] in finite-dimensional spaces to general Hilbert spaces; consequently it is able to fully recover the SDE framework proposed from [Song et al., 2021].
>
> > ***Could you elaborate on why you believe the DDPM model performed so poorly in Table 1, given that previous works [3-4] have successfully developed function-space generalizations of this model? E.g. it was unclear if a white-noise DDPM model was used or a trace-class DDPM model was used (as discussed in [3-4]).***
>
> In our 1D function generation experiment, we utilized a white-noise DDPM [Ho et al., 2020], as shown in Table 1. White-noise DDPM models show limited adaptability to variable discretizations of the data as discussed in [Lim et al., 2023; Kerrigan et al., 2023]. By potentially ignoring correlated information, white-noise DDPM could lead to a deterioration in sample efficiency, particularly when modeling smooth functions. In this point of view, trace-class DDPM models [Lim et al., 2023; Kerrigan et al., 2023] probably show comparable results with HDM-SDE.
>
> In our analysis, we examined the performance of the trace-class Wiener process, specifically considering $\mathbf{B}_t = I, \mathbf{G}_t = \sqrt{2}I$ case, as discussed in [Section 4.1, Lim et al., 2023], aligns closely with the results of SP-SGM [Phillips et al., 2022]. To further elucidate the comparison, we conducted additional 1D function generation experiments, utilizing the infinite-dimensional Langevin dynamics method as proposed by [Lim et al., 2023]. The power(%) of a kernel two-sample test on the Quadratic and Melbourne datasets are reported as follows:
>
> - Quadratic: $6.9 \pm 0.5$, Melbourne: $4.0 \pm 0.3$
>
> The above result is comparable to SP-SGM (Quadratic: $5.4 \pm 0.7$, Melbourne: $5.3 \pm 0.7$) but are generally less favorable than HDM-SDE (Quadratic: $4.2 \pm 0.3$, Melbourne: $0.2 \pm 0.1$). We claim that this discrepancy in performance is attributable to the scheduling strategy; hence our results underscore the significance of the dual drift formula presented in Theorem 2.1.
>
> However, it should be noted that the above experiment does not undervalue previous works since [Lim et al., 2023; Kerrigan et al., 2023] also present **discrete-time models** considering time scheduling, which recovers the DDPM framework [Ho et al., 2020]. A key difference between previous works and ours is that we consider a **continuous-time model**, which fully recover SDEs (VP-SDE and sub-VP-SDE) from [Song et al., 2021] and can utilize time-dependent coefficients, thereby enabling cosine-beta scheduling.
>
> ---
> [B. D. O Anderson, 1982] Reverse time diffusion equation models. *Stochastic Processes and their Applications,* 1982.
>
> [Follmer & Wakolbinger, 1986] Time reversal of infinite-dimensional diffusions. *Stochastic Processes and their Applications*, 1986.
>
> [Haussmann & Pardoux, 1987] Time reversal of diffusions. *Annals of Probability*, 1987.
>
> [Ho et al., 2020] Denoising diffusion probabilistic models. *NeurIPS*, 2020.
>
> [Kerrigan et al., 2023] Diffusion generative models in infinite dimensions. *AISTATS*, 2023.
>
> [Lim et al., 2023] Score based diffusion models in function space. *arXiv*, 2023.
>
> [Millet et al., 1989] Time reversal for infinite-dimensional diffusions. *Probability theory and related fields,* 1989.
>
> [Phillips et al., 2022] Spectral diffusion processes. NeurIPS Workshop on SBM, 2022.
>
> [Song et al., 2021] Score-based generative modeling through stochastic differential equations. *ICLR*, 2021.
>
> [Dalecky & Fomin, 1991] Measures and differential equations in infinite-dimensional space, 1991.
>
> [Belopolskaya & Dalecky, 2012] Stochastic equations and differential geometry, 2012.
>
> [Krylov & Rozovskii, 1981, 2007] Stochastic evolution equations, 1981 (Russian) 2007 (English).

---

> > ### Comment · Reviewer_yicF · 2023-08-14
> >
> > Thank you for the detailed response and clarifications. The response sufficiently addresses my questions and the limitations I point out.
> > If the paper is accepted, I highly recommend using some of the additional allotted space for related work. I found the discussion regarding Theorem 2.1 in your response highly valuable and would be worth including in the paper itself.

---

> > > ### Author Response · Authors · 2023-08-17
> > > **Re: Official Comment by Reviewer yicF**
> > >
> > > We appreciate reviewer yicF for additional comments. We promise to allot the space for related works in the revised manuscript and contain the above discussions in detail.
> > >
> > > Thanks again for the considerate discussion!

---

### Official Review · Reviewer_sBaK · 2023-07-06

**Soundness:** 4 excellent
**Presentation:** 3 good
**Contribution:** 4 excellent
**Rating:** 7
**Confidence:** 3

**Summary:**

The paper proposes a time-dependent stochastic generative model that can be implemented in Hilbert spaces. The presented method unifies diffusion processes (commonly defined in diffusion models) with a more general stochastic evolution equation. A stochastic evolution equation is defined with satisfied conditions which mimics the framework of the forward process of the SDE equation derived for diffusion models. A time-reversal formula is derived from forward stochastic evolution equation based on the Kolmogorov equation and closely resembles the SDE equation of the score-based model.

Similar to score-based models, the score function is learned by a deep learning model with Fourier layers in the network. The model shows promising qualitative and quantitative results for unique tasks (e.g., motion generation, infinite dimension),

**Strengths:**

- The paper is well written/organized and easy to follow. The author does a good job introducing related theoretical information.
- The figures are mostly helpful for visualization.
- Equations are well structured, but it may be helpful if there is a brief background or insight on some claims (e.g., line 124 the reason behind choosing the class of smooth cylinder functions, Line 109 basic description of Fomin differentiable probability measures).
- In general, a very theoretically sound paper.


**Weaknesses:**

- It would be helpful if some of the datasets are briefly explained.
- Many of the figure descriptions are not very clear (e.g., Figure 1 (a) and I didn't quite understand (b), add title for Figure 3 images)
- It would be nice if some intuition or insight could be given on why the baseline DDPM model cannot perform well with some of these tasks.

**Questions:**

- Do you think this proposed method would naturally perform better than the DDPM baseline model for audio datasets due to the Fourier transform layers?
- What are some real world applications for infinite dimension tasks? Could these relate to Gigapixel images and multimodal datasets?

**Limitations:**

- The limitations were properly addressed.

---

> ### Author Rebuttal · Authors · 2023-08-08
>
> Thank you for your valuable feedback. In the camera-ready version, we will supplement additional explanations regarding the dataset, Figures 1 and 3, and the theoretical background such as smooth cylinder functions and Fomin differentiable measures, which were briefly discussed due to the space limit.
>
> > ***It would be nice if some intuition or insight could be given on why the baseline DDPM model cannot perform well with some of these tasks.***
> >
>
> One of the main advantages of the proposed method is its ability to determine the solution space of stochastic equations driven by a Hilbert space-valued Wiener process formulated by a specific kernel function, which can contain the characteristics of target functionals such as smoothness or curvature information by optimizing the kernel function from the training data, e.g., a model selection method for a Gaussian process, to benefit from sampling efficiency. On the contrary, DDPM is based on white noise which ignores correlated information, which may deteriorate the sample efficiency when modeling smooth functions. In particular, human motion is represented as a sequence of skeletons where a skeleton consists of a sequence of 3D rotation matrices (or, equivalently, a product of SO(3) groups) whose movement should be continuous and smooth, and this motivates us to propose a diffusion model on function spaces to synthesize human motions. Consequently, our motion generation results well examined our hypothesis in that our proposed method (HDM-SDE) successfully models the human motion distribution, whereas the baseline model fails. We would like to note that coefficient scheduling is also a critical component in our implementation, distinct from other diffusion models on function spaces.
>
> > ***Do you think this proposed method would naturally perform better than the DDPM baseline model for audio datasets due to the Fourier transform layers?***
> >
>
> Even though we did not try to experiment with our methodology in the audio dataset, we believe it could be applied for future research. As mentioned earlier, DDPM destroys correlated information in the signal space, requiring more data for training the score model. With the proposed method, HDM-SDE, correlated information can be preserved through Hilbert space-valued Wiener process and Fourier transform layers in Fourier Neural Operator (FNO). However, there have been some claims that FNO is limited to restoring high-frequency information, so appropriate architecture design will be necessary.
>
> > ***What are some real world applications for infinite dimension tasks? Could these relate to Gigapixel images and multimodal datasets?***
> >
>
> In the context of Infinite-dimensional tasks, real applications can be proposed in two main scopes. Firstly, as suggested in Section 4.1, it demonstrates the possibility of sampling from functional data spaces with multi-modality properties even though the proposed experiment is limited to 1D. Additionally, resolution-independent sampling is also possible since HDM-SDE utilizes the time-conditioned FNO. Secondly, for motion generation tasks requiring smooth trajectory sampling in higher dimensional settings, the proposed HDM-SDE model shows better performance than DDPM regarding sample efficiency, as shown in Section 4.2.
>
> Although HDM-SPDE experiments with image generation with the U-Net architecture for comparison with IHDM in a finite-dimensional setting, it can be converted to an infinite-dimensional setting using FNO architectures, similar to HDM-SDE. This direction is planned to be proposed as future research in subsequent studies.

---

> > ### Comment · Reviewer_sBaK · 2023-08-21
> >
> > Thank you for your detailed response, I am pleased with your answers. I believe this paper should be accepted.

---

> ### Comment · Area_Chair_eqPS · 2023-08-20
> **Reminder from AC**
>
> Dear reviewer,
>
> The author-reviewer discussion period ends in 2 days. Please review the authors' rebuttal and engage with them if you have additional questions or feedback. Your input during the discussion period is valued and helps improve the paper.
>
> Thanks, Area Chair

---

### Author Rebuttal · Authors · 2023-08-08

We appreciate the reviewers' valuable comments. Since some reviewers question the theoretical difference between the proposed method and the existing methodologies, e.g., [Kerrigan et al., 2023, Lim et al., 2023, Pidstrigach et al., 2023], etc., we would like to address those issues. We promise to elaborate on it in more detail in the camera-ready version.

First, [Kerrigan et al., 2023] introduced a **discrete-time model** in infinite-dimensional space, effectively generalizing the DDPM [Ho et al., 2020]. This work serves as one of the key inspirations for our study. Within that paper, the authors acknowledged a remaining challenge for functional diffusion models, stating: *'A remaining challenge for functional diffusion models is to consider the continuous-time limit and elucidating connections with score-based methods [Song et al., 2021].'* It is worth noting that the our main result, Theorem 2.1, provides a comprehensive response to this challenge.

Next, the works of [Lim et al., 2023; Pidstrigach et al., 2023] focus on continuous-time models, contrary to [Kerrigan et al., 2023]. Both methods and our approach start on a standard theoretical setting, utilizing Cameron-Martin space and Gaussian measure theory on Hilbert spaces. It is crucial to recognize, however, that the theoretical results in [Lim et al., 2023; Pidstrigach et al., 2023] are confined to SDEs with **constant-time coefficients** (i.e., $B_{t}=I$ and $G_{t}=\sqrt{2}I$, as detailed in [Section 3, Pidstrigach et al., 2023] and [Section 4.1, Lim et al., 2023]). In [Section 4.4, Lim et al., 2023], the multiple noise scale case is revealed, yet it is constrained to discrete-time models only. It is because their approach is largely grounded at theory of linear semigroups [Da Prato & Zabczyk, 2014], which is applicable to a wide range of differential equations but has a limitation when dealing with variable coefficients. [Hagemann et al., 2023] proposes using a discretized reverse process with time-dependent coefficients and [Franzese et al.,2023] considers constant-time diffusion coefficient based on [Millet et al., 1989], which do not fully recover the SDE framework in [Song et al., 2021]. [Phillips et al., 2022; Dutordoir et al., 2022] are also infinite-dimensional approaches, but they differ significantly from our methodology in theoretical aspects. [Dutordoir et al., 2022] is a discrete-time model proposed to generalize neural processes, but it utilizes white noise, which causes an ill-posed problem in infinite-dimensional spaces. [Phillips et al., 2022] proposed a continuous-time model based on spectral decomposition and demonstrated its effectiveness on multi-modal datasets, but it does not include a general SDE framework and SPDE approach.

On the contrary, our approach is grounded in the study of variational approach to stochastic evolution equations [Krylov & Rozovskii, 1981, 2007] and forward and backward Kolmogorov equations with functional derivatives [Dalecky & Fomin, 1991; Belopolskaya & Dalecky, 2012], which serves as a more established approach to identifying relations between time-dependent evolution operators and invariant measures of stochastic equations in Hilbert spaces [Y. Belopolskaya, 2020]. It is noteworthy that proof of Theorem 2.1 (Appendix B.2) can be viewed as a generalization of the classical techniques employed by [B.D.O Anderson, 1982] and [Haussmann & Pardoux, 1987] in finite-dimensional spaces to general Hilbert spaces; consequently it is able to fully recover the SDE framework proposed from [Song et al., 2021], and can utilize time-dependent coefficients, thereby enabling cosine-beta scheduling.

We also point out that we newly design the time-conditioned FNO (Figure 2 in Section 3) for HDM-SDE by adding a positional embedding and a linear layer to embed the time variable, which has a different structure from the original FNO architecture [Zongyi Li et al., 2020]. This modification is beneficial, especially for functional and motion-generation tasks, as shown in Tables 1 and A.1.

---

[B.D.O Anderson, 1982] Reverse time diffusion equation models. *Stochastic Processes and their Applications,* 1982.

[Y. Belopolskaya, 2020]. Invariant measures of diffusion processes in Hilbert spaces and Hilbert manifolds, 2020.

[Belopolskaya & Dalecky, 2012] Stochastic equations and differential geometry, 2012.

[Dalecky & Fomin, 1991] Measures and differential equations in infinite-dimensional space, 1991.

[Da Prato & Zabczyk, 2014] Stochastic equations in infinite dimensions, 2014.

[Dutordoir et al., 2022] Neural Diffusion Processes, *arXiv*, 2022.

[Franzese et al.,2023] Continuous-time functional diffusion processes. *arXiv*, 2023.

[Hagemann et al., 2023] Multilevel Diffusion: Infinite Dimensional Score-Based Diffusion Models for Image Generation. *arXiv*, 2023.

[Haussmann & Pardoux, 1987] Time reversal of diffusions. *Annals of Probability*, 1987.

[Ho et al., 2020] Denoising diffusion probabilistic models. *NeurIPS*, 2020.

[Kerrigan et al., 2023] Diffusion generative models in infinite dimensions. *AISTATS*, 2023.

[Krylov & Rozovskii, 1981, 2007] Stochastic evolution equations, 1981 (Russian) 2007 (English).

[Lim et al., 2023] Score based diffusion models in function space. *arXiv*, 2023.

[Millet et al., 1989] Time reversal for infinite-dimensional diffusions. *Probability theory and related fields,* 1989.

[Phillips et al., 2022] Spectral diffusion processes. *NeurIPS Workshop on SBM*, 2022.

[Pidstrigach et al., 2023] Infinite-dimensional diffusion models for function spaces. *arXiv*, 2023.

[Song et al., 2021] Score-based generative modeling through stochastic differential equations. *ICLR*, 2021.

[Zongyi Li et al., 2020] Fourier neural operator for parametric partial differential equations, *ICLR*, 2021.

---

### Decision · Program_Chairs · 2023-09-21

**Decision:**

Accept (spotlight)

**Comment:**

This paper presents a novel theoretical framework called Hilbert Diffusion Models (HDM) that unifies time-reversal diffusion models and stochastic differential equations in infinite-dimensional Hilbert spaces.

The reviewers unanimously gave scores of 7 (Accept) in their initial reviews, with reasonable confidence levels. The authors provided an extensive rebuttal comprehensively addressing the reviewers' questions and concerns. In particular, they clarified the differences from related prior work, especially the ability to handle time-dependent coefficients in a continuous-time setting. They also provided additional motivation and use cases for infinite-dimensional models. The reviewers seemed satisfied with the clarifications and two reviewers increased their score accordingly.

The paper is technically solid, with reviewers praising the strong theoretical foundations. The exposition and experiments are also of good quality. The feedback from reviewers was largely constructive, aimed at improving the contextualization and motivation. I believe the authors have adequately addressed these concerns in their rebuttal.

In conclusion, I recommend accepting this paper given its solid technical contributions, promising empirical results on functional data and human motion modeling, and the authors' responsiveness to feedback about framing and comparisons. The reviews validate it being an impactful advancement worthy of inclusion and presentation at the conference. I expect the camera-ready version to be improved based on the reviewer discussion.